# Multimodal deep learning for Alzheimer's disease dementia assessment

Shangran Qiu[1,2,24], Matthew I. Miller[1,24], Prajakta S. Joshi[3,4,5], Joyce C. Lee[1], Chonghua Xue[1,3], Yunruo Ni[1], Yuwei Wang[1], Ileana De Anda-Duran [6], Phillip H. Hwang [3], Justin A. Cramer[7], Brigid C. Dwyer[8], Honglin Hao[9], Michelle C. Kaku[8], Sachin Kedar[10,11,12], Peter H. Lee[13], Asim Z. Mian[14], Daniel L. Murman[10], Sarah O'Shea[8], Aaron B. Paul[13], Marie-Helene Saint-Hilaire[8], E. Alton Sartor[8], Aneeta R. Saxena[8], Ludy C. Shih [8], Juan E. Small [13], Maximilian J. Smith[13], Arun Swaminathan[10], Courtney E. Takahashi[8], Olga Taraschenko[10], Hui You[15], Jing Yuan [9], Yan Zhou[9], Shuhan Zhu[8], Michael L. Alosco[8,16], Jesse Mez[5,8,16], Thor D. Stein[16,17,18,19], Kathleen L. Poston [20], Rhoda Au[3,5,8,16,21] & Vijaya B. Kolachalama [1,16,22,23✉]

Worldwide, there are nearly 10 million new cases of dementia annually, of which Alzheimer's disease (AD) is the most common. New measures are needed to improve the diagnosis of individuals with cognitive impairment due to various etiologies. Here, we report a deep learning framework that accomplishes multiple diagnostic steps in successive fashion to identify persons with normal cognition (NC), mild cognitive impairment (MCI), AD, and non-AD dementias (nADD). We demonstrate a range of models capable of accepting flexible combinations of routinely collected clinical information, including demographics, medical history, neuropsychological testing, neuroimaging, and functional assessments. We then show that these frameworks compare favorably with the diagnostic accuracy of practicing neurologists and neuroradiologists. Lastly, we apply interpretability methods in computer vision to show that disease-specific patterns detected by our models track distinct patterns of degenerative changes throughout the brain and correspond closely with the presence of neuropathological lesions on autopsy. Our work demonstrates methodologies for validating computational predictions with established standards of medical diagnosis.

A full list of author affiliations appears at the end of the paper.

Alzheimer's disease (AD) is the most common cause of dementia worldwide[1], and future expansions in caseload due to an aging population are likely to accentuate existing needs for health services[2]. This increase in clinical demand will likely contribute to an already considerable burden of morbidity and mortality among the elderly[3], thus requiring improvements in the treatment and timely identification of AD. Significant efforts have been made in recent years towards the development of cerebrospinal fluid (CSF) biomarkers[4], as well as advanced imaging modalities such as amyloid and tau positron emission tomography (PET)[5–8]. Furthermore, novel generations of disease-modifying therapies for AD are now coming into clinical purview[9], though their efficacy remains controversial. Despite this progress, many emerging diagnostic and treatment modalities remain limited to research contexts, and the backbone of antemortem diagnosis remains traditional clinical assessment, neuropsychological testing[10], and magnetic resonance imaging (MRI)[11]. Mild cognitive impairment (MCI), a prodromal stage of dementia, may also be a subtle early presentation of AD whose diagnosis similarly requires significant clinical acumen from qualified specialists. Complicating matters is the presence of a multitude of other non-Alzheimer's disease dementia (nADD) syndromes whose clinical presentations often overlap with AD. Thus, common causes of dementias outside of AD such as vascular dementia (VD), Lewy body dementia (LBD), and frontotemporal dementia (FTD) widen the differential diagnosis of neurodegenerative conditions and contribute to variability in diagnostic sensitivity and specificity[12].

Reliably differentiating between normal cognitive aging, MCI, AD, and other dementia etiologies requires significant clinical acumen from qualified specialists treating memory disorders, yet timely access to memory clinics is often limited for patients and families. This is a major problem in remote, rural regions within developed countries and in still economically developing nations, where there is a dearth of specialized practitioners. Furthermore, the need for skilled clinicians is rising, yet the United States is facing a projected shortage of qualified clinicians, such as neurologists, in coming decades[13,14]. As increasing clinical demand intersects with a diminishing supply of medical expertise, machine learning methods for aiding neurologic diagnoses have begun to attract interest[15]. Complementing the high diagnostic accuracy reported by other groups[16], we have previously reported interpretable deep learning approaches capable of distinguishing participants with age-appropriate normal cognition (NC) from those with AD using magnetic resonance imaging (MRI) scans, age, sex, and mini-mental state examination (MMSE)[17]. Others have also demonstrated the efficacy of deep learning in discriminating AD from specific types of nADD[18–20]. However, clinical evaluation of persons presenting in a memory clinic involves consideration of multiple etiologies of cognitive impairment. Therefore, the ability to successfully differentiate between NC, MCI, AD, and nADD across diverse study cohorts in a unified framework remains to be developed.

In this study, we report the development and validation of a deep learning framework capable of accurately classifying subjects with NC, MCI, AD, and nADD in multiple cohorts of participants with different etiologies of dementia and varying levels of cognitive function (Table 1, Fig. 1). Using data from the National Alzheimer's Coordinating Center (NACC)[21,22], we developed and externally validated models capable of classifying cognitive status using MRI, non-imaging variables, and combinations thereof. To validate our approach, we demonstrated comparability of our model's accuracy to the diagnostic performance of a team of practicing neurologists and neuroradiologists. We then leveraged SHapley Additive exPlanations (SHAP)[23], to link computational predictions with well-known anatomical and pathological markers of neurodegeneration. Our strategy provides evidence that automated methods driven by deep learning may approach clinical standards of accurate diagnosis even amidst heterogeneous datasets.

## Results

We divided the process of differential diagnosis into staged tasks. The first, which we refer to as the COG task, labeling persons as having either NC, MCI, or dementia (DE) due to any cause. Of note, the COG task may be seen as comprising three separate binary classification subtasks: (i) $COG_{NC}$ task: Separation of NC and MCI/DE cases (ii) $COG_{MCI}$ task: Separation of MCI from NC/DE cases, and (iii) $COG_{DE}$ task: Separation of DE from NC/MCI cases. After completion of the overall COG task, we next formulated the ADD task, in which we assigned all persons labeled as DE to a diagnosis of either AD or nADD. Successive completion of the COG and the ADD tasks allowed execution of an overarching 4-way classification that fully delineated NC, MCI, AD, and nADD cases (See Supplementary Information: Glossary of Tasks, Models, and Metrics).

We also created three separate models: (i) MRI-only model: A convolutional neural network (CNN) that internally computed a continuous DEmentia MOdel (DEMO) score to complete the COG task, as well as an ALZheimer's (ALZ) score to complete the ADD task. (ii) Non-imaging model: A traditional machine learning classifier that took as input only scalar-valued clinical variables from demographics, past medical history, neuropsychological testing, and functional assessments. As in the MRI-only model, the non-imaging model also computed the DEMO and the ALZ scores from which the COG and the ADD tasks could be completed. We tested multiple machine learning architectures for these purposes and ultimately selected a CatBoost model as our final non-imaging model architecture. (iii) Fusion model: This framework linked a CNN to a CatBoost model. With this approach, the DEMO and the ALZ scores computed by the CNN were recycled and used alongside available clinical variables. The CatBoost model then recalculated these scores in the context of the additional non-imaging information. We provide definitions of our various prediction tasks, cognitive metrics, and model types within the Supplementary Information. Further details of model design may be found within the Methods.

**Assessment for confounding**. We used two-dimensional t-distributed stochastic neighbor embedding (tSNE) to assess for the presence of confounding relationships between disease status and certain forms of metadata. Using this approach, we observed no obvious clustering of post-processed MRI embeddings among the eight cohorts used for testing of MRI-only models (Fig. 2a, b). Within the NACC cohort, we also observed no appreciable clustering based on individual Alzheimer's Disease Research Centers (ADRCs, Fig. 2c, d) or scanner manufacturer (Fig. 2e, f). Relatedly, although tSNE analysis of CNN hidden layer activations did yield clustering of NACC data points (Fig. 2b), this was an expected phenomenon given the selection of NACC as our cohort for model training. Otherwise, we appreciated no obvious conglomeration of embeddings from hidden layer activations due to specific ADRCs (Fig. 2d) or scanner manufacturers (Fig. 2f). Lastly, Mutual Information Scores (MIS) computed from the NACC cohort indicated negligible correlation of diagnostic labels (NC, MCI, AD, and nADD) between specific scanner manufacturers (MIS = 0.010, Fig. 2g) and ADRCs (MIS = 0.065, Fig. 2h).

**Deep learning model performance**. We observed that our fusion model provided the most accurate classification of cognitive status for NC, MCI, AD and nADD across a range of clinical

**Table 1 Study population and characteristics.**

| Dataset (group) [subjects] | Age mean ± std | Gender male (percent) | Education in years mean ± std | Race (White; Black; Asian; Indian; Pacific; Multi-race) | ApoE4 positive (percent) | MMSE mean ± std | CDR mean ± std | MOCA mean ± std |
|---|---|---|---|---|---|---|---|---|
| **ADNI** | | | | | | | | |
| NC [n = 481] | 74.26 ± 6.00 | 235 (48.86%) | 16.34 ± 2.67 | (436, 36, 8, 1, 0, 3)^ | 138 (29.61%)^ | 29.05 ± 1.12 | 0.00 ± 0.00^ | 25.71 ± 2.59^ |
| MCI [n = 971] | 72.84 ± 7.71 | 572 (58.91%) | 15.94 ± 2.81 | (903, 34, 15, 2, 2, 12)^ | 438 (47.20%)^ | 27.62 ± 1.81 | 0.50 ± 0.04 | 23.18 ± 3.23^ |
| AD [n = 369] | 74.91 ± 7.84 | 203 (55.01%) | 15.18 ± 2.97 | (343, 15, 7, 0, 0, 4) | 229 (64.33%)^ | 23.19 ± 2.11 | 0.77 ± 0.26 | 16.80 ± 4.50^ |
| p value | 2.565e-6 | 1.364e-3 | 1.872e-8 | 1.132e-1 | 3.117e-22 | <1.0e-200 | <1.0e-200 | 1.010e-116 |
| **NACC** | | | | | | | | |
| NC [n = 2524] | 69.82 ± 9.93^ | 871 (34.51%) | 15.92 ± 2.95^ | (2120, 303, 55, 31, 2, 0)^ | 599 (29.95%)^ | 28.98 ± 1.31^ | 0.06 ± 0.16^ | 26.80 ± 2.44^ |
| MCI [n = 1175] | 74.01 ± 8.74^ | 555 (47.23%) | 15.36 ± 3.35^ | (965, 160, 25, 17, 1, 0)^ | 322 (38.66%)^ | 26.79 ± 2.51^ | 0.46 ± 0.18^ | 22.68 ± 3.41^ |
| AD [n = 948] | 74.97 ± 9.13^ | 431 (45.46%) | 14.64 ± 3.64^ | (816, 85, 23, 11, 0, 0)^ | 346 (52.19%)^ | 20.48 ± 5.69^ | 1.02 ± 0.60^ | 15.39 ± 5.44^ |
| nADD [n = 175] | 69.35 ± 10.84^ | 110 (62.86%) | 14.86 ± 3.60^ | (161, 10, 2, 1, 0, 0)^ | 34 (25.95%)^ | 22.23 ± 6.14^ | 1.07 ± 0.70^ | 17.53 ± 6.35^ |
| p value | 1.145e-56 | 1.130e-22 | 1.846e-25 | 5.349e-2 | 8.026e-49 | <1.0e-200 | <1.0e-200 | <1.0e-200 |
| **NIFD** | | | | | | | | |
| NC [n = 124] | 63.21 ± 7.27 | 56 (45.16%) | 17.48 ± 1.87^ | (89, 0, 0, 0, 0, 3)^ | N.A. | 29.35 ± 0.76 | 0.03 ± 0.12^ | 27.58 ± 1.53^ |
| nADD [n = 129] | 63.66 ± 7.33 | 75 (58.14%) | 16.18 ± 3.29^ | (109, 1, 1, 0, 0, 4)^ | N.A. | 24.75 ± 4.54^ | 0.82 ± 0.54^ | 19.69 ± 5.72^ |
| p value | 6.266e-1 | 5.246e-2 | 2.606e-4 | 6.531e-1 | | 1.961e-23 | 4.333e-28 | 2.645e-16 |
| **PPMI** | | | | | | | | |
| NC [n = 171] | 62.74 ± 10.12 | 109 (63.74%) | 15.82 ± 2.93 | (163, 3, 2, 0, 0, 1)^ | N.A. | N.A. | N.A. | 27.51 ± 2.37^ |
| MCI [n = 27] | 68.04 ± 7.32 | 22 (81.48%) | 15.52 ± 3.08 | (24, 1, 1, 0, 0, 1) | N.A. | N.A. | N.A. | 24.69 ± 3.27^ |
| p value | 1.006e-2 | 1.115e-1 | 6.194e-1 | 2.910e-1 | | | | 3.004e-7 |
| **AIBL** | | | | | | | | |
| NC [n = 480] | 72.45 ± 6.22 | 203 (42.29%) | N.A. | N.A. | 12 (2.50%) | 28.70 ± 1.24 | 0.03 ± 0.12 | N.A. |
| MCI [n = 102] | 74.73 ± 7.11 | 53 (51.96%) | N.A. | N.A. | 12 (11.77%) | 27.10 ± 2.08 | 0.47 ± 0.14 | N.A. |
| AD [n = 79] | 73.34 ± 7.77 | 33 (41.77%) | N.A. | N.A. | 14 (17.72%) | 20.42 ± 5.46 | 0.93 ± 0.54 | N.A. |
| p value | 5.521e-3 | 1.887e-1 | | | 8.951e-9 | 4.585e-121 | 4.542e-158 | |
| **OASIS** | | | | | | | | |
| NC [n = 424] | 71.34 ± 9.43 | 164 (38.70%) | 15.79 ± 2.62^ | (53, 18, 1, 0, 0, 0)^ | 121 (29.88%)^ | 28.99 ± 1.25^ | 0.00 ± 0.02 | N.A. |
| MCI [n = 27] | 75.04 ± 7.25 | 14 (51.85%) | 15.19 ± 2.76 | (4, 1, 0, 0, 0, 0)^ | 9 (36.00%) | 28.15 ± 1.67 | 0.52 ± 0.09 | N.A. |
| AD [n = 193] | 76.01 ± 8.01 | 108 (55.96%) | 14.68 ± 3.09 | (35, 9, 0, 0, 0, 0)^ | 102 (56.98%)^ | 23.84 ± 4.17 | 0.77 ± 0.33 | N.A. |
| nADD [n = 22] | 72.64 ± 8.77 | 16 (72.73%) | 15.00 ± 2.91 | (6, 0, 0, 0, 0, 0)^ | 8 (47.06%)^ | 24.14 ± 4.69^ | 0.75 ± 0.47 | N.A. |
| p value | 5.896e-8 | 3.190e-5 | 9.665e-5 | 8.098e-1 | 1.689e-9 | 2.122e-85 | <1.0e-200 | |
| **FHS** | | | | | | | | |
| NC [n = 212] | 73.37 ± 9.63 | 112 (52.83%) | 1.79 ± 0.96* | (207, 2, 1, 0, 0, 0)^ | 42 (20.19%)^ | 28.14 ± 1.72^ | N.A. | N.A. |
| MCI [n = 75] | 76.23 ± 6.83 | 34 (45.33%) | 1.59 ± 0.98* | (73, 0, 1, 0, 0, 0)^ | 17 (23.61%)^ | 27.22 ± 2.01^ | N.A. | N.A. |
| AD [n = 17] | 78.82 ± 7.20 | 4 (23.53%) | 1.82 ± 0.92* | (17, 0, 0, 0, 0, 0) | 7 (43.75%)^ | 24.00 ± 2.13^ | N.A. | N.A. |
| nADD [n = 9] | 79.44 ± 4.17 | 5 (55.56%) | 1.00 ± 1.15* | (9, 0, 0, 0, 0, 0) | 0 (0.00%) | 22.00 ± 2.45^ | N.A. | N.A. |
| p value | 4.755e-3 | 1.032e-1 | 5.918e-2 | 9.380e-1 | 5.704e-2 | 1.211e-13 | | |
| **LBDSU** | | | | | | | | |
| NC [n = 134] | 68.77 ± 7.62 | 61 (45.52%) | 17.27 ± 2.47^ | N.A. | N.A. | N.A. | N.A. | 27.43 ± 2.23^ |
| MCI [n = 35] | 70.16 ± 8.41 | 26 (74.29%) | 16.60 ± 2.58 | N.A. | N.A. | N.A. | N.A. | 24.00 ± 3.14 |
| nADD [n = 13] | 73.42 ± 7.81 | 8 (61.54%) | 16.77 ± 2.15 | N.A. | N.A. | N.A. | N.A. | 16.69 ± 4.75 |
| p value | 1.033e-1 | 7.863e-3 | 3.243e-1 | | | | | 2.231e-30 |

Eight independent datasets were used for this study, including NACC, ADNI, AIBL, FHS, LBDSU, OASIS, NIFD, and PPMI. The NACC dataset was used to develop three separate types of models: (i) an MRI-only CNN model that exclusively utilized imaging data, (ii) non-imaging models in the form of traditional machine learning classifiers, which did not use any MRI data, and (iii) a fusion model that combined imaging and non-imaging data within a hybrid architecture joining a CNN to a CatBoost model. The MRI-only model was validated across all eight cohorts, whereas external validation of non-imaging and fusion models was performed only on the OASIS cohort. All the MRI scans considered for this study were performed on individuals within ± 6 months from the date of clinical diagnosis. The p value for each dataset indicates the statistical significance of inter-group differences per column. We used one-way ANOVA and χ2 tests for continuous and categorical variables, respectively.
NC normal cognition, MCI mild cognitive impairment, AD Alzheimer's disease dementia, nADD non-Alzheimer's disease dementia, NA not available.
*FHS education code: 0 = high school did not graduate, 1= high school graduate, 2 = some college graduate, 3 = college graduate.
The symbol ^ indicates that data was not available for some subjects.

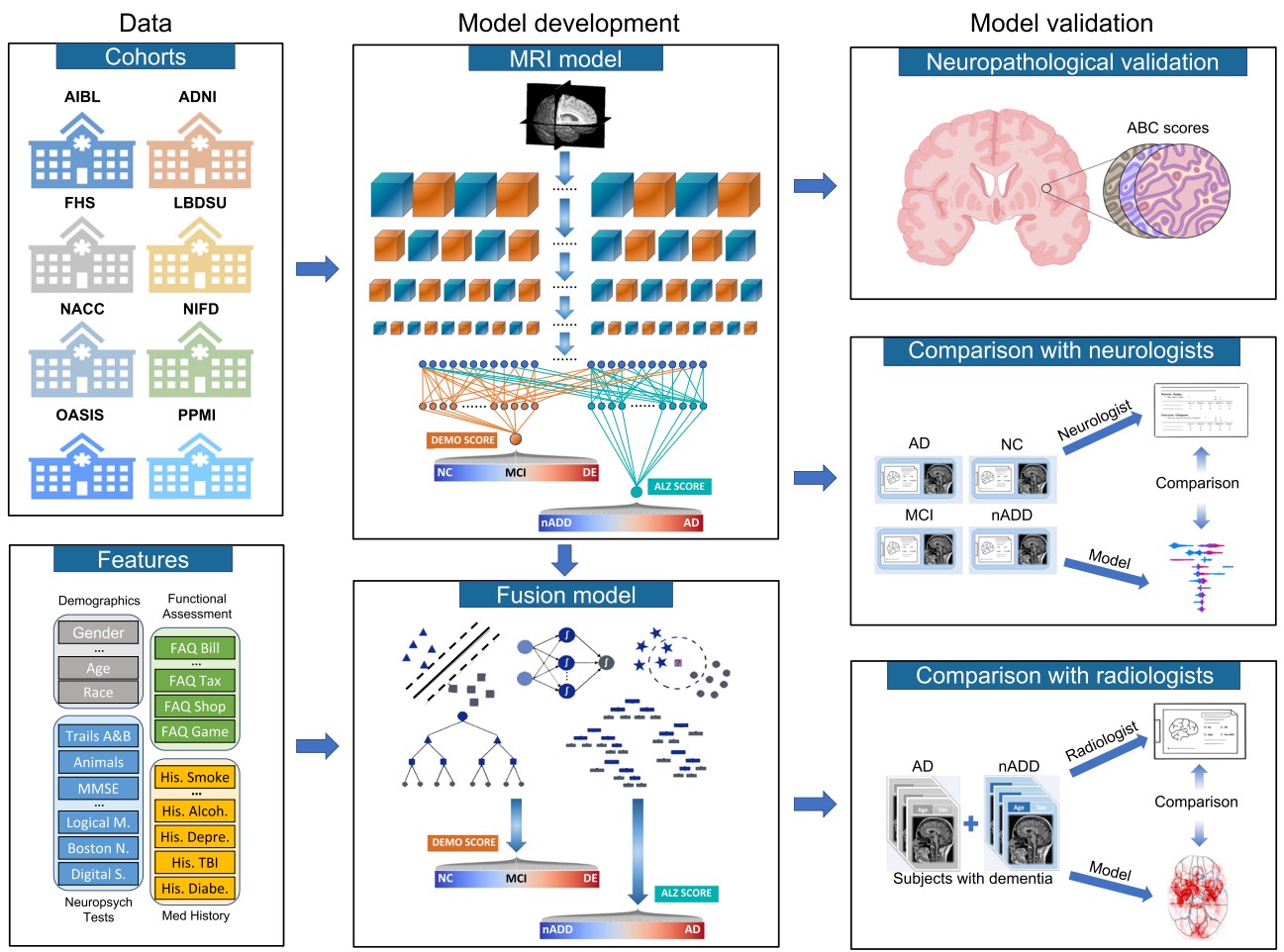

**Fig. 1 Modeling framework and overall strategy.** Multimodal data including MRI scans, demographics, medical history, functional assessments, and neuropsychological test results were used to develop deep learning models on various classification tasks. Eight independent datasets were used for this study, including NACC, ADNI, AIBL, FHS, LBDSU, NIFD, OASIS, and PPMI. We selected the NACC dataset to develop three separate models: (i) an MRI-only CNN model (ii) non-imaging models in the form of traditional machine learning classifiers, which did not use any MRI data (iii) a fusion model combining imaging and non-imaging data within a hybrid architecture joining a CNN to a CatBoost model. The MRI-only model was validated across all eight cohorts, whereas external validation of non-imaging and fusion models was performed only on OASIS. First, T1-weighted MRI scans were input to a CNN to calculate a continuous DEmentia MOdel (DEMO) score to assess cognitive status on a 0 to 2 scale, where "0" indicated NC "1" indicated MCI and "2" indicated DE. DEMO scores were converted to class labels using an optimal thresholding algorithm, with these assignments constituting the COG task. For individuals with DE diagnosis, the multi-task CNN model simultaneously discriminated their risk of having AD versus nADD, a classification that we refer to as the ADD task. We denoted the probability of AD diagnosis as the ALZheimer (ALZ) score. Both MRI-derived DEMO scores and ALZ scores were then input alongside non-imaging variables to various machine learning classifiers to form fusion models, which then predicted outcomes on the COG and ADD tasks, respectively. A portion of cases with confirmed dementia ($n = 50$) from the NACC testing cohort was randomly selected for direct comparison of the fusion model with an international team of practicing neuroradiologists. Both the model and neuroradiologists completed the ADD task using available MRI scans, age, and gender. Additionally, a portion of NACC cases ($n = 100$) was randomly selected to compare the fusion model performance to practicing neurologists, with both the model and clinicians having access to a common set of multimodal data. Lastly, model predictions were compared with neuropathology grades from NACC, ADNI and FHS cohorts ($n = 110$).

diagnosis tasks (Table 2). We found strong model performance on the $\text{COG}_{\text{NC}}$ task between both the NACC test set (Fig. 3a, Row 1) and an external validation set (OASIS; Fig. 3b, Row 1) as indicated by area under the receiver operating characteristic (AUC) curve values of 0.945 [95% confidence interval (CI): 0.939, 0.951] and 0.959 [CI: 0.955, 0.963], respectively. Similar values for area under precision-recall (AP) curves were also observed, yielding 0.946 [CI: 0.940, 0.952] and 0.969 [CI: 0.964, 0.974], respectively. Such correspondence between AUC and AP performance supports robustness to class imbalance across datasets. In the $\text{COG}_{\text{DE}}$ task, comparable results were also seen, as the fusion model yielded respective AUC and AP scores of 0.971 [CI: 0.966, 0.976]/0.917 [CI: 0.906, 0.928] (Fig. 3a, Row 2) in the NACC dataset and 0.971 [CI: 0.969, 0.973]/0.959 [CI: 0.957,

0.961] in the OASIS dataset (Fig. 3b, Row 2). Conversely, classification performance dropped slightly for the ADD task, with respective AUC/AP values of 0.773 [CI: 0.712, 0.834]/0.938 [CI: 0.918, 0.958] in the NACC dataset (Fig. 3a, Row 3) and 0.773 [CI: 0.732, 0.814]/0.965 [CI: 0.956, 0.974] in the OASIS dataset (Fig. 3b, Row 3).

Relative to the fusion model, we observed moderate performance reductions across classifications in our MRI-only model. For the $\text{COG}_{\text{NC}}$ task, the MRI-only framework yielded AUC and AP scores of 0.844 [CI: 0.832, 0.856]/0.830 [CI: 0.810, 0.850] (NACC) and 0.846 [CI: 0.840, 0.852]/0.890 [CI: 0.884, 0.896] (OASIS). Model results were comparable on the $\text{COG}_{\text{DE}}$ task, in which the MRI-only model achieved respective AUC and AP scores of 0.869 [CI: 0.850, 0.888]/0.712 [CI: 0.672, 0.752] (NACC)

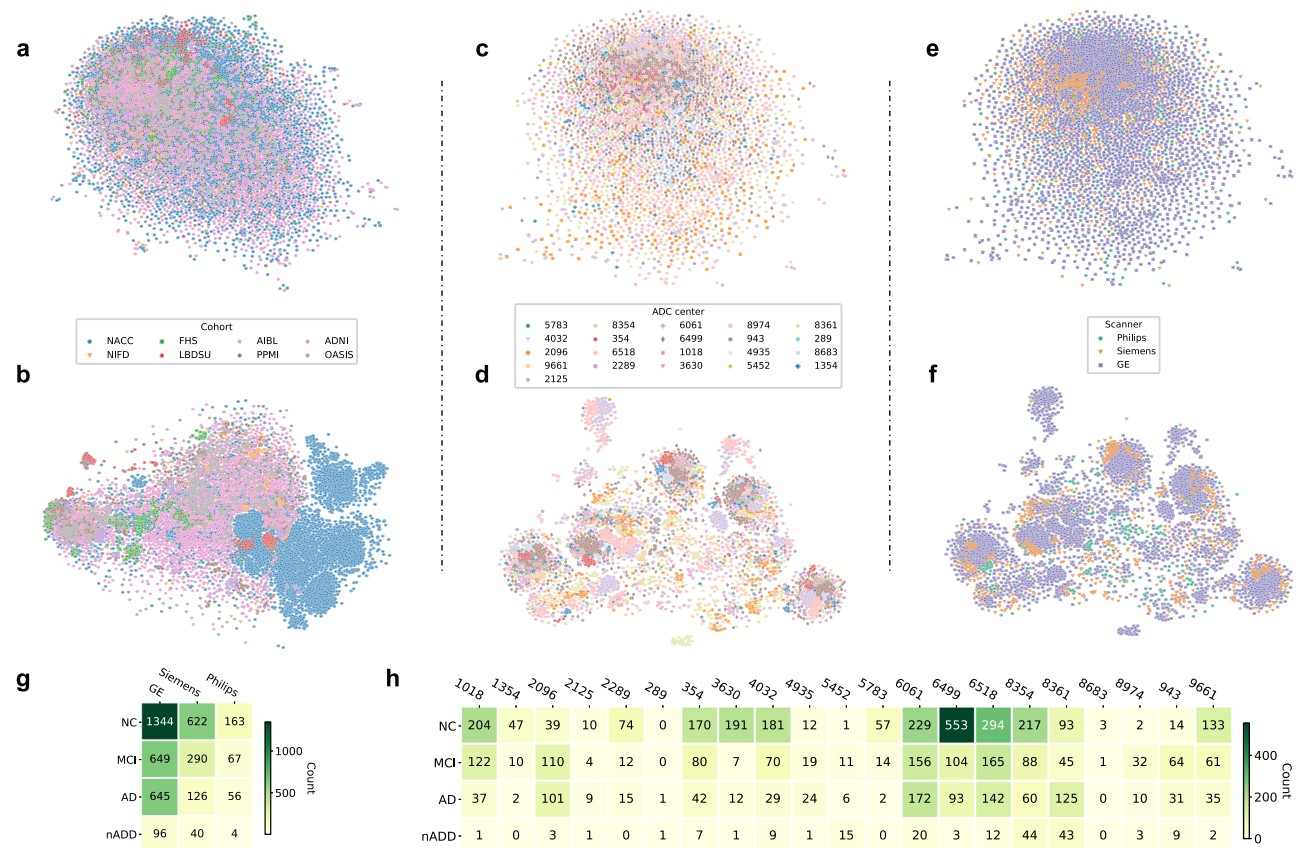

**Fig. 2 Site- and scanner-specific observations.** Unsupervised clustering of post-processed MRIs and hidden layer activations assessed for systematic biases in input data and model predictions, respectively. **a** Two-dimensional (2D) t-distributed stochastic neighbor embedding (tSNE) embeddings of downsampled MRI scans are shown. The downsampling was performed on the post-processed MRI scans using spline interpolation with a downsampling factor of 8 on each axis. Individual points represent MRIs from a single subject and are colored according to their original cohort (either NACC, ADNI, AIBL, FHS, LBDSU, NIFD, OASIS, or PPMI). **b** We demonstrate 2D tSNEs of hidden-layer activations from the penultimate CNN hidden layer. Individual points correspond to internal representations of MRI scans during testing and are colored by cohort label. **c** Plot of 2D tSNE embeddings of downsampled MRI scans from the NACC dataset is shown. Individual points representing MRI scans are colored by the unique identifier of one of twenty-one Alzheimer Disease Research Centers (ADRCs) that participate in the NACC collaboration. **d** tSNE embeddings for penultimate layer activations colored by ADRC ID are shown. **e** Plot of 2D tSNE embeddings of downsampled MRI scans from the NACC dataset is shown. Embeddings in this plot are the same as those in **c** but colored according to the manufacturer of the scanner used to acquire each MRI, either General Electric (GE), Siemens, or Philips. **f** Plot of 2D tSNE of penultimate layer activations is shown for cases in the NACC dataset. Embeddings are equivalent to those visualized in **d** but are now colored by the manufacturer of the scanner used for image acquisition. **g** A tabular representation of disease category counts by manufacturer is presented. Only cases from the NACC dataset are included. We provide the Mutual Information Score (MIS) to quantify the correlation between disease type and scanner manufacturer. **h** We also provided a tabular representation of disease category counts stratified by ADRC ID in the NACC dataset. MIS is once again shown to quantify the degree of correlation between diagnostic labels and individual centers participating in the NACC study. Source data are provided as a Source Data file.

and 0.858 [CI: 0.854, 0.862]/0.772 [CI: 0.763, 0.781] (OASIS). For the ADD task as well, the results of the MRI-only model were approximately on par with those of the fusion model, giving respective AUC and AP scores of 0.766 [CI: 0.734, 0.798]/0.934 [CI: 0.917, 0.951] (NACC) and 0.694 [CI: 0.659,0.729]/0.942 [CI: 0.931, 0.953] (OASIS). For both fusion and MRI-only models, we also reported ROC and PR curves for the ADD task stratified by nADD subtypes in the Supplementary Information (Figs. S1 and S2).

Interestingly, we note that a non-imaging model generally yielded similar results to those of both the fusion and MRI-only models. Specifically, a CatBoost model trained for the $COG_{NC}$ task gave AUC and AP values 0.936 [CI: 0.929, 0.943] /0.936 [CI: 0.930, 0.942] (NACC), as well as 0.959 [CI: 0.957, 0.961]/0.972 [CI: 0.970, 0.974] (OASIS). Results remained strong for the $COG_{DE}$ task, with AUC/PR pairs of 0.962 [CI: 0.957, 0.967]/0.907 [0.893, 0.921] (NACC) and 0.971 [CI: 0.970, 0.972]/0.955 [CI:

0.953, 0.957] (OASIS). For the ADD task, the non-imaging model resulted in respective AUC/PR scores of 0.749 [CI: 0.691, 0.807]/ 0.935 [CI: 0.919, 0.951] (NACC) and 0.689 [CI: 0.663, 0.715]/ 0.947 [CI: 0.940, 0.954] (OASIS). A full survey of model performance metrics across all classification tasks may be found in the Supplementary Information (Tables S1–S4). Performance of the MRI-only model across all external datasets is demonstrated via ROC and PR curves (Fig. S3).

To assess the contribution of various imaging and non-imaging features to classification outcomes, we calculated fifteen features with highest mean absolute SHAP values for the COG (Fig. 3c) and the ADD prediction tasks using the fusion model (Fig. 3d). Though MMSE score was the primary discriminative feature for the COG task, the DEMO score derived from the CNN portion of the model ranked third in predicting cognitive status. Analogously, the ALZ score derived from the CNN was the most salient feature in solving the ADD task. Interestingly, the relative

**Table 2 Fusion model performance.**

| | COG | COG$_{NC}$ | COG$_{MCI}$ | COG$_{DE}$ | ADD | 4-way |
|---|---|---|---|---|---|---|
| **NACC** | | | | | | |
| Accuracy | 0.804 ± 0.011 [0.790–0.818] | 0.872 ± 0.009 [0.861–0.883] | 0.809 ± 0.011 [0.795–0.823] | 0.926 ± 0.004 [0.921–0.931] | 0.847 ± 0.013 [0.831–0.863] | 0.777 ± 0.011 [0.763–0.791] |
| F-1 | 0.772 ± 0.012 [0.757–0.787] | 0.880 ± 0.009 [0.869–0.891] | 0.592 ± 0.023 [0.563–0.621] | 0.843 ± 0.010 [0.831–0.855] | 0.914 ± 0.008 [0.904–0.924] | 0.601 ± 0.017 [0.580–0.622] |
| Sensitivity | 0.771 ± 0.013 [0.755–0.787] | 0.893 ± 0.020 [0.868–0.918] | 0.569 ± 0.046 [0.512–0.626] | 0.851 ± 0.025 [0.820–0.882] | 0.968 ± 0.019 [0.944–0.992] | 0.602 ± 0.013 [0.586–0.618] |
| Specificity | 0.895 ± 0.006 [0.888–0.902] | 0.850 ± 0.021 [0.824–0.876] | 0.887 ± 0.022 [0.860–0.914] | 0.949 ± 0.007 [0.940–0.958] | 0.189 ± 0.074 [0.097–0.281] | 0.914 ± 0.005 [0.908–0.920] |
| MCC | 0.670 ± 0.018 [0.648–0.692] | 0.744 ± 0.018 [0.722–0.766] | 0.470 ± 0.027 [0.436–0.504] | 0.796 ± 0.012 [0.781–0.811] | 0.249 ± 0.097 [0.129–0.369] | 0.534 ± 0.025 [0.503–0.565] |
| **OASIS** | | | | | | |
| Accuracy | 0.754 ± 0.047 [0.696–0.812] | 0.852 ± 0.037 [0.806–0.898] | 0.765 ± 0.050 [0.702–0.827] | 0.890 ± 0.006 [0.883–0.898] | 0.879 ± 0.013 [0.863–0.895] | 0.730 ± 0.045 [0.675–0.786] |
| F-1 | 0.610 ± 0.023 [0.581–0.638] | 0.873 ± 0.037 [0.827–0.919] | 0.156 ± 0.024 [0.126–0.186] | 0.800 ± 0.015 [0.781–0.818] | 0.935 ± 0.008 [0.925–0.945] | 0.468 ± 0.020 [0.443–0.493] |
| Sensitivity | 0.670 ± 0.025 [0.639–0.701] | 0.807 ± 0.066 [0.725–0.888] | 0.526 ± 0.092 [0.412–0.640] | 0.678 ± 0.025 [0.648–0.709] | 0.965 ± 0.023 [0.936–0.993] | 0.514 ± 0.010 [0.501–0.527] |
| Specificity | 0.900 ± 0.013 [0.883–0.916] | 0.932 ± 0.016 [0.912–0.952] | 0.775 ± 0.055 [0.707–0.843] | 0.992 ± 0.003 [0.988–0.995] | 0.127 ± 0.083 [0.024–0.231] | 0.917 ± 0.009 [0.906–0.929] |
| MCC | 0.536 ± 0.029 [0.500–0.573] | 0.716 ± 0.055 [0.648–0.785] | 0.142 ± 0.034 [0.099–0.185] | 0.750 ± 0.014 [0.734–0.767] | 0.128 ± 0.060 [0.053–0.204] | 0.411 ± 0.022 [0.383–0.438] |

Performance of the fusion (CNN + CatBoost) model on the NACC test set and the OASIS cohort is shown. For each model, we reported accuracy, sensitivity, specificity, F-1 score, and Matthew's Correlation Coefficient (MCC) for all diagnostic tasks. In both datasets, we report classification performance on the overall COG task (i.e., the full classification of NC, MCI, and DE cases) as well as classification performance for constituent subtasks, including the binary classification of NC vs. non-NC (COG$_{NC}$ column), MCI vs. non-MCI (COG$_{MCI}$ column), and DE vs. non-DE (COG$_{DE}$ column). We also report metrics for the ADD task (i.e., the binary classification of AD and nADD following internal designation of DE in the COG task). By inferring the COG task first and the ADD task second, the model generated a 4-way classification of NC, MCI, AD, nADD (4-way column).

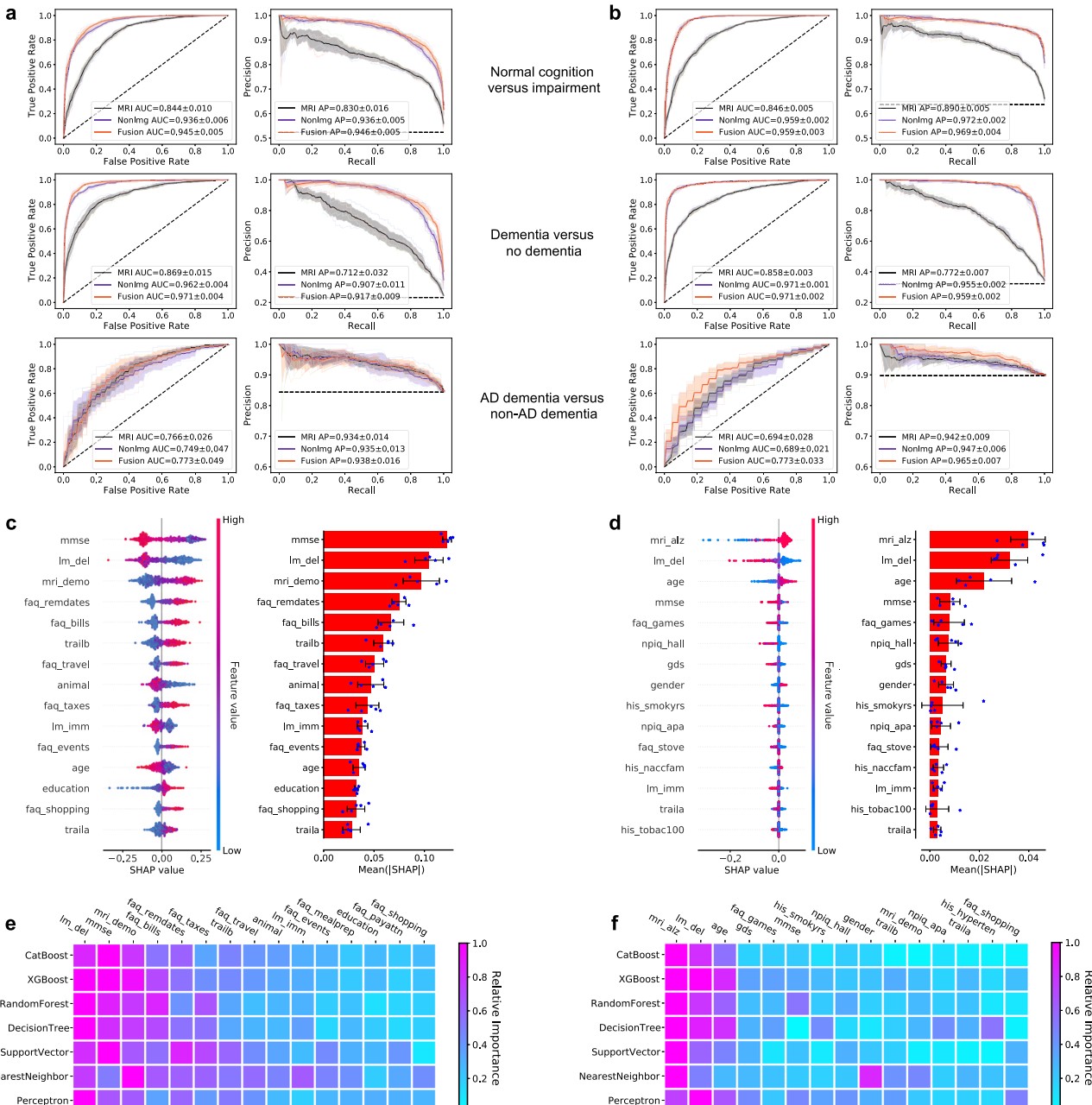

**Fig. 3 Performance of the deep learning models. a**, **b** ROC curves showing true positive rate versus false positive rate and PR curves showing the positive predictive value versus sensitivity on the **a** NACC test set and **b** OASIS dataset. The first row in **a** and **b** denotes the performance of the MRI-only model, the non-imaging model, and the fusion model (CNN + CatBoost) trained to classify cases with NC from those without NC (COG$_{NC}$ task). The second row shows ROC and PR curves of the MRI-only model, the non-imaging model, and the fusion model for the COG$_{DE}$ task aimed at distinguishing cases with DE from those who do not have DE. The third row illustrates performance of the MRI-only model, the non-imaging model, and the fusion model focused on discriminating AD from nADD. For each curve, mean AUC was computed. In each plot, the mean ROC/PR curve and standard deviation are shown as bolded lines and shaded regions, respectively. The dotted lines in each plot indicate the classifier with the random performance level. **c**, **d** Fifteen features with highest mean absolute SHAP values from the fusion model are shown for the COG and ADD tasks, respectively across cross-validation rounds ($n = 5$). Error bars overlaid on bar plots are centered at the mean of the data and extend $+/-$ one standard deviation. For each task, the MRI scans, demographic information, medical history, functional assessments, and neuropsychological test results were used as inputs to the deep learning model. The left plots in **c** and **d** illustrate the distribution of SHAP values and the right plots show the mean absolute SHAP values. All the plots in **c** and **d** are organized in decreasing order of mean absolute SHAP values. **e**, **f** For comparison, we also constructed traditional machine learning models to predict cognitive status and AD status using the same set of features used for the deep learning model, and the results are presented in **e** and **f**, respectively. The heat maps show fifteen features with the highest mean absolute SHAP values obtained for each model. Source data are provided as a Source Data file.

importance of features remained largely unchanged when a variety of other machine learning classifiers were substituted to the fusion model in lieu of the CatBoost model (Fig. 3e, f). This consistency indicated that our prediction framework was robust

to the specific choice of model architecture, and instead relied on a consistent set of clinical features to achieve discrimination between NC, MCI, AD, and nADD classes. Relatedly, we also observed that non-imaging and fusion models retained predictive

performance across a variety of input feature combinations, showing flexibility to operate across differences in information availability. Importantly, however, the addition of MRI-derived DEMO and ALZ scores improved 4-way classification performance across all combinations of non-imaging variables (Figs. S4 and S5).

**Neuroimaging signatures of AD and non-AD dementia.** The provenance of model predictions was visualized by pixel-wise SHAP mapping of hidden layers within the CNN model. The SHAP matrices were then correlated to physical locations within each subject's MRI to visualize conspicuous brain regions implicated in each stage of cognitive decline from NC to dementia (Fig. 4a). This approach allowed neuroanatomical risk mapping to distinguish regions associated with AD from those with nADD (Fig. 4b). Indeed, the direct overlay of color maps representing disease risk on an anatomical atlas derived from traditional MRI scans facilitates interpretability of the deep learning model. Also, the uniqueness of the SHAP-derived representation allows us to observe disease suggestive regions that are specific to each outcome of interest (Table S5 and Fig. S6).

A key feature of SHAP is that a single voxel or a sub-region within the brain can contribute to accurate prediction of one or more class labels. For example, the SHAP values were negative in the hippocampal region in NC participants, but they were positive in participants with dementia, underscoring the well-recognized role of the hippocampus in memory function. Furthermore, positive SHAP values were observed within the hippocampal region for AD and negative SHAP values for the nADD cases, indicating that hippocampal atrophy has direct proportionality with AD-related etiology. The SHAP values sorted according to their importance on the parcellated brain regions also further confirm the role of hippocampus and its relationship with dementia prediction, particularly in the setting of AD (Fig. 4c), as well as nADD cases (Fig. S7). In the case of nADD, the role of other brain regions such as the lateral ventricles and frontal lobes was also evident. Evidently, SHAP-based network analysis revealed pairwise relationships between brain regions that simultaneously contribute to patterns indicative of AD (Fig. 4d). The set of brain networks evinced by this analysis also demonstrate marked differences in structural changes between AD and nADD (Fig. 4e).

**Neuropathologic validation.** In addition to mapping hidden layer SHAP values to original neuroimaging, correlation of deep learning predictions with neuropathology data provided further validation of our modeling approach. Qualitatively, we observed that areas of high SHAP scores for the COG task correlated with region-specific neuropathological scores obtained from autopsy (Fig. 5a). Similarly, the severity of regional neuropathologic changes in these persons demonstrated a moderate to high degree of concordance with the regional cognitive risk scores derived from our CNN using the Spearman's rank correlation test. Of note, the strongest correlations appeared to occur within areas affected by AD pathology such as the temporal lobe, amygdala, hippocampus, and parahippocampal gyrus (Fig. 5b). Using the one-way ANOVA test, we also rejected a null hypothesis of there being no significant differences in DEMO scores between semi-quantitative neuropathological score groups (0–3) with a confidence level of 0.95, including for the global ABC severity scores of Thal phase for Aβ (A score F-test: $F_{(3, 51)} = 3.665$, $p = 1.813\text{e-}2$), Braak & Braak for neurofibrillary tangles (NFTs) (B score F-test: $F_{(3, 102)} = 11.528$, $p = 1.432\text{e-}6$), and CERAD neuritic plaque scores (C score F-test: $F_{(3, 103)} = 4.924$, $p = 3.088\text{e-}3$) (Fig. 5c). We further performed post hoc testing using Tukey's

procedure to compare pairwise group means of DEMO scores, observing consistently significant differences between individuals with the highest and lowest burdens of neurodegenerative findings, respectively (Fig. S8). Of note, we also observed an increasing trend of ALZ score with the semi-quantitative neuropathological scores (Fig. 5d).

**Expert-level validation.** Lastly, to provide clinical benchmarking of our modeling approach, both neurologists and neuroradiologists were recruited to perform diagnostic tasks on a subset of NACC cases. The approach and performance of the neurologists and the neuroradiologists indicated variability across different clinical practices (See Supplementary Information: Neurologist and Neuroradiologist Accounts), with a moderate inter-rater agreement as evaluated using pairwise kappa (κ) scoring for all the tasks. Among neurologists specifically, we observed average $\kappa = 0.600$ for the $COG_{NC}$ task (Fig. 6a, Row 1) and average $\kappa = 0.601$ for the $COG_{DE}$ task (Fig. 6a, Row 2). Among neuroradiologists performing the ADD task, we found average $\kappa = 0.292$ (Fig. 6b). In the overall 4-way classification of NC, MCI, AD, and nADD, we observed that the accuracy of our fusion model (mean: 0.558, 95% CI: [0.482,0.634]) reached that of neurologists (mean: 0.565, 95% CI: [0.529,0.601]). Interestingly, a similar level of 4-way accuracy was achieved by a non-imaging CatBoost model (mean: 0.544, 95% CI: [0.517,0.571]), though not on an MRI-only model (mean: 0.412, 95% CI: [0.380,0.444]). However, an MRI-only model did yield a moderate improvement in diagnostic accuracy (mean: 0.692, 95% CI: [0.649,0.735]) over neuroradiologists (mean: 0.566, 95% CI: [0.516,0.616]) in the ADD task (Fig. 6b). Full performance metrics (including accuracy, sensitivity, specificity, F-1 score, and Matthews Correlation Coefficient) may be found in Tables S6 and S7 for respective comparison of machine learning models to neurologists and neuroradiologists in diagnostic simulations. Performance metrics for simple thresholding of various neuropsychologic test scores can be found in Table S8. We also sought to correlate region-specific SHAP values with structural changes observed by the neuroradiologists throughout the brain, with particular attention towards limbic and temporal lobe structures. Statistically significant correlations between regional SHAP averages and clinically graded atrophy severity suggested a connection between CNN features and widely known markers of dementia (Fig. 6c).

## Discussion

In this work, we presented a range of machine learning models that can process multimodal clinical data to accurately perform a differential diagnosis of AD. These frameworks can achieve multiple diagnostic steps in succession, first delineating persons based on overall cognitive status (NC, MCI, and DE) and then separating likely cases of AD from those with nADD. Importantly, our models are capable of functioning with flexible combinations of imaging and non-imaging data, and their performance generalized well across multiple datasets featuring a diverse range of cognitive statuses and dementia subtypes.

Our fusion model demonstrated the highest overall classification accuracy across diagnostic tasks, achieving results on par with neurologists recruited from multiple institutions to complete clinical simulations. Notably, similar levels of performance were observed both in the NACC testing set, and in the OASIS external validation set. Our MRI-only model also surpassed the average diagnostic accuracy of practicing neuroradiologists and maintained a similar level of performance in 6 additional external cohorts (ADNI, AIBL, FHS, NIFD, PPMI, and LBDSU), thereby suggesting that diagnostic capability was not biased to any single data source. It is also worth noting that the DEMO and the ALZ

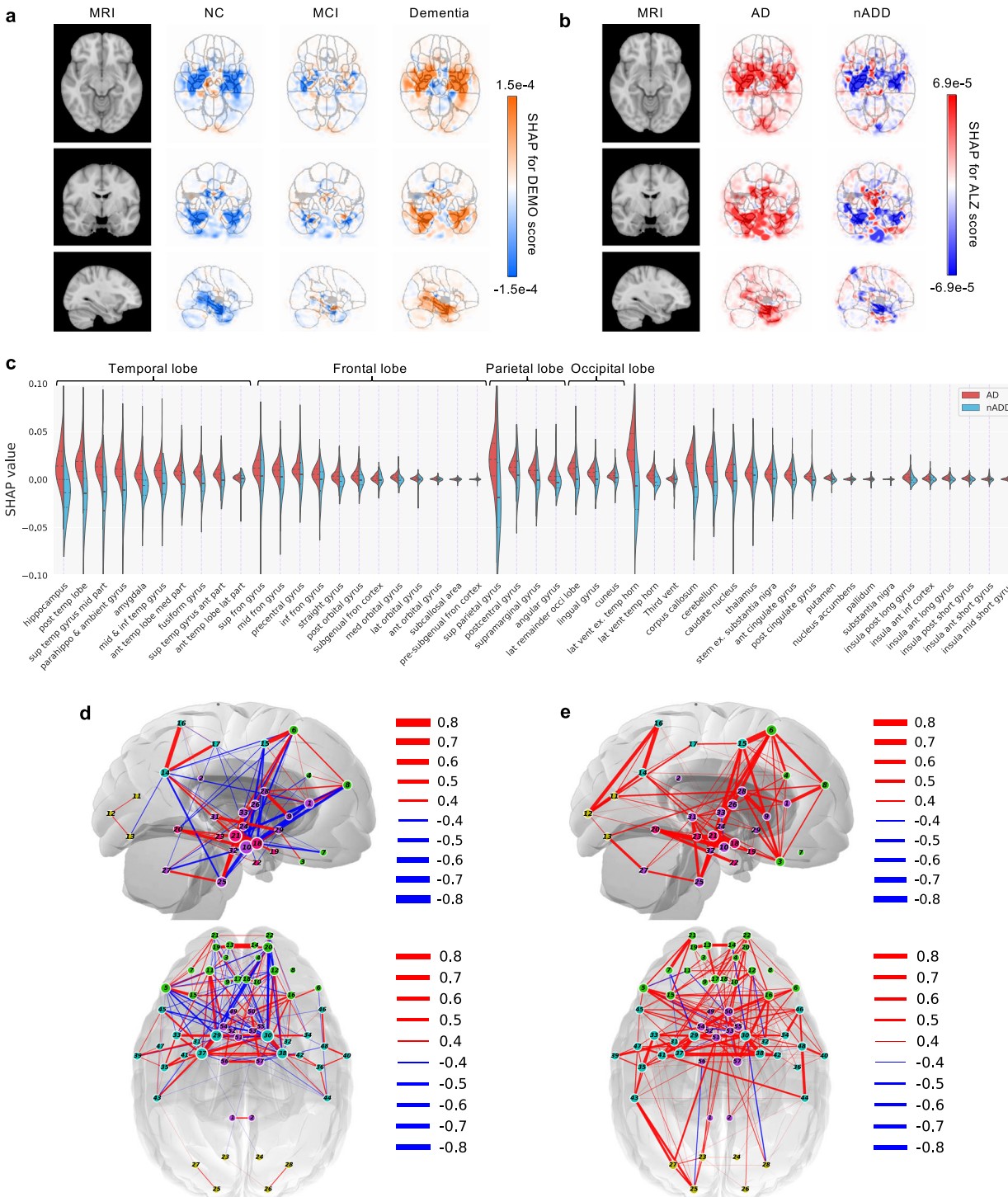

**Fig. 4 Neuroimaging signatures of dementia. a**, **b** SHAP value-based illustration of brain regions that are most associated with the outcomes. The first columns in both **a** and **b** show a template MRI oriented in axial, coronal, and sagittal planes. In **a**, the second, third and fourth columns show SHAP values from the input features of the second convolutional block of the CNN averaged across all NACC test subjects with NC, MCI, and dementia, respectively. In **b**, the second and third columns show SHAP values averaged across all NACC test subjects with AD and nADD, respectively. **c** Brain region-specific SHAP values for both AD and nADD cases obtained from the NACC testing data are shown. The violin plots are organized per lobe and in decreasing order of mean absolute SHAP values. **d**, **e** Network of brain regions implicated in the classification of AD and nADD, respectively. We selected 33 representative brain regions for graph analysis and visualization of sagittal regions, as well as 57 regions for axial analyses. Nodes representing brain regions are overlaid on a two-dimensional brain template and sized according to weighted degree. The color of the segments connecting different nodes indicates the sign of correlation and the thickness of the segments indicates the magnitude of the correlation. It must be noted that not all nodes can be seen either from the sagittal or the axial planes. Source data are provided as a Source Data file.

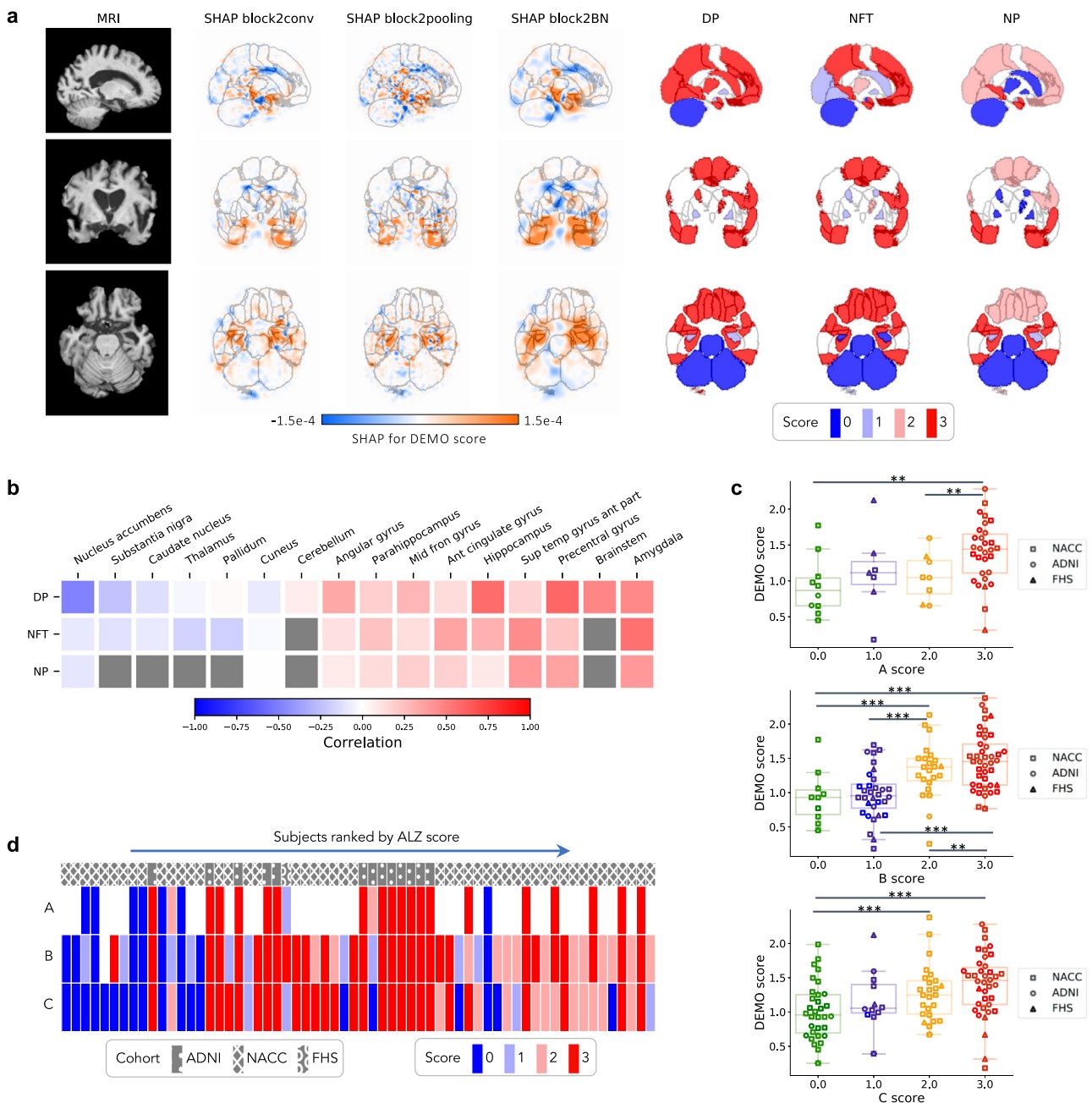

**Fig. 5 Neuropathological validation.** We correlated model findings with regional ABC scores of neuropathologic severity obtained autopsied participants in NACC, ADNI, and FHS cohorts ($n = 110$). **a** An example case from the Alzheimer's Disease Neuroimaging Initiative (ADNI) dataset is displayed in sagittal, axial, and coronal views. The SHAP values derived from the second convolutional block and neuropathologic ABC scores are mapped to brain regions where they were measured at the time of autopsy. Visually, high concordance is observed between anatomically mapped SHAP values regardless of the hidden layer from which they are derived. Concordance is observed between the SHAP values and neurofibrillary tangles (NFT) scores within the temporal lobe. **b** A heatmap is shown demonstrating Spearman correlations between population-averaged SHAP values from the input features of the second convolutional layer and stain-specific ABC scores at various regions of the brain. A strong positive correlation is observed between the SHAP values and neuropathologic changes within several areas well-known to be affected in AD such as the hippocampus/parahippocampus, amygdala and temporal gyrus. **c** Beeswarm plots with overlying box-and-whisker diagrams are shown to denote the distribution of ABC system sub-scores (horizontal axis) versus model-predicted cognitive scores (vertical axis). The displayed data points represent a pooled set of participants from ADNI, NACC, and FHS for whom neuropathology reports were available from autopsy. Each symbol represents a study participant, boxes are centered at the median and extend over the interquartile range (IQR), while bottom and top whiskers represent 1st and 3rd quartiles $-/+$ 1.5 x IQR, respectively. We denote $p < 0.05$ as *; $p < 0.001$ as **, and $p < 0.0001$ as *** based on post-hoc Tukey testing. **d** A heatmap demonstrating the distribution of neuropathology scores versus model predicted AD probabilities. Herein, each column within the map represents a unique individual whose position along the horizontal axis is a descending function of AD risk according to the deep learning model. The overlying hatching pattern represents the dataset (ADNI, NACC, and FHS), from which everyone is drawn. Source data are provided as a Source Data file.

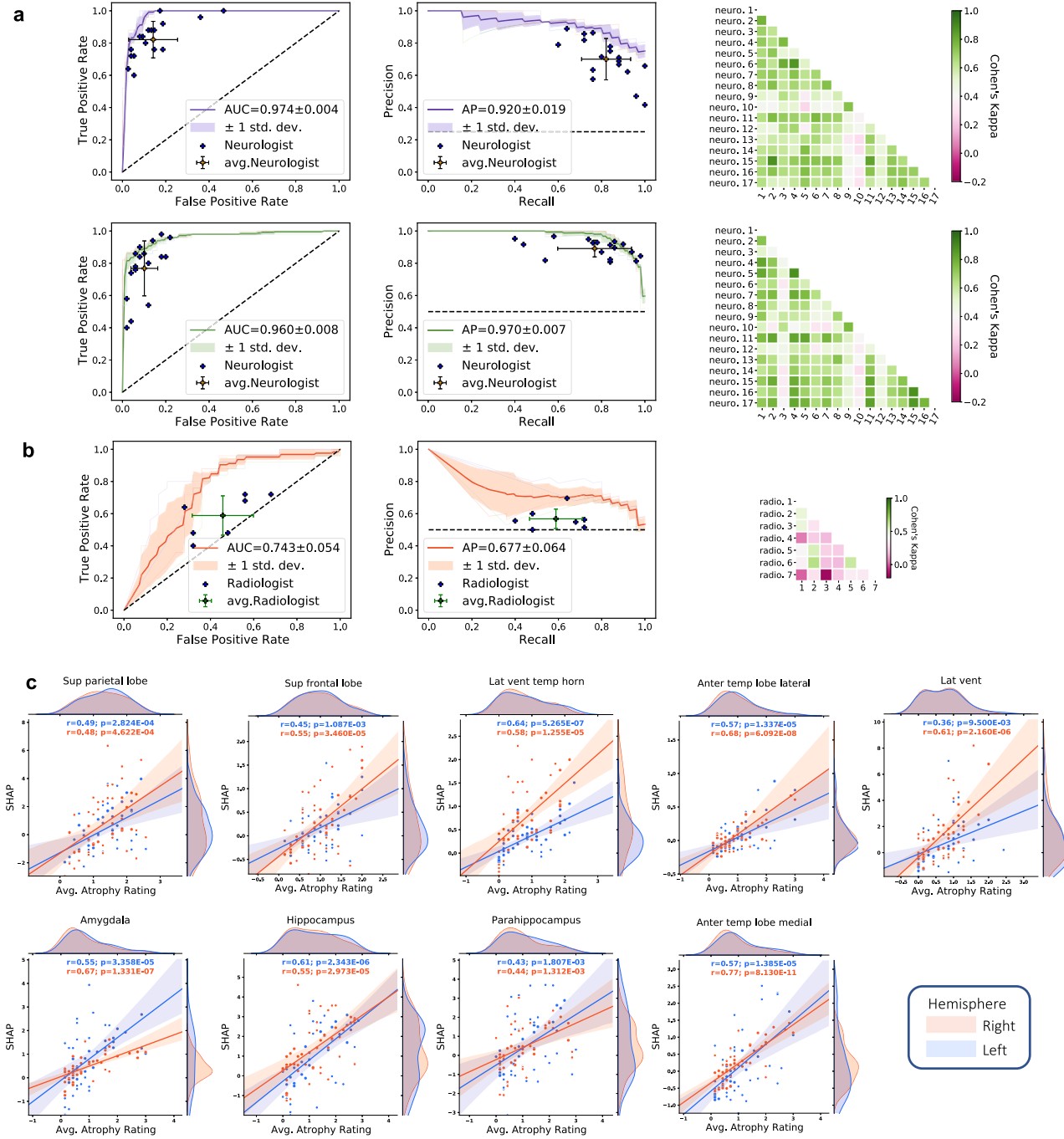

**Fig. 6 Expert-level validation. a** For the COG$_{NC}$ task (Row 1), the diagnostic accuracy of board-certified neurologists ($n = 17$) is compared to the performance of our deep learning model using a random subset of cases from the NACC dataset ($n = 100$). Metrics from individual clinicians are plotted in relation to the ROC and PR curves from the trained model. Individual clinician performance is indicated by the blue plus symbol and averaged clinician performance along with error bars is indicated by the green plus symbol on both the ROC and PR curves. The mean ROC/PR curve and the standard deviation are shown as the bold line and shaded region, respectively. A heatmap of pairwise Cohen's kappa statistic is also displayed to demonstrate inter-rater agreement across the clinician cohort. For the COG$_{DE}$ task (Row 2), ROC, PR, and interrater agreement graphics are illustrated with comparison to board-certified neurologists in identical fashion. For these tasks, all neurologists were granted access to multimodal patient data, including MRIs, demographics, medical history, functional assessments, and neuropsychological testing. The same data was used as input to train the deep learning model. **b** For validation of our ADD task, a random subset ($n = 50$) of cases with dementia from the NACC cohort was provided to the team of neuroradiologists ($n = 7$), who classified AD versus those with dementia due to other etiologies (nADD). As above, the diagnostic accuracy of the physician cohort is compared to model performance using ROC and PR curves. Graphical conventions for visualizing model and clinician performance are as described above in **a** and, once more, pairwise Cohen's kappa values are shown to demonstrate inter-rater agreement. **c** SHAP values from the second convolutional layer averaged from selected brain regions are shown plotted against atrophy scores assigned by neuroradiologists. Orange and blue points (and along with regression lines and 95% confidence intervals) represent left and right hemispheres, respectively. Spearman correlation coefficients and corresponding two-tailed $p$ values are also shown and demonstrate a statistically significant proportionality between SHAP scores, and the severity of regional atrophy assigned by clinicians. Source data are provided as a Source Data file.

scores bore strong analytic importance like that of traditional information used for dementia diagnosis. For instance, in the ADD task, the ALZ score was shown by SHAP analysis to have a greater impact in accurately predicting disease status than key demographic and neuropsychological test variables used in standard clinical practice such as age, sex, and MMSE score. These CNN-derived scores maintained equal levels of importance when used in other machine learning classifiers, suggesting wide utility for digital health workflows.

Furthermore, post-hoc analyses demonstrated that the performance of our machine learning models was grounded in well-established patterns of dementia-related neurodegeneration. Network analyses evinced differing regional distributions of SHAP values between AD and nADD populations, which were most pronounced in areas such as the hippocampus, amygdala, and temporal lobes. The SHAP values in these regions also exhibited a strong correlation with atrophy ratings from neuroradiologists. Although recent work has shown that explainable machine learning methods may identify spurious correlations in imaging data[24], we feel that our ability to link regional SHAP distributions to both anatomic atrophy and also semi-quantitative scores of Aβ amyloid, neurofibrillary tangles, and neuritic plaques links our modeling results to a gold standard of postmortem diagnosis. More generally, our approach demonstrates a means by which to assimilate deep learning methodologies with validated clinical evidence in health care.

Our work builds on prior efforts to construct automated systems for the diagnosis of dementia. Previously, we developed and externally validated an interpretable deep learning approach to classify AD using multimodal inputs of MRI and clinical variables[17]. Although this approach provided a novel framework, it relied on a contrived scenario of discriminating individuals into binary outcomes, which simplified the complexity of a real-world setting. Our current work extends this framework by mimicking a memory clinic setting and accounting for cases along the entire cognitive spectrum. Though numerous groups have taken on the challenge of nADD diagnosis using deep learning[18–20,25,26], even these tasks were constructed as simple binary classifications between disease subtypes. Given that the practice of medicine rarely reduces to a choice between two pathologies, integrated models with the capability to replicate the differential diagnosis process of experts more fully are needed before deep learning models can be touted as assistive tools for clinical-decision support. Our results demonstrate a strategy for expanding the scope of diagnostic tasks using deep learning, while also ensuring that the predictions of automated systems remain grounded in established medical knowledge.

Interestingly, it should be noted that the performance of a non-imaging model alone approached that of the fusion model. However, the inclusion of neuroimaging data was critical to enable verification of our modeling results by clinical criteria (e.g., cross-correlation with post-mortem neuropathology reports). Such confirmatory data sources cannot be readily assimilated to non-imaging models, thus limiting the ability to independently ground their performance in non-computational standards. Therefore, rather than viewing the modest contribution of neuroimaging to diagnostic accuracy as a drawback, we argue that our results suggest a path towards balancing demands for transparency with the need to build models using routinely collected clinical data. Models such as ours may be validated in high-resource areas where the availability of advanced neuroimaging aids interpretability. As physicians may have difficulty entrusting medical decision-making to black box model in artificial intelligence[27], grounding our machine learning results in the established neuroscience of dementia may help to facilitate clinical uptake. Nevertheless, we note that our non-imaging model

may be best suited for deployment among general practitioners (GPs) and in low-resource settings.

Functionally, we also contend that the flexibility of inputs afforded by our approach is a necessary precursor to clinical adoption at multiple stages of dementia. Given that sub-group analyses suggested significant 4-way diagnostic capacity on multiple combinations of training data (i.e., demographics, clinical variables, and neuropsychological tests), our overall framework is likely adaptable to many variations of clinical practice without requiring providers to significantly alter their typical workflows. For example, GPs frequently perform cognitive screening with or without directly ordering MRI tests[28–30], whereas memory specialists typically expand testing batteries to include imaging and advanced neuropsychological testing. This ability to integrate along the clinical care continuum, from primary to tertiary care allows our deep learning solution to address a two-tiered problem within integrated dementia care by providing a tool for both screening and downstream diagnosis.

Our study has several limitations. To begin, in cases of mixed dementia, the present models default to a diagnosis of AD whenever this condition is present, thus attributing a single diagnosis to participants with multiple comorbidities. Given the considerable prevalence of mixed dementias[31], future work may include the possibility of a multi-label classification that may allow for the identification of co-occurring dementing conditions (e.g., LBD and AD, VD and AD) within the same individual. Our cohorts also did not contain any confirmed cases of atypical AD, which is estimated to affect approximately 6% of elderly-onset cases and one-third of patients with early-onset disease[32]. We must also note that MCI is a broad category by itself that includes persons who may or may not progress to dementia. When relevant data becomes available across many cohorts, future investigations could include MCI subjects who are amnestic and non-amnestic, to understand distinct signatures of those who have prodromal AD. We also acknowledge that our study data is predominantly obtained from epidemiologic studies which primarily focus on AD and that variables that optimize the identification of this illness may in fact detract from the accurate diagnosis of certain nADDs. For instance, we noted that the performance of our fusion models was slightly lower than that of the MRI-only model for distinguishing AD from non-parkinsonian dementias such as FTD and VD. We speculate that certain forms of neuropsychological testing such as the MMSE, which have well-known limitations in specificity[33], may bias predictions towards more common forms of dementia such as AD. Although we validated the various models using data from a population-based cohort (i.e., FHS), it is possible that multimodal analysis frameworks have the potential to decrease diagnostic accuracy for less common dementias. Future modeling efforts may optimize for the identification of these diseases by including additional clinical data tailored to their diagnosis: for instance, the inclusion of motor examination to assess for parkinsonism, FLAIR images for vascular injury, or cognitive fluctuations and sleep behavior abnormalities for LBD. Lastly, although we have compared our model to the performance of individual neurologists and neuroradiologists, future studies may consider comparison to consensus reviews by teams of collaborating clinicians.

In conclusion, our interpretable, multimodal deep learning framework was able to obtain high accuracy signatures of dementia status from routinely collected clinical data, which was validated against data from independent cohorts, neuropathological findings, and expert-driven assessment. Moreover, our approach provides a solution that may be utilized across different practice types, from GPs to specialized memory clinics at tertiary

care centers. We envision performing a prospective observational study in memory clinics to confirm our model's ability to assess dementia status at the same level as the expert clinician involved in dementia care. If confirmed in such a head-to-head comparison, our approach has the potential to expand the scope of machine learning for AD detection and management, and ultimately serve as an assistive screening tool for healthcare practitioners.

## Methods

**Study population.** This study was exempted from local institutional review board approval, as all neuroimaging and clinical data were obtained in deidentified format upon request from external study centers, who ensured compliance with ethical guidelines and informed consent for all participants. No compensation was provided to participants.

We collected demographics, medical history, neuropsychological tests, and functional assessments as well as magnetic resonance imaging (MRI) scans from 8 cohorts (Table 1), totaling 8916 participants after assessing for inclusion criteria. There were 4550 participants with normal cognition (NC), 2412 participants with mild cognitive impairment (MCI), 1606 participants with Alzheimer's disease dementia (AD) and 348 participants with dementia due to other causes. The eight cohorts include the Alzheimer's Disease Neuroimaging Initiative (ADNI) dataset ($n = 1821$)[34–36], the National Alzheimer's Coordinating Center (NACC) dataset ($n = 4822$)[21,22], the frontotemporal lobar degeneration neuroimaging initiative (NIFD) dataset ($n = 253$)[37], the Parkinson's Progression Marker Initiative (PPMI) dataset ($n = 198$)[38], the Australian Imaging, Biomarker and Lifestyle Flagship Study of Ageing (AIBL) dataset ($n = 661$)[39–41], the Open Access Series of Imaging Studies-3 (OASIS) dataset ($n = 666$)[42], the Framingham Heart Study (FHS) dataset ($n = 313$)[43,44], and in-house data maintained by the Lewy Body Dementia Center for Excellence at Stanford University (LBDSU) ($n = 182$)[45].

We labeled the participants according to the clinical diagnosis (See Supplementary Information: Data to clinicians and diagnostic criterion). Subjects were labeled according to the clinical diagnoses provided by each study cohort. We kept MCI diagnoses without further consideration of underlying etiology to simulate a realistic spectrum of MCI presentations. For any subjects with documented dementia and primary diagnosis of Alzheimer's disease dementia, an AD label was assigned regardless of the presence of additional dementing comorbidities. Subjects with dementia but without confirmed AD diagnosis were labeled as nADD. Notably, we elected to conglomerate all nADD subtypes into a singular label given that subdividing model training across an arbitrary number of prediction tasks ran the risk of diluting overall diagnostic accuracy. The ensemble of these 8 cohorts provided us a considerable number of participants with various forms of dementias as their primary diagnosis, including Alzheimer's disease dementia (AD, $n = 1606$), Lewy body dementia (LBD, $n = 63$), frontotemporal dementia (FTD, $n = 193$), vascular dementia (VD, $n = 21$), and other causes of dementia ($n = 237$). We provided a full survey of nADD dementias by cohort in the Supplementary Information (Table S9).

**Data inclusion criterion.** Subjects from each cohort were eligible for study inclusion if they had at least one T1-weighted volumetric MRI scan within 6 months of an officially documented diagnosis. We additionally excluded all MRI scans with fewer than 60 slices. For subjects with multiple MRIs and diagnosis records within a 6-month period, we selected the closest pairing of neuroimaging and diagnostic label. Therefore, only one MRI per subject was used. For the NACC and the OASIS cohorts, we further queried all available variables relating to demographics, past medical history, neuropsychological testing, and functional assessments. We did not use the availability of non-imaging features to exclude individuals in these cohorts and used K-nearest neighbor imputation to fill any missing data fields. Our overall data inclusion workflow may be found in Fig. S9, where we reported the total number of subjects from each cohort before and after application of the inclusion criterion. See Information Availability by Cohort in the Supplementary Information.

**MRI harmonization and preprocessing.** To harmonize neuroimaging data between cohorts, we developed a pipeline of preprocessing operations (Fig. S10) that was applied in identical fashion to all MRIs used in our study. This pipeline broadly consisted of two phases of registration to a standard MNI-152 template. We describe Phase 1 as follows:

- Scan axes were reconfigured to match the standard orientation of MNI-152 space.
- Using an automated thresholding technique, a 3D volume-of-interest within the original MRI was identified containing only areas with brain tissue.
- The volume-of-interest was skull-stripped to isolate brain pixels.
- A preliminary linear registration of the skull-stripped brain to a standard MNI-152 template was performed. This step approximated a linear transformation matrix from the original MRI space to the MNI-152 space.

Phase 2 was designed to fine-tune the quality of linear registration and parcellate the brain into discrete regions. These goals were accomplished by the following steps:

- The transformation matrix computed from linear registration in Phase 1 was applied to the original MRI scan.
- Skull stripping was once again performed after applying the linear registration computed from the initial volume of interest to isolate brain tissue from the full registered MRI scan.
- Linear registration was applied again to alleviate any misalignments to MNI-152 space.
- Bias field correction was applied to account for magnetic field inhomogeneities.
- The brain was parcellated by applying a nonlinear warp of the Hammersmith Adult brain atlas to the post-processed MRI.

All steps of our MRI-processing pipeline were conducted using FMRIB Software Library v6.0 (FSL) (Analysis Group, Oxford University). The overall preprocessing workflow was inspired by the harmonization protocols of the UK Biobank (https://git.fmrib.ox.ac.uk/falmagro/UK_biobank_pipeline_v_1). We manually inspected the outcome of the MRI pipeline on each scan to filter out cases with poor quality or significant processing artifacts.

**Evaluation of MRI harmonization.** We further assessed our image harmonization pipeline by clustering the data using the t-distributed stochastic neighbor embedding (tSNE) algorithm[46]. We performed this procedure in order to ensure that (i) input data for all models was free of site-, scanner-, and cohort-specific biases and (ii) such biases could not be learned by a predictive model. To accomplish (i), we performed tSNE using pixel values from post-processed, 8x-downsampled MRI scans. For (ii), we performed tSNE using hidden-layer activations derived from the penultimate layer of a convolutional neural network (CNN) developed for our prediction tasks (see "Model Development" below). For the NACC dataset, we assessed clustering of downsampled MRIs and hidden layer activations based on specific Alzheimer's Disease Research Centers (ADRCs) and scanner manufacturers (i.e., Siemens, Philips, and General Electric). We also repeated tSNE analysis based on specific cohorts (i.e., NACC, ADNI, FHS, etc.) using all available MRIs across our datasets. We also calculated mutual information scores (MIS) between ADRC ID, scanner brand, and diagnostic labels (NC, MCI, AD, and nADD) in the NACC dataset. This metric calculates the degree of similarity between two sets of labels on a common set of data. As with the tSNE analysis, the MIS calculation helped us to exclude the presence of confounding site- and scanner-specific biases on MRI data.

**Harmonization of non-imaging data.** To harmonize the non-imaging variables across datasets, we first surveyed the available clinical data in all eight cohorts (See Information Availability by Cohort and Non-Imaging Features Used in Model Development in the Supplementary Information). We specifically examined information related to demographics, past medical history, neuropsychological test results, and functional assessments. Across a range of clinical features, we found the greatest availability of information in the NACC and the OASIS datasets. Additionally, given that the NACC and the OASIS cohorts follow Uniform Data Set (UDS) guidelines, we were able to make use of validated conversion scales between UDS versions 2.0 and 3.0 to align all cognitive measurements onto a common scale. We supply a full listing of clinical variables along with missing information rates per cohort in Fig. S11.

**Overview of the prediction framework.** We developed predictive models to meet two main objectives. The first, which we designated the COG task, was to predict the overall degree of cognitive impairment (either NC, MCI, or dementia [DE]) in each participant based on neuroimaging. To meet this goal, we predicted a continuous 0–2 score (NC: 0, MCI: 1, DE: 2), which we denote as the DEmentia MOdel (DEMO) score. Of note, the COG task may also be regarded as consisting of three separate subtasks: (i) separation of NC from MCI and DE (COG$_{NC}$ task), (ii) separation of MCI from NC and DE (COG$_{MCI}$ task), and (iii) separation of DE from NC and MCI (COG$_{DE}$ task). The second objective, which we designated the ADD task, was to predict whether a participant held a diagnosis of AD or nADD given that they were already predicted as DE in the COG task. For ease of reference, we denoted the probability of a person holding an AD diagnosis as the ALZheimer (ALZ) score. Following the sequential completion of the COG and ADD tasks, we were able to successfully separate AD participants from NC, MCI, and nADD subjects.

**MRI-only model.** We used post-processed volumetric MRIs as inputs and trained a CNN model. To transfer information between the COG and ADD tasks, we trained a common set of convolutional blocks to act as general-purpose feature extractors. The DEMO and the ALZ scores were then calculated separately by appending respective fully connected layers to the shared convolutional backbone. We conducted the COG task as a regression problem using mean square error loss between the DEMO score and available cognitive labels. We performed the ADD task as a classification problem using binary cross entropy loss between the reference AD

label and the ALZ score. The MRI-only model was trained using the NACC dataset and validated on all the other cohorts. To facilitate presentation of results, we pooled data from all the external cohorts (ADNI, AIBL, FHS, LBDSU, NIFD, OASIS, and PPMI), and computed all the model performance metrics.

**Non-imaging model**. In addition to an MRI-only model, we developed a range of traditional machine learning classifiers using all available non-imaging variables shared between the NACC and the OASIS datasets. We first compiled vectors of demographics, past medical history, neuropsychological test results, and functional assessments. We scaled continuous variables by their mean and standard deviations and one-hot encoded categorical variables. These non-imaging data vectors were then passed as input to CatBoost, XGBoost, random forest, decision tree, multi-layer perceptron, support vector machine and K-nearest neighbor algorithms. Like the MRI-only model, each non-imaging model was sequentially trained to complete the COG and the ADD tasks by calculating the DEMO and the ALZ scores, respectively. We ultimately found that a CatBoost model yielded the best overall performance per area-under-receiver-operating-characteristic curve (AUC) and area-under-precision-recall curve (AP) metrics. We, therefore, selected this algorithm as the basis for follow-up analyses.

To mimic a clinical neurology setting, we developed a non-imaging model using data that is routinely collected for dementia diagnosis. A full listing of the variables used as input may be found in our Supplementary Information. While some features such as genetic status (APOE ε4 allele)[47], or cerebrospinal fluid measures[10] have great predictive value, we have purposefully not included them for model development because they are not part of the standard clinical work-up of dementia.

To infer the extent to which completeness of non-imaging datasets influenced model performance, we conducted multiple experiments using different combinations of clinical data variables. The following combinations were input to the CatBoost algorithm for comparison: (1) demographic characteristics alone, (2) demographic characteristics and neuropsychological tests, (3) demographic characteristics and functional assessments, (4) demographic characteristics and past medical history, (5) demographic characteristics, neuropsychological tests and functional assessments, (6) demographic characteristics, neuropsychological tests and past medical history, and (7) demographic characteristics, neuropsychological tests, past medical history, and functional assessments.

**Fusion model**. To best leverage every aspect of the available data, we combined both MRI and non-imaging features into a common "fusion" model for the COG and the ADD tasks. The combination of data sources was accomplished by concatenating the DEMO and the ALZ scores derived from the MRI-only model to lists of clinical variables. The resultant vectors were then used as input to traditional machine learning classifiers as described above. Based on the AUC and the AP metrics, we ultimately found that a CNN linked with CatBoost model yielded the highest performance in discriminating different cognitive categories; the combination of CNN and CatBoost models was thus used as the final fusion model for all further experiments. Similarly, to our procedure with the non-imaging model, we studied how MRI features interacted with different subsets of demographic, past medical history, neuropsychological, and functional assessment variables. As with our non-imaging model, development and validation of fusion models was limited to NACC and OASIS only given limited availability of non-imaging data in other cohorts.

**Training strategy and data splitting**. We trained all models on the NACC dataset using cross validation. NACC was randomly divided into 5 folds of equal size with constant ratios of NC, MCI, AD, and nADD cases. We trained the model on 3 of the 5 folds and used the remaining two folds for validation and testing, respectively. Each tuned model was also tested on the full set of available cases from external datasets. Performance metrics for all models were reported as a mean across five folds of cross validation along with standard deviations and 95% confident intervals. A graphical summary of our cross-validation strategy may be found within Fig. S12. Prior to training, we also set aside two specialized cohorts within NACC for neuropathologic validation and head-to-head comparison with clinicians. In the former case, we identified 74 subjects from whom post-mortem neuropathological data was available within 2 years of an MRI scan. In the latter, we randomly selected 100 age- and sex-matched groups of patients (25 per diagnostic category) to provide simulated cases to expert clinicians.

**SHAP analysis**. SHAP is a unified framework for interpreting machine learning models which estimates the contribution of each feature by averaging over all possible marginal contributions to a prediction task[23]. Though initially developed for game theory applications[48], this approach may be used in deep learning-based computer vision by considering each image voxel or a network node as a unique feature. By assigning SHAP values to specific voxels or by mapping internal network nodes back to the native imaging space, heatmaps may be constructed over input MRIs.

Though a variety of methods exist for estimating SHAP values, we implemented a modified version of the DeepLIFT algorithm[49], which computes SHAP by estimating differences in model activations during backpropagation relative to a

standard reference. We established this reference by integrating over a "background" of training MRIs to estimate a dataset-wide expected value. For each testing example, we then calculated SHAP values for the overall CNN model as well as for specific internal layers. Two sets of SHAP values were estimated for the COG and ADD tasks, respectively. SHAP values calculated over the full model were directly mapped back to native MRI pixels whereas those derived for internal layers were translated to the native imaging space via nearest neighbor interpolation.

**Network analysis**. We sought to perform a region-by-region graph analysis of SHAP values to determine whether consistent differences in ADD and nADD populations could be demonstrated. To visualize the relationship of SHAP scores across various brain regions, we created graphical representations of inter-region SHAP correlations within the brain. We derived region-specific scores by averaging voxel-wise SHAP values according to their location within the registered MRI. Subsequently, we constructed acyclic graphs in which nodes were defined as specific brain regions and edges as inter-regional correlations measured by Spearman's rank correlation and Pearson correlation coefficient, separately. To facilitate visualization and convey structural information, we manually aligned the nodes to a radiographic projection of the brain.

Once correlation values were calculated between every pair of nodes, we filtered out the edges with $p$ value larger than 0.05 and ranked the remaining edges according to the absolute correlation value. We used only the top N edges ($N = 100$ for sagittal view, $N = 200$ for axial view) for the graph. We used color to indicate the sign of correlation and thickness to represent the magnitude of correlation. We used the following formula to derive the thickness:

$$thickness(corr.) = const. \times (abs(corr.) - threshold) \qquad (1)$$

where the threshold is defined as the minimum of the absolute value of all selected edges' correlation value. The radius of nodes represents the weighted degree of the node which is defined as the sum of the edge weights for edges incident to that node. More specifically, we calculated the radius using the following equation:

$$radius(node_i) = 20 + 3*\left(\sum_j correlation(node_i, node_j)\right) \qquad (2)$$

In the above equation, we used 20 as a bias term to ensure that every node has at-least a minimal size to be visible on the graph. Note as well that the digit inside each node represents the index of the region name. Derivation of axial and sagittal nodes from the Hammersmith atlas is elaborated in Table S5.

**Neuropathologic validation**. Neuropathologic evaluations are considered to be the gold standard for confirming the presence and severity of neurodegenerative diseases[50]. We validated our model's ability to identify regions of high risk of dementia by comparing the spatial distribution of the model-derived scores with post-mortem neuropathological data from NACC, FHS, and ADNI study cohorts, derived from the National Institute on Aging Alzheimer's Association guidelines for the neuropathologic assessment of AD[51]. Hundred and ten participants from NACC ($n = 74$), ADNI ($n = 25$) and FHS ($n = 11$) who met the study inclusion criteria, had MRI scans taken within 2 years of death and with neuropathologic data were included in the neuropathologic validation. The data was harmonized in the format of the Neuropathology Data Form Version 10 of the NACC established by the National Institute on Aging. The neuropathological lesions of AD (i.e., amyloid β deposits (Aβ), neurofibrillary tangles (NFTs), and neuritic plaques (NPs)) were assessed in the entorhinal, hippocampal, frontal, temporal, parietal, and occipital cortices. The regions were based on those proposed for standardized neuropathological assessment of AD and the severity of the various pathologies were classified into four semi-quantitative score categories ($0 =$ None, $1 =$ Mild, $2 =$ Moderate, $3 =$ Severe)[52]. Based on the NIA-AA protocol, the severity of neuropathologic changes were evaluated using a global "ABC" score which incorporates histopathologic assessments of amyloid β deposits by the method of Thal phases:[53] (A), staging of neurofibrillary tangles (B) silver-based histochemistry[54], or phospho-tau immunohistochemistry[55], and scoring of neuritic plaques (C). Spearman's rank correlation was used to correlate the DEMO score predictions with the A, B, C scores, and ANOVA and Tukey's tests were used to assess the differences in the mean DEMO scores across the different levels of the scoring categories. Lastly, a subset of the participants from ADNI ($n = 25$) and FHS ($n = 11$) had regional semi-quantitative Aβ, NFT, and NP scores, which was also used to validate the model predictions.

**Expert-level validation**. We sought to test our model's predictions against the diagnostic acumen of clinicians who are involved in care of patients with dementia. We recruited an international cohort of practicing neurologists and neuroradiologists to participate in simulated diagnostic tasks using a subset of NACC cases (see "Training strategy and data splitting" above). Neurologists were provided with 100 cases that included imaging data (T1-weighted brain MRI scans) and non-imaging data (demographics, medical history, neuropsychological tests, and functional assessments) and asked to provide diagnostic impressions of NC, MCI, AD, and nADD. Notably, the model was not directly compared to neurologists for the ADD task given that our framework only performs this prediction on patients internally identified as demented. Due to this computational pre-selection, it was not feasible to consistently compare a common cohort of persons with neurologists who also

must perform a differential diagnosis of NC, MCI, AD, and nADD. Neuroradiologists were provided with imaging data (T1-weighted brain MRI scans), age, and gender from 50 known DE cases and then asked to provide diagnostic impressions of AD or nADD. For each case, the neuroradiologists also answered a questionnaire to grade the extent of atrophy in each sub-region of the brain on a scale of 0 to 4, where higher values indicate greater atrophy. A case sample and example questionnaires provided to neurologists and neuroradiologists, respectively, may be found within the Supplementary Information (Data to clinicians and diagnostic criterion). For both groups of clinicians, we also calculated inter-annotator agreement using Cohen's kappa (κ). Additionally, to compare our machine learning models to neuropsychological assessments, we performed the COG_NC, COG_DE, and ADD tasks using all possible whole number cutoffs of neuropsychiatric test scores available in the NACC dataset. Following this approach, we performed simple thresholding for binary classifications.

**Performance metrics**. We presented the performance by computing the mean and the standard deviation over the model runs. We generated receiver operating characteristic (ROC) and precision-recall (PR) curves based on model predictions on the NACC test data as well as on the other datasets. For each ROC and PR curve, we also computed the area under curve (AUC & AP) values. Additionally, we computed sensitivity, specificity, F1-score and Matthews correlation coefficient on each set of model predictions. The F1-score considers both precision and recall of a test whereas the MCC is a balanced measure of quality for dataset classes of different sizes of a binary classifier. We also calculated inter-annotator agreement using Cohen's kappa (κ), as the ratio of the number of times two experts agreed on a diagnosis. We computed average pairwise κ for each sub-group task that provided an overall measure of agreement between the neurologists and the neuroradiologists, respectively.

**Statistical analysis**. We used one-way ANOVA test and the χ2 test for continuous and categorical variables, respectively to assess the overall levels of differences in the population characteristics between NC, MCI, AD, and nADD groups across the study cohorts. To validate our CNN model, we evaluated whether the presence and severity of the semi-quantitative neuropathology scores across the neuropathological lesions of AD (i.e., amyloid β deposits (Aβ), neurofibrillary tangles (NFTs), and neuritic plaques (NPs)) reflected the DEMO score predicted by the CNN model. We stratified the lesions based on A, B, and C scores and used Spearman's rank correlation to assess their relationship with the DEMO scores. Next using one-way ANOVA analysis, we evaluated the differences in the mean DEMO scores across the different levels of the scoring categories for the A, B, and C scores. We used the Tukey-Kramer test to identify the pairwise statistically significant differences in the mean DEMO score between the levels of scoring categories (0–3). Similarly, to analyze the correspondence between SHAP values and a known marker of neurodegenerative disease, we correlated SHAPs with the radiologist impressions of atrophy. Utilizing the segmentation maps derived from each participant, we calculated regional SHAP averages on each of the 50 test cases given to neuroradiologists with the 0–4 regional atrophy scales assigned by the clinicians. We calculated Pearson's correlation coefficients with two-tailed p values that indicates the probability that an uncorrelated system producing Pearson's correlation coefficient as extreme as the observed value in the neuroanatomic regions known to be implicated in AD pathology. All statistical analyses were conducted at a significance level of 0.05. Confidence intervals for model performance were calculated by assuming a normal distribution of AUC and AP values across cross-validation experiments using t-student distribution with 4 degrees of freedom.

**Computational hardware and software**. We processed all MRIs and non-imaging data on a computing workstation with Intel i9 14-core 3.3 GHz processor, and 4 NVIDIA RTX 2080Ti GPUs. Python (version 3.7.7) was used for software development. Each deep learning model was developed using PyTorch (version 1.5.1), and plots were generated using the Python library matplotlib (version 3.1.1) and numpy (version 1.18.1) was used for vectorized numerical computation. Other Python libraries used to support data analysis include pandas (version 1.0.3), scipy (version 1.3.1), tensorflow (version 1.14.0), tensorboardX (version 1.9), torchvision (version 0.6) and scikit-learn (version 0.22.1). Using a single 2080Ti GPU, the average run time for training the deep learning model was 10 h, and inference task took less than a minute. All clinicians reviewed MRIs using 3D Slicer (version 4.10.2) (https://www.slicer.org/) and recorded impressions in REDCap (version 11.1.3). Additionally, statistics for neuropathology analysis were completed using SAS (version 9.4).

**Reporting summary**. Further information on research design is available in the Nature Research Reporting Summary linked to this article.

## Data availability
Data from ADNI, AIBL, NACC, NIFD, OASIS and PPMI can be downloaded from publicly available resources. Data from FHS and LBDSU are available upon request and will be subjected to institutional approval. Source data for figures are provided with this paper. Source data are provided with this paper.

## Code availability
Python scripts are made available on GitHub (https://github.com/vkola-lab/ncomms2022).

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

## Acknowledgements

This project was supported by grants from the Karen Toffler Charitable Trust (VBK), the Michael J. Fox Foundation (KLP), the Lewy Body Dementia Association (KLP), the Alzheimer's Drug Discovery Foundation (RA), the American Heart Association (20SFRN35460031, VBK), and the National Institutes of Health (R01-HL159620 [VBK], R21-CA253498 [VBK], RF1-AG062109 [RA], RF1-AG072654 [RA], U19-AG065156 [KLP], P30-AG066515 [KLP], R01-NS115114 [KLP], K23-NS075097 [KLP], U19-AG068753 [RA] and P30-AG013846 [RA, VBK]). We acknowledge the efforts of several investigators from the ADNI, AIBL, FHS, LBDSU, NACC, NIFD, OASIS, and PPMI studies for providing access to data.

## Author contributions

S.Q. and M.I.M. contributed equally to this work. S.Q. and M.I.M. performed literature search. S.Q. designed and developed the deep learning framework, performed image processing, and constructed the models. S.Q., P.S.J., and J.C.L., performed data collection, data harmonization and statistical analysis. C.X., Y.N., Y.W., I.D.A., and P.H.H. assisted with data collection and performed sub-group analyses. S.Q., M.I.M., P.S.J., J.C.L., and C.X. generated the figures and tables. P.S.J. and S.Q. performed the analysis on the neuropathology data. J.A.C., B.C.D., H.H., M.C.K., S.K., P.H.L., A.Z.M., D.L.M., S.O., A.B.P., M-H.S-H., E.A.S., A.R.S., L.C.S., J.E.S., M.J.S., A.S., C.E.T., O.T., H.Y., J.Y., Y.Z., S.Z. and K.L.P. are the practicing clinicians who reviewed the cases. M.L.A., J.M., T.D.S, K.L.P., and R.A. provided the clinical relevance context. K.L.P. and R.A. provided access to data. M.I.M. and V.B.K. wrote the manuscript. All authors reviewed and approved the manuscript. V.B.K. conceived, designed, and directed the entire study.

## Competing interests

V.B.K. reports honoraria from invited scientific presentations to industry not exceeding $5000/year. He also serves as a consultant to Davos Alzheimer's Collaborative. R.A. is a scientific advisor to Signant Health and consultant to Biogen. K.L.P. reports honoraria from invited scientific presentations to universities and professional societies not exceeding $5,000/year and has received consulting fees from Curasen. The remaining authors declare no competing interests.

## Additional information

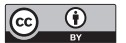

[1]Department of Medicine, Boston University School of Medicine, Boston, MA, USA. [2]Department of Physics, College of Arts & Sciences, Boston University, Boston, MA, USA. [3]Department of Anatomy and Neurobiology, Boston University School of Medicine, Boston, MA, USA. [4]Department of General Dentistry, Boston University School of Dental Medicine, Boston, MA, USA. [5]The Framingham Heart Study, Boston University School of Medicine, Boston, MA, USA. [6]School of Public Health and Tropical Medicine, Tulane University, New Orleans, LA, USA. [7]Department of Radiology, College of Medicine, University of Nebraska Medical Center, Omaha, NE, USA. [8]Department of Neurology, Boston University School of Medicine, Boston, MA, USA. [9]Department of Neurology, Peking Union Medical College Hospital, Peking Union Medical College, Chinese Academy of Medical Sciences, Beijing, China. [10]Department of Neurological Sciences, College of Medicine, University of Nebraska Medical Center, Omaha, NE, USA. [11]Department Neurology, Emory University School of Medicine, Atlanta, GA, USA. [12]Department Ophthalmology, Emory University School of Medicine, Atlanta, GA, USA. [13]Department of Radiology, Lahey Hospital & Medical Center, Burlington, MA, USA. [14]Department of Radiology, Boston University School of Medicine, Boston, MA, USA. [15]Department of Radiology, Peking Union Medical College Hospital, Peking Union Medical College, Chinese Academy of Medical Sciences, Beijing, China. [16]Boston University Alzheimer's Disease Research Center, Boston, MA, USA. [17]Department of Pathology and Laboratory Medicine, Boston University School of Medicine, Boston, MA, USA. [18]Boston VA Healthcare System, Boston, MA, USA. [19]Bedford VA Healthcare System, Bedford, MA, USA. [20]Department of Neurology, Stanford University, Palo Alto, CA, USA. [21]Department of Epidemiology, Boston University School of Public Health, Boston, MA, USA. [22]Department of Computer Science, Boston University, Boston, MA, USA. [23]Faculty of Computing & Data Sciences, Boston University, Boston, MA, USA. [24]These authors contributed equally: Shangran Qiu, Matthew I. Miller. ✉email: vkola@bu.edu

