## [Peer Review File · Nature Communications]

Multimodal deep learning for Alzheimer's disease dementia assessmentReviewers' comments:

Reviewer #1 (Remarks to the Author):

This study presents a deep-learning model to predict a clinical diagnosis of AD dementia, non-AD dementia, as well as normal cognition and MCI (due to any aetiology). The noteworthy aspect of this study is the model. The results are, however, unsurprising and do not seem to provide a significant contribution to the literature (it is well established that in AD dementia there is substantial temporal lobe atrophy).

There are several issues:

- It is put forward that a deep-learning model like presented in this study is important for non-specialised clinicians, including general practitioners. GPs usually do not have access to an MRI scanner, and it would be for them to use this model, since one of the most important features is MRI based. For clinical trials biomarker evidence for e.g., amyloid pathology would be preferred, and is not included in this paper,
- A complication is that the model is trained on a clinical diagnosis. Although this is an understandable choice when a lot of data is required for training the model, it is not the most accurate choice, since a clinical label may not reflect the underlying aetiology. E.g., up to 30% of individuals with clinical AD dementia do not have AD underlying pathology. Indeed, figure 4d suggests that a very high AD probability score may not correspond to the highest pathology scores, and there is even 1 individual with 0 A, B, C pathological scores. This makes it also problematic to compare model performance with the clinician ratings: deviation between model prediction and clinicians reflects differences between the new clinicians vs the old ones who made the first diagnosis, rather than a better ability of the model to detect AD (which is trained on the 'old' first diagnoses). Also it is not clear why this comparison was not performed for AD vs nonAD, which would be a typical neurologist task.
- Classification of dementia vs non-dementia: A dementia diagnosis is determined based on low performance on neuropsychological tests scores and the MMSE, as such it is not a surprise that these perform highly accurate (since the tests are part of the diagnosis). A more difficult and relevant task would be to tease apart the AD vs nonAD dementias, and indeed this is reflected by the model performance. It is not so clear, however, why nonAD dementias are all put into 1 group? Do the authors expect that differences with AD in terms of symptoms and atrophy patterns are the same for FTD and DLB patients?

On combining cohorts:

- How was non-imaging data pooled, there is mention of NACC and OASIS-3 following Uniform Data Set guidelines), but how do the other cohorts fit in?
- How were MRI scans harmonised across cohorts? It is not clear whether all MRIs were processed again using the same pipeline. This still leaves other cohort specific effects (e.g., differences in scanning protocol, magnetic field strength etc.).
- How were cohort-specific missing data dealt with? E.g., APOE is missing in the parkinson and frontotemporal dementia cohorts. As such, based on the presence of APOE alone it would already be possible to predict AD vs nonAD dementia almost perfectly.

Fig4d :It would be helpful to see scatterplots of the probabilities vs the A, B and C scores, preferably with colors or symbols that indicate the cohort.

Table 1:

Please indicate n with pathologically confirmed diagnosis;

I find it hard to believe that AIBL has no data available about educational level, have the authors applied for AIBL data directly from the AIBL website, or was this through e.g., GAAIN?

the missing data seems to be related directly to AD vs nonAD diagnosis, for e.g., APOE is unavailable for nonAD cohorts. How was missing data handled in the model?

Reviewer #2 (Remarks to the Author):

This manuscript has looked at a combination of datasets consisting of more than 8,000 patients with dementia symptoms and sought to differentiate Alzheimer's disease diagnosis from other dementia causing disorders. The manuscript has performed in-depth analysis, with several layers of validation, including technical, clinical, and neuropathological. There are many similar studies using machine learning methods in the literature; therefore, while this study confirms previous studies, it brings extra validation and new insights by combining data sets at much larger scales than previous studies and with several different modalities (as mentioned above). I therefore believe this study is highly novel. The methods are clearly reported and meticulously designed. My comments are listed below:

- 1) It was unclear how the external validation data set was chosen to report AUC. The manuscript mentions splitting the dataset into folds, but the explanation is vague in reporting results. Is it an average across folds? Did the manuscript choose an external validation data set from the beginning to prevent bias?
- 2) Related to the point above, it remains unclear how reporting AUC, and other metrics are independent of tuning and training models.
- 3) The network analysis is not motivated enough and feels separated from the rest of the work.
- 4) Can the manuscript discuss the utility of the model in a real-world setting? The quality of real-world data is likely to differ from research data and may benefit from the discussion for deployment in future.
- 5) I suggest changing the labels in the figure to spell out the full words for readability.
- 6) Please report the confidence intervals for AUC measures in the results.
- 7) Do MRI data need harmonizations? There is a mention of harmonizing clinical data but not imaging. How can effects of centers be adjusted for?
- 8) It seems some disorders are likely to be drawn from specific data sets only (e.g., FTD). I wonder whether this may introduce bias in results (mixing effects of interest with confounders).

Arman Eshaghi

Reviewer #3 (Remarks to the Author):

Qiu et al presented deep learning models trained from clinical information and brain MRI to perform 3 classification tasks: 1) cognitively normal individuals vs cognitively impaired individuals (COGnc), 2) patients with dementia vs individuals without dementia (COGde), 3) Alzheimer's disease (AD) dementia cases vs non-AD dementia cases (AD vs non-ADD). For each task, the authors trained models with 3 different combinations of features: 1) MRI only, 2) non-imaging features only, 3) a "fusion" model that included both MRI and non-imaging features. The authors also presented "neuroimaging signatures" of AD and non-AD dementia via feature interpretation of brain regions that were implicated in the AD vs non-ADD task. The authors investigated correlations between their models' predictions and AD neuropathology. Finally, the authors compared the performance of their models against the diagnostic accuracy of neurologists and neuroradiologists.

The authors presented a very dense manuscript with lots of material. The comparisons between their models and the clinicians were particularly interesting, and this aspect of the paper will likely draw in a wide readership. I have some comments for the authors that are aimed at clarifying their results and improving the readability of their manuscript.

Major comments:

- 1) The results section was quite hard to follow. A lot of core ideas/terms are presented without enough context or explanation in this section. I had to jump back and forth between the methods and supplementary information sections a lot to understand the results. I get the impression that the authors may have originally structured the paper with the methods section presented before the results section. If the results are to be presented before the methods (as in the current form), it would be helpful to readers if the authors could briefly define their tasks (e.g. COGnc, COGde, COG-3way,

etc) and models (e.g. MRI-only, fusion, etc) and indicate which datasets were used for training and testing in a short summary paragraph at the beginning of the results section. In its current shape, the results don't stand alone as its own section without the reader being required to read the methods section first.

2) Can the authors expand Supplementary Table S8 to include the counts of AD and Parkinson's disease dementia (PDD) cases? The authors looked at PDD as part of their nADD subgroup analyses (for example in Supplementary Figures S1 and S2), but it's not described which datasets contain PDD or how many cases of PDD there were in each dataset.

3) The authors repeatedly claim throughout their manuscript that they used data from 8 separate cohorts. This is slightly misleading because the majority of the datasets (ADNI, AIBL, PPMI, NIFD, LBDSU, FHS) were used only in validation for one model (MRI-only). While this is revealed in Supplementary Table S1 and Supplementary Figure S9, this should be stated more explicitly in the main text of the manuscript.

4) Can the authors please clarify the caption of Figure 2? Are each of the rows in (a) and (b) representing results from the fusion models? The caption also states that (b) contains results for external datasets that include ones that were only used for the MRI-only model validation - so this simply cannot be correct for the top row, the COGnc task, because the caption says it is the fusion model.

5) The authors state on lines 213-215, "On visual inspection of the individual case, there was notable similarity between areas of high SHAP scores for the COG 3-way task and region-specific semiquantitative neuropathological scores obtained from autopsy". What is considered "notable similarity"? For example, I see a lot of differences between the distribution of the SHAP values and the neuropathological scores among the parietal and frontal lobes of the brain.

6) Since the neurologists and neuroradiologists were asked to perform different tasks (neurologists did the COGnc and COGde tasks, while the neuroradiologists did the AD vs nADD task), and they were provided different data (the neuroradiologists only received MRI, age and gender, while the neurologists also received information from clinical assessments), the text-based analysis seems biased so that the clustering would inherently result in the separation of the two types of clinicians. Because of these confounding factors, I'm not convinced that this analysis revealed patterns that were truly distinct to each of these clinicians' expertise as the authors suggest in lines 252-259.

Minor comments:

1) Figure 4a is missing the scale for SHAP value.

2) Is PET imaging of amyloid or quantifying amyloid or tau by CSF a part of the standard of care for clinicians who specialize in dementia? There are studies that have shown that clinicians' diagnoses or subsequent care management can change after receiving information about a patient's amyloid status (see Rabinovici et al 2019 JAMA <https://jamanetwork.com/journals/jama/fullarticle/2729371>, for example). How do the authors think the lack of information regarding amyloid may have impacted the diagnostic accuracy of the clinicians in their study?

Reviewer #4 (Remarks to the Author):

This is a well-designed study presenting a novel deep learning framework able to differentiate normal cognition, MCI, Alzheimer's disease and other dementias leveraging on a large multi-study cohort. The study reports superior performance for the developed framework compared to experienced radiographers and radiologists. One of the main advantages of the study is that a comprehensive evaluation of the cases by neurologists and neuroradiologists is reported and was used for comparisons. Four categories were classified: normal cognition, MCI, AD, non-AD dementia. Two tasks were evaluated first was cognition based on MRI – 3 class and then given dementia, AD or non-AD. Results are interpretable due to the use of SHAP. Overall conclusions are supported by the presented data.

There are several points that need to be further discussed and some suggested improvements.

Methods

Lines 36- 51

It appears the authors have relied on normalisation to MNI space as a step for harmonization however, this might not be enough. Methods such as ComBat (Fortin et al. 2018) have been developed and applied for multi-site data harmonisation. This would be particularly important since it needs to be ruled out that the observed performance might be driven partly by site differences. Though in figure SF11 the authors demonstrate no clear clustering, they should demonstrate also that the deep learning scheme is not sensitive to study/site effects. Hence, I suggest a further exploratory analysis where study ID will be included as a feature in the applied machine learning pipelines, especially when imaging data are included.

Line 42: There is a discordance in the number of regions reported here and in the caption of figure 3 (95 vs 57).

Line 94: It is unclear what imaging data have gone in the models

Lines 142-144

More details are needed about the manual alignment step in order for it to be reproducible.

Since interpretability of the results following deep learning is a key issue, more methodological details are needed about SHAP.

Results

The number of excluded scans due to the applied QC should be reported.

Though the fusion model improves performance on the non-parkinsonian dementias, there is no mention or explanation about a slightly worse performance in VD and FTD (SFigures 1 and 2 - Lines 139-141)

Discussion

As neuroradiologist 1 points, there wasn't a consensus approach between radiologists. This should be reported as a limitation.

It is unclear whether atypical AD cases existed in any of the cohorts, if this information is available it should be included and discussed in relation to the findings. This should also be discussed in the limitations.

The achieved performance and importance of the framework are discussed, however the discussion would benefit from an additional paragraph discussing the observed findings in figures 3 and 4.

Multimodal deep learning for Alzheimer's disease dementia assessment

Response to reviewer comments:

We sincerely thank the reviewers for providing constructive feedback on our original submission. We followed the suggestions from the reviewers and revised our manuscript accordingly. Our resubmission includes a document that tracked all the changes from the original submission as well as a clean copy.

Reviewer 1:

Comment 1: This study presents a deep-learning model to predict a clinical diagnosis of AD dementia, non-AD dementia, as well as normal cognition and MCI (due to any aetiology). The noteworthy aspect of this study is the model. The results are, however, unsurprising and do not seem to provide a significant contribution to the literature (it is well established that in AD dementia there is substantial temporal lobe atrophy).

Response 1: We thank the reviewer for their time in reviewing our manuscript and for recognition of the ‘noteworthy’ model around which we have constructed our research. We agree that temporal lobe atrophy is a well-established marker of AD pathology. However, the purpose of our work is not to demonstrate the role of temporal lobe changes and AD diagnosis. As the reviewer notes, this connection is well-established. Rather, the present research is aimed at more fully replicating the task of differentially diagnosing dementia within a single modeling framework. To this end, any correlations between model results and temporal lobe atrophy are meant to serve as confirmatory evidence of our computational results. By assimilating our outcomes with known patterns of neurodegenerative changes, we sought to confirm our novel deep learning approach with well-validated measures of AD pathology. Thus, while temporal lobe atrophy in AD dementia may be unsurprising, we believe that its high degree of correlation with our novel computational results is an important step to show consistency between our work and the accumulated knowledge surrounding this devastating disease.

Comment 2: It is put forward that a deep-learning model like presented in this study is important for non-specialized clinicians, including general practitioners. GPs usually do not have access to an MRI scanner, and it would be for them to use this model, since one of the most important features is MRI based.

Response 2: We greatly appreciate the reviewer’s comment, and their constructive intention to make us clarify the clinical utility of our deep learning model. While we agree that general practitioners typically do not have access to or review MRIs, our understanding is that some may order an MRI if deemed necessary. There are a few published papers that have delved into GP’s experience with ordering and reviewing MRIs:

1. S Skinner. MRI brain imaging. Aust Fam Physician. 2013 Nov;42(11):794-7.
2. M Robling, P Kinnersley, H Houston, M Hourihan, D Cohen, J Hale. An exploration of GPs’ use of MRI: a critical incident study. 1998 Jun;15(3):236-43.
3. A L Gough-Palmer, C Burnett, W M Gedroyc. Open access to MRI for general practitioners: 12 years’ experience at one institution - a retrospective analysis. The British Journal of Radiology, 82 (2009), 687–690.
4. S Kara, A Smart, T Officer, C Dassanayake, P Clark, A Smit, A Cavadino. Guidelines, training and quality assurance: influence on general practitioner MRI referral quality. Journal of Primary Health Care 11(4) 387-387, 2019.

We do appreciate that MRIs may or may not be available in general healthcare settings. Even so, we believe that our deep learning models can be useful to both specialists and GPs. For a situation when the practitioner (either a GP or a specialist such as a neurologist) can order an MRI, the deep learning model can provide a fair assessment of the person’s cognitive status using several modes of available data (i.e., MRI + non-imaging). Alternatively, when MRI cannot be ordered, we believe our non-imaging model can be useful. We anticipated such

scenarios while pursuing our research, and this is the reason why we developed several alternative models using different forms/modalities of data that can carry out the classification tasks. Certainly, our results demonstrate that the inclusion of MRI features boosts the performance of the predictive algorithm. However, the main goal of our work (i.e., the ability to carry out a multitask differential diagnosis of AD) is not limited by the availability of neuroimaging data. Thus, we feel that the diverse range of inputs that we have tested lends itself to clinical translation for specialized and non-specialized practitioners alike.

Comment 3: For clinical trials, biomarker evidence for e.g., amyloid pathology would be preferred, and is not included in this paper.

Response 3: We share the reviewer's opinion that amyloid pathology data would be the optimal means by which to confirm AD diagnosis in the context of clinical trials. While focus on clinical trials is extremely important to discover disease modifying therapies, our manuscript's primary goal was to develop and validate deep learning models that can serve as assistive tools for diagnosis across the care continuum. For this reason, we wanted to make sure our models can predict the output class label at least as well as the expert-level diagnosis using data that is collected within various clinical settings. Therefore, we selected primary clinical diagnosis as our output label during model development. Moreover, amyloid pathology data is not widely available, so relying on them to create deep learning models was not practical. Nonetheless, we used available neuropathology data on 110 subjects from the ADNI, NACC and FHS cohorts to validate our model. We believe neuropathological validation of our models is a strength of our study, and it was reassuring to see that models that were primarily developed using clinical information to have confirmatory evidence on autopsy data. To honor the reviewer's comment and to stay consistent with the broad theme of our work, we did not include any discussion related to clinical trials in our revised manuscript.

Comment 4: A complication is that the model is trained on a clinical diagnosis. Although this is an understandable choice when a lot of data is required for training the model, it is not the most accurate choice, since a clinical label may not reflect the underlying aetiology, e.g., up to 30% of individuals with clinical AD dementia do not have AD underlying pathology. Indeed, figure 4d suggests that a very high AD probability score may not correspond to the highest pathology scores, and there is even 1 individual with 0 A, B, C pathological scores. This makes it also problematic to compare model performance with the clinician ratings: deviation between model prediction and clinicians reflects differences between the new clinicians vs the old ones who made the first diagnosis, rather than a better ability of the model to detect AD (which is trained on the 'old' first diagnoses).

Response 4: We truly appreciate the reviewer's comment. As described in the previous response, our goal was to develop and validate deep learning models that can serve as assistive tools for diagnosis across the care continuum. For this reason, we wanted to make sure our models can predict the output class label at least as well as the expert-level clinical diagnosis. Therefore, we selected primary clinical diagnosis as our output label during model development. We must note that the clinical labels in each dataset were obtained by a consensus review of all relevant data by multidisciplinary panels including neurologists, neuropsychologists, and neuroradiologists. We have tried to emulate clinical scenarios at multiple levels by referring to information that would be routinely collected in a primary care setting as well as that collected in specialty neurology clinics. Using these varied depths of information, we presented multiple non-imaging and fusion

models each with distinct AUC values. Moreover, not all large multicenter databases such as those used in our work collect neuropathology data in equal measure to clinical and imaging information. Therefore, we feel that our usage of this information remains well up to the standards of the AD research community.

We also thank the reviewer for rightly pointing out a single case, whose correlation with model predictions and pathology scores was not in correspondence. Clearly, this confirms that models or even clinicians may not always be accurate in terms of identifying disease, and this observation speaks to the real-world diagnostic standards in clinical neurology. We may speculate that the individual noted as having no relevant pathology is representative of the “false positives” that the reviewer has astutely pointed out and we have included this in our limitations section. Nevertheless, correlation of model predictions with neuropathology in most of the other cases provides a statistical basis by which to underscore the validity of our deep learning model. In summary, we believe that our labels are reflective of more than just the diagnostic idiosyncrasies of individual practitioners.

Lastly, we would like to demonstrate to the reviewer our efforts to develop models based upon biospecimens. Thus, we trained two additional models using CSF biomarker measurements: (i) a “CSF-only” CatBoost classifier that took as input only scalar values of $A\beta$ amyloid, phosphorylated tau, and total tau and (ii) a “fusion CSF” model in which CNN-derived COG and ADD scores from MRI were used alongside CSF biomarkers. These models were developed using 195 subjects from the NACC dataset for whom CSF information was available within 6 months of MRI. Below, we present (a) ROC and PR curves for the COG_{NC} task. The “CSF-only” (red curves) and “fusion CSF” (yellow curves) are shown in comparison to an MRI-only model (green) and a separate CatBoost model trained with only non-imaging clinical variables (blue). We also show (b) ROC and PR curves for the COG_{DE} task. Notably, there were no individuals with nADD in the NACC dataset with CSF information available, thus precluding development of biomarker models for ADD vs. nADD classification.

Based on these results, it appears that CSF-based modeling is approximately on par with the accuracy of clinical diagnoses. Given that our present work does not focus on biomarkers for AD diagnosis, we feel that CSF modeling may not fit within our overall narrative for this study. However, if the Reviewer feels that this would be a necessary adjunct to our work, we would be happy to include the above figure within the Supplement.

Comment 5: Also, it is not clear why this comparison was not performed for AD vs nonAD, which would be a typical neurologist task.

Response 5: We appreciate the reviewer's close reading of our work and their attention in noting the decision to not include a model-to-neurologist comparison of AD vs. non-AD diagnosis. This choice reflects differences in the model's prediction process and the diagnostic simulation in which clinicians participated.

As we described in the manuscript, both neurologists and the model received MRI scans and non-imaging features from 100 patients spread evenly across NC, MCI, AD, and non-AD categories. During neurologist review, all simulated patients were eligible to receive any of the four labels depending on the physician's clinical impression. Conversely, during the deep learning model's processing, only those patients who first received an internal label of *dementia* were eligible for further assessment of AD vs. non-AD. The physicians' diagnostic task was thus a "one-step" process, whereas the model's classification was a "two-step" process which depended on separating out a sub-population of demented patients. Therefore, by the time of AD vs. non-AD classification, the model has honed in on a sub-population of persons, likely composed of distinct cases than those identified by the neurologist. For this task, then, the algorithm and the clinicians are operating on different sets of persons, thus eliminating the possibility of a "fair" head-to-head comparison. We have noted this within our updated Methods within the "*Head-to-head comparison with clinicians*" and write as follows:

"Notably, the model was not directly compared to neurologists for the task of ADD versus nADD discrimination, given that our framework only performs this prediction on patients internally-identified as demented. Given this computational pre-selection, it was impossible to consistently compare a common cohort of patients with neurologists who also must perform a differential diagnosis of NC, MCI, AD, and nADD."

Nevertheless, we did feel it important to provide some grounding by which readers can compare our deep learning framework with clinician performance for AD vs. nonAD diagnosis. As such, for both the model and neurologists, we have provided the sensitivity, specificity, F-1 score, and MCC for detecting each disease class within Table S5. We hope that these metrics will allow direct comparison where ROC and PR curves could not.

Comment 6: Classification of dementia vs non-dementia: A dementia diagnosis is determined based on low performance on neuropsychological tests scores and the MMSE, as such it is not a surprise that these perform highly accurate (since the tests are part of the diagnosis). A more difficult and relevant task would be to tease apart the AD vs nonAD dementias, and indeed this is reflected by the model performance. It is not so clear, however, why nonAD dementias are all put into 1 group? Do the authors expect that differences with AD in terms of symptoms and atrophy patterns are the same for FTD and DLB patients?

Response 6: We appreciate the reviewer’s careful consideration of the variables used within model development and their role within the differential diagnosis of dementia. We would like to note, however, that the essential focus of our work is the specific identification of AD amidst potential confounding diagnoses. Certainly, we do not expect that frontal and temporal lobar changes present in FTD would be shared in a subcortical dementia such as DLB or Parkinson’s dementia. However, given that our approach was still geared towards the discrimination of AD from other dementia syndromes, we still felt it appropriate within our main analysis to group non-AD dementias as a singular entity. Indeed, we feel that the temporo-parietal changes within AD would be likely to distinguish it from non-AD dementias. We have also acknowledged our reasoning for grouping together nADDs as a singular entity as a deliberate decision within our Methods section as follows:

“Notably, we elected to conglomerate all nADD subtypes into a singular label given that subdividing model training across an arbitrary number of prediction tasks ran the risk of diluting overall diagnostic accuracy.”

However, we fully agree with the reviewer that the atrophy patterns and symptoms across various types of dementia are different. Thus, we systematically conducted several sub-group analyses on AD versus each non-AD type of dementia. Firstly, we present the classification performance between AD and each other type of dementia in Figures S1-S2. Secondly, we also conducted the subgroup analyses based on the interpretable SHAP heatmaps. In Figure S8, we generated similar violin plots as Figure 4c to make direct comparison on the atrophy patterns between AD and each of the non-AD dementia types, including vascular dementia, frontotemporal lobe dementia, Lewy body dementia (which is a combination of the Parkinson disease dementia and dementia with Lewy bodies).

Comment 7: How was non-imaging data pooled, there is mention of NACC and OASIS-3 following Uniform Data Set guidelines), but how do the other cohorts fit in?

Response 7: Our consideration of which cohorts to use for development on non-imaging models was informed by a) data standardization protocols and b) data availability. In terms of standardization of non-imaging variables, we note that both NACC and OASIS-3 follow Uniform Dataset (UDS) guidelines [1]. Therefore, non-imaging variables derived from these datasets may already be considered as harmonized given that their definition and collection follows identical protocols.

Our consideration of which cohorts to use for development on non-imaging models was informed by a) data standardization protocols and b) data availability. In terms of standardization of non-imaging variables, we note that both NACC and OASIS-3 follow Uniform Dataset (UDS) guidelines [1]. Therefore, non-imaging variables derived from these datasets may already be considered as harmonized given that their definition and collection follows identical protocols. Furthermore, we note that non-imaging variables had considerably less availability or the data was not collected in a comparable manner in the ADNI, AIBL, FHS, NIFD, PPMI, and LBDSU. We completed a survey of the available non-imaging data in each of the 8 cohorts (See File: Data_Harmonization_Table.xlsx). Clinical testing has evolved over the years and thus, although ADNI, AIBL, FHS, and NIFD studies may collect comparable data in similar clinical domains, direct comparisons may not be feasible. To demonstrate this, we have now included an additional Supplementary Figure S12 which shows the availability of all non-imaging variables across the

dataset. Due to the significant differences between the cohorts regarding non-imaging information, we were limited in our ability to develop and test non-imaging models in these populations without resorting to excessive feature imputation.

[1] Monsell SE, Dodge HH, Zhou XH, et al. Results From the NACC Uniform Data Set Neuropsychological Battery Crosswalk Study. *Alzheimer Dis Assoc Disord.* 2016;30(2):134-139. doi:10.1097/WAD.0000000000000111

Comment 8: How were MRI scans harmonised across cohorts? It is not clear whether all MRIs were processed again using the same pipeline. This still leaves other cohort specific effects (e.g., differences in scanning protocol, magnetic field strength etc.).

Response 8: All MRIs were harmonized according to the common FSL pipeline described within our Methods section. We have adjusted the description of this pipeline within our Methods to highlight the fact that it was applied identically to all cohorts. Specifically, we now state:

“To harmonize neuroimaging data between cohorts, we developed a pipeline of preprocessing operations (Fig. S11) that was applied in identical fashion to all MRIs used in our study.”

Furthermore, based on the reviewer’s helpful feedback, we have taken steps to rule out cohort-, site-, and scanner-specific biases both within our processed training data and the hidden layers of our trained models. Specifically, we now include a series of t-distributed Stochastic Neighbor Embedding (tSNE) plots drive from both downsampled (8x) MRI data and hidden-layer embeddings within our CNN to rule-out systematic biases due to a) study cohort (e.g. NACC, ADNI, etc.) b) specific NACC Alzheimer’s Disease Research Center (ADRC) and c) MRI scanner manufacturer. We have elected to include these plots as a separate figure within our main manuscript (revised Figure 2). The graphic of this figure may be found at the end of this response, and we describe the subsections as follows:

- **Figure 2a. MRI tSNE by Cohort:** This figure demonstrates 2D tSNE embeddings of 8x-downsampled MRIs. Each datapoint is colored by its overall cohort label. No discernible clustering is observed, indicating that raw imaging data likely did not differ systematically between our training set and various testing sets.
- **Figure 2b. MRI tSNE by ADC Site: MRI tSNE by ADRC Site:** This figure demonstrates 2D tSNE embeddings of 8x-downsampled MRIs. Each data point is colored according to its unique ADRC ID. No discernible clustering is observed based on specific ADRC, thus indicating that imaging data within our NACC training cohort is unlikely to have site-specific biases
- **Figure 2c. MRI tSNE by MRI Manufacturer:** This figure demonstrates 2D tSNE embeddings of 8x-downsampled MRIs from the NACC dataset. Each data point is colored according to the brand of the MRI scanner-either Philips, GE, or Siemens. No discernible clustering is observed based on specific companies. Therefore, imaging data within our training cohort is unlikely to have systematic biases due to differing manufacturers.
- **Figure 2d. Hidden Layer tSNE by Cohort Label:** This figure demonstrates 2D tSNE embeddings derived from the penultimate hidden layer of our CNN model. Each

datapoint is colored according to its overall cohort label. While there is notable clustering among NACC data points, we note that this pattern did not extend to external validation datasets. This likely indicates that cohort-specific biases did not affect validation performance.

- Figure 2e. Hidden Layer tSNE by ADRC ID: This figure demonstrates 2D tSNE embeddings derived from the penultimate hidden layer of our CNN model for NACC scans. Each data point is colored according to the corresponding ADRC ID, and there is no specific clustering observed on this basis. Therefore, we conclude that our model did not learn site-specific information while training.
- Figure 2f. Hidden Layer tSNE by MRI Manufacturer: This figure demonstrates 2D tSNE embeddings derived from the penultimate hidden layer of our CNN model when trained with NACC data. Each data point is colored according to the brand of the MRI scanner—either Philips, GE, or Siemens. No discernible clustering is observed based on specific companies. Therefore, our clinical model’s predictions were unlikely to have been unduly influenced by systematic differences in MRI instruments.

Overall, we believe that our use of tSNE on both raw data and internal model representations helps us to conclude neither our post processed MRIs nor our model predictions were significantly affected by differences in scanning protocol, magnetic field strength, or other imaging parameters that could have biased our prediction tasks.

Lastly, in addition to our tSNE conclusions above, we calculated Mutual Information Scores (MIS) between diagnostic labels, scanner brand, and ADRC ID. The MIS measures the similarity between two labels on the same dataset. Values of 1 indicate perfect concordance between two sets of label assignments, whereas values of 0 indicate random association between these sets. For the relationship of scanner brand to diagnostic label, we found a MIS of 0.010. For the relationship of ADRC ID to the diagnostic label, we found a MIS of 0.065. These values indicate near-randomness of association between diagnoses, scanner, and clinical site. We feel that this result strongly evidences a null association between the potential confounders noted by the reviewer and our outcomes of interest. We report our MIS values in the Results section of our manuscript. Lastly within Figure 2g and Figure 2h, we illustrate counts of diagnostic labels stratified by scanner brand and ADRC ID, respectively. We hope that these subfigures, like our tSNE plots, will help readers to appreciate a lack of confounding relationships.

Comment 9: How were cohort-specific missing data dealt with? E.g., APOE is missing in the parkinson and frontotemporal dementia cohorts. As such, based on the presence of APOE alone it would already be possible to predict AD vs nonAD dementia almost perfectly.

Response 9: We appreciate the reviewer's careful consideration of the potential for biasing cohort-specific data. Certainly, we too foresaw the potential for asymmetry in variable selection to falsely confer inflated testing accuracy; as such, we only utilized non-imaging variables which were common to the NACC and OASIS-3 datasets. APOE was only included within our demographic table for the purposes of presenting cohort characteristics. We apologize that a fuller explanation of non-imaging variable selection was not included, and have now explicitly addressed this aspect of our modeling approach within the Methods section, with a sentence that reads as follows:

“In addition to an MRI-only model, we developed a range of traditional machine learning classifiers using all available non-imaging variables shared between the NACC and the OASIS datasets.”

Comment 10: Fig4d :It would be helpful to see scatterplots of the probabilities vs the A, B and C scores, preferably with colors or symbols that indicate the cohort.

Response 10: In addition to the map included in Figure 4d, we provided a swarmplot demonstrating COG scores (i.e., dementia probabilities) derived from the model as a function of A, B, and C scores. We note that triangular, circular, and rectangular markers are supplied in this figure to indicate the cohort from which an individual datapoint is derived.

Comment 11: Table 1: Please indicate n with pathologically confirmed diagnosis; I find it hard to believe that AIBL has no data available about educational level, have the authors applied for AIBL data directly from the AIBL website, or was this through e.g., GAAIN?

Response 11: We thank the reviewer for their helpful suggestion. Unfortunately, all diagnostic labels were clinically-determined rather than pathologically. Additionally, we indeed applied from GAAIN, and therefore we did not obtain educational data by applying directly to AIBL.

Comment 12: The missing data seems to be related directly to AD vs nonAD diagnosis, for e.g., APOE is unavailable for nonAD cohorts. How was missing data handled in the model?

Response 12: We agree with the reviewer that there are systematic differences in data availability between cohorts, and we have now elected to demonstrate this graphically in Fig. S12. However, as we note above, we only compared non-imaging and fusion models in NACC and OASIS-3. Missing data in non-AD cohorts (PPMI, NIFD, LBDSU) therefore played no role in biasing model predictions, as non-imaging data from these sources was not used in our work. In the case of APOE, we report this genotype only for demographic purposes and do not include it as a feature in any of our models.

In both the NACC and OASIS studies, missing non-imaging data was simulated with the k-nearest neighbor (kNN) feature imputation method provided from the Scikit Learn package with default settings. Of note, kNN imputation was in no way dependent on disease label, and we therefore feel this procedure was unlikely to have biased disease predictions in either of these datasets as well.

Reviewer #2:

Comment 1: This manuscript has looked at a combination of datasets consisting of more than 8,000 patients with dementia symptoms and sought to differentiate Alzheimer's disease diagnosis from other dementia causing disorders. The manuscript has performed in-depth analysis, with several layers of validation, including technical, clinical, and neuropathological. There are many similar studies using machine learning methods in the literature; therefore, while this study confirms previous studies, it brings extra validation and new insights by combining data sets at much larger scales than previous studies and with several different modalities (as mentioned above). I therefore believe this study is highly novel. The methods are clearly reported and meticulously designed.

Response 1: We greatly appreciate the reviewer for providing a nice summary of our manuscript, and for noting that our manuscript is *highly novel*, and the *methods were clearly reported and meticulously designed*. We are grateful for the reviewer's appreciation that our manuscript has performed an *in-depth analysis with several layers of validation*.

Comment 2: It was unclear how the external validation data set was chosen to report AUC. The manuscript mentions splitting the dataset into folds, but the explanation is vague in reporting results. Is it an average across folds? Did the manuscript choose an external validation data set from the beginning to prevent bias? Related to the point above, it remains unclear how reporting AUC, and other metrics are independent of tuning and training models.

Response 2: We thank the reviewer for the opportunity to clarify this point. During the course of training, we attempted various groupings of hyperparameters. For each, we utilized 5-fold cross validation using the NACC dataset for development. In each iteration of cross-validation, we utilized 3 folds for model training, 1-fold for selection of the optimal performance epoch, and 1-fold for testing of the model at the selected epoch. In addition to the hold-out test fold from NACC, the selected model was also tested on our external cohorts. Our overall training process is summarized graphically in Fig. S13. In our manuscript, we report mean, standard deviation, and 95% confidence intervals of all performance metrics (including AUC) across these five folds. To minimize confusion, we have also added an additional sentence within our Methods section to describe our cross-validation strategy, which now reads:

“We trained all models on the NACC dataset using cross validation. NACC was randomly divided into 5 folds of equal size with constant ratios of NC, MCI, AD, and nADD cases. We trained the model on 3 of the 5 folds and used the remaining two folds for validation and testing, respectively. Each tuned model was also tested on the full set of available cases from external datasets. Performance metrics for all models were reported as a mean across five folds of cross validation along with standard deviations and 95% confident intervals. A graphical summary of our cross-validation strategy may be found within Fig. S13.”

Comment 3: The network analysis is not motivated enough and feels separated from the rest of the work.

Response 4: We appreciate the reviewer's concern in this regard. We agree that the certain network analyses may be reasonably viewed as orthogonal to the overall aim of our work.

Specifically, we feel that this is particularly true of the text-based network currently presented in our original submission as Figures 5c and 5d. To this end, we have elected to remove these figures from our paper.

Conversely, we do believe that the network analysis originally included as Figures 3d and 3e presents useful information regarding global differences in SHAP scoring (and thus, model inference) between AD and non-AD populations. Therefore, we hope to keep this aspect of our work in the main body of the manuscript, and have presented them within our updated Figure 4d and 4e.

Comment 4: Can the manuscript discuss the utility of the model in a real-world setting? The quality of real-world data is likely to differ from research data and may benefit from the discussion for deployment in future.

Response 4: The reviewer insightfully notes that clinical translation of an academic model is likely to encounter challenges, and we are happy to expand upon this aspect. Ultimately, we hope that this paper will serve as an exemplary study from which to start prospective testing of automated systems in dementia diagnosis. Therefore, we have now included the following within the Discussion section:

“Our work builds on prior efforts to construct automated systems for the diagnosis of dementia. Previously, we developed and externally validated an interpretable deep learning approach to classify AD using multimodal inputs of MRI and clinical variables.¹⁷ This approach relied on a contrived scenario of discriminating individuals into binary outcomes, which simplified the complexity of a real-world setting. Our current work extends this framework by mimicking a memory clinic setting that accounts for cases along the entire cognitive spectrum. Though numerous groups have taken on the challenge of nADD diagnosis using deep learning,^{18,19,26-28} even these tasks were constructed as simple binary classifications between disease subtypes. Given that the clinical practice of medicine rarely reduces to a choice between two pathologies, integrated models with the capability to more fully replicate the differential diagnosis process of experts are needed before deep learning models can be touted as assistive tools for clinical-decision support. Our results demonstrate a strategy for expanding the scope of diagnostic tasks using deep learning, while also ensuring that the predictions of automated systems remain grounded in established medical knowledge.”

Functionally, we also contend that the flexibility of inputs afforded by our approach is a necessary precursor to clinical adoption at multiple stages of dementia. Given that subgroup analyses suggested significant 4-way diagnostic capacity on multiple combinations of training data (i.e., demographics, clinical variables, and neuropsychological tests), our overall framework is likely adaptable to many variations of clinical practice without requiring providers to significantly alter their typical workflows. For example, general practitioners (GPs) frequently perform cognitive screening with or without directly ordering MRI tests, whereas memory specialists typically expand testing batteries to include imaging and advanced neuropsychological testing. This ability to

integrate along the clinical care continuum, from primary to tertiary care allows our deep learning solution to address a two-tiered problem within integrated dementia care by providing a tool for both screening and downstream diagnosis.

Comment 5: I suggest changing the labels in the figure to spell out the full words for readability.

Response 5: We thank the reviewer for suggesting this as an improvement. We have now made sure to fully list all abbreviated terms in figure captions in our revised manuscript.

Comment 6: Please report the confidence intervals for AUC measures in the results.

Response 6: We share the reviewer's opinion that reporting of confidence intervals improves the strength of our results. 95% confidence intervals are now included with all AUC and AP measures.

Comment 7: Do MRI data need harmonizations? There is a mention of harmonizing clinical data but not imaging. How can the effects of centers be adjusted for?

Response 7: We indeed took steps to harmonize imaging data using a common FSL pipeline described within our Methods section and illustrated in Figure S11. Steps of this process included standardization of MRI studies to standard MNI-152 space and applying bias field corrections. We also shared the reviewer's concern that "effects of centers" must be excluded as biasing factors in our analysis. Therefore, we have now included a series of tSNE plots as an additional figure (Figure 2) within our main manuscript to exclude systematic biases in raw data and internal network representations alike. A full discussion of this figure and its corresponding subplots may be found in Response 8 to Reviewer 1 above, and we feel that these analyses effectively rule-out cohort-, site-, and scanner-specific biases for disease prediction tasks. Similarly, we note that additional description of our MRI harmonization pipeline has been added to our revised Methods, with specific wording that may once again be found in Response 8 to Reviewer 1.

Comment 8: It seems some disorders are likely to be drawn from specific data sets only (e.g., FTD). I wonder whether this may introduce bias in results (mixing effects of interest with confounders).

Response 8: We thank the reviewer for noting this potential bias, and for the opportunity to address this important concern. Following from our results referenced above, we note that our new Figure 2a and Figure 2d illustrate a lack of tSNE clustering based on specific cohorts. The absence of cohort-specific aggregations extends to both MRI data and hidden layer activations within our deep learning framework, thus indicating that raw input data and learned representations of disease state were unbiased by dataset-specific effects.

Furthermore, in the development of all non-imaging models, we made sure to only use variables that were shared between NACC and OASIS-3. Similarly, to our efforts in harmonizing MRI data, we feel that this approach allowed us to avoid dataset-specific variables that a model could unfairly use for inferring classification labels.

Reviewer #3:

Comment 1: Qiu et al presented deep learning models trained from clinical information and brain MRI to perform 3 classification tasks: 1) cognitively normal individuals vs cognitively impaired individuals (COGnc), 2) patients with dementia vs individuals without dementia (COGde), 3) Alzheimer's disease (AD) dementia cases vs non-AD dementia cases (AD vs non-ADD). For each task, the authors trained models with 3 different combinations of features: 1) MRI only, 2) non-imaging features only, 3) a "fusion" model that included both MRI and non-imaging features. The authors also presented "neuroimaging signatures" of AD and non-AD dementia via feature interpretation of brain regions that were implicated in the AD vs non-ADD task. The authors investigated correlations between their models' predictions and AD neuropathology. Finally, the authors compared the performance of their models against the diagnostic accuracy of neurologists and neuroradiologists. The authors presented a very dense manuscript with lots of material. The comparisons between their models and the clinicians were particularly interesting, and this aspect of the paper will likely draw in a wide readership. I have some comments for the authors that are aimed at clarifying their results and improving the readability of their manuscript.

Response 1: We thank the reviewer for nicely summarizing our work, and for noting that the task related to model comparison with the clinicians *will likely draw in a wide readership*. We also appreciate the reviewer for their comments to improve the readability of our manuscript.

Comment 2: The results section was quite hard to follow. A lot of core ideas/terms are presented without enough context or explanation in this section. I had to jump back and forth between the methods and supplementary information sections a lot to understand the results. I get the impression that the authors may have originally structured the paper with the methods section presented before the results section. If the results are to be presented before the methods (as in the current form), it would be helpful to readers if the authors could briefly define their tasks (e.g. COGnc, COGde, COG-3way, etc) and models (e.g. MRI-only, fusion, etc) and indicate which datasets were used for training and testing in a short summary paragraph at the beginning of the results section. In its current shape, the results don't stand alone as its own section without the reader being required to read the methods section first.

Response 2: We thank the reader for helpful suggestions which will help to improve the readability of our work. We have made extensive revisions to our Results section which include an initial "Overview" section that introduces our models, tasks, and general training strategy. In addition, we have also organized a glossary of terms that will assist readers with the terminology we make use of in our paper. This reference may be found as **Box 1** within the Results section of our revised manuscript, and is included for the Reviewer's convenience below:

Box 1. Glossary of Terms

Diagnostic Tasks:

COG task: Multiclass prediction of NC, MCI, and DE categories. May be further subdivided into the following subtasks:

COG_{NC}: The separation of persons with NC from those with MCI or DE.

COG_{MCI}: The separation of persons with MCI from those with NC or DE.

COG_{DE}: The separation of persons with DE from those with NC or MCI.

ADD task: Separation of persons with AD from those with nADD given an initial diagnosis of DE.

4-way task: Complete separation of NC, MCI, AD, and nADD cases. Accomplished by successive completion of the COG and ADD tasks.

Model-Derived Cognitive Metrics

DEMO score: "DEmentia MOdel" score. A continuous measure for overall cognitive status ranging from 0 (NC) to 1 (MCI) to 2 (DE). DEMO score thresholding enables completion of the COG task and its subtasks.

ALZ score: "ALZheimer's" score. A continuous measure from 0 (nADD) to 1 (AD) that corresponds with the probability that a person has Alzheimer's disease dementia. ALZ score thresholding enables completion of the ADD task.

Model Types:

MRI-only model: A convolutional neural network (CNN) that uses MRI scans and no other information to complete the COG and ADD tasks.

Non-imaging model: A traditional machine learning classifier that uses demographics, past medical history, neuropsychological testing, and functional assessments to complete the COG and ADD tasks.

Fusion model: A hybrid model composed of a CNN linked to a CatBoost classifier. The CNN portion computes DEMO and ALZ scores from MRI which are concatenated with non-imaging clinical variables. The CatBoost model then successively completes COG and ADD tasks.

Comment 3: Can the authors expand Supplementary Table S8 to include the counts of AD and Parkinson's disease dementia (PDD) cases? The authors looked at PDD as part of their nADD subgroup analyses (for example in Supplementary Figures S1 and S2), but it's not described which datasets contain PDD or how many cases of PDD there were in each dataset.

Response 3: We thank the reviewer for this helpful suggestion, which will help to improve the transparency of our methods. We have now expanded our Supplementary Table S8 (now Supplementary Table S7) to include counts of PDD cases. These are included along with counts of Dementia with Lewy Bodies (DLB) cases, under the larger class of Lewy Body Dementia (LBD).

Comment 4: The authors repeatedly claim throughout their manuscript that they used data from 8 separate cohorts. This is slightly misleading because the majority of the datasets (ADNI, AIBL, PPMI, NIFD, LBDSU, FHS) were used only in validation for one model (MRI-only). While this is revealed in Supplementary Table S1 and Supplementary Figure S9, this should be stated more explicitly in the main text of the manuscript.

Response 4: We agree that a more explicit acknowledgement of how datasets were used for validation would both enhance readability and dispel concerns about transparency. To this end,

we have adjusted the manuscript to reflect a more precise specification of the fact that OASIS was the only external cohort used for fusion model validation, whereas the “full” 8-cohort collection was available for MRI model validation. In our Methods section, under the description of each model and its development, we now explicitly state which datasets were used for training and external testing. For our MRI-only model, this reads:

The MRI-only model was trained using the NACC dataset and validated on all the other cohorts. To facilitate presentation of results, we pooled data from all the external cohorts (ADNI, AIBL, FHS, LBDSU, NIFD, OASIS and PPMI), and computed all the model performance metrics.

For our non-imaging model, we state:

In addition to an MRI-only model, we developed a range of traditional machine learning classifiers using all available non-imaging variables shared between the NACC and the OASIS datasets.

For our fusion model, we state:

As with our non-imaging model, development and validation of fusion models was limited to NACC and OASIS only given limited availability of non-imaging data in other cohorts.

Comment 5: Can the authors please clarify the caption of Figure 2? Are each of the rows in (a) and (b) representing results from the fusion models? The caption also states that (b) contains results for external datasets that include ones that were only used for the MRI-only model validation - so this simply cannot be correct for the top row, the COGnc task, because the caption says it is the fusion model.

Response 5: Indeed, as the reviewer points out, each row in this figure corresponds to fusion models. Furthermore, we regrettably acknowledge that the caption refers to all external datasets when the result in actuality corresponds only to OASIS. We thank the reviewer for their close reading, which has now allowed us to correct this error. The caption of the new figure (now updated as Figure 3) will read as follows:

Figure 3: Fusion model performance. (a-b) The receiver operating characteristic (ROC) curves showing the true positive rate versus the false positive rate and the precision-recall (PR) curves showing the positive predictive value versus sensitivity on **the (a) the National Alzheimer’s Coordinating Center (NACC) test set and (b) OASIS external validation set, respectively.** The first row in (a) and (b) denotes the performance of the fusion model (CNN + CatBoost) on the COGNC task. The second row shows ROC and PR curves for the COGDE task. The third row illustrates model performance for the classification task focused on discriminating Alzheimer’s disease dementia (ADD) from dementia cases with other etiologies (nADD). For each curve, the mean area under the curve (AUC) was computed. In each plot, the mean ROC/PR curve and the standard deviation are shown as the bold line and shaded region, respectively.

The dotted lines in each plot indicate the classifier with the random performance level. (c-d) Fifteen features with highest mean absolute SHAP values from the fusion model are shown for the COG and ADD tasks, respectively. For each of these tasks, the MRI scans, demographic information, medical history, functional assessments, and neuropsychological test results were used as inputs to the deep learning model. The left plots in (c) and (d) illustrate the distribution of SHAP values and the right plots show the mean absolute SHAP values. All the plots in (c) and (d) are organized in decreasing order of mean absolute SHAP values. (e-f) For comparison, we also constructed traditional machine learning models to predict cognitive status and ADD status using the same set of features used for the deep learning model, and the results are presented in (e) and (f), respectively. The heat maps show fifteen features with the highest mean absolute SHAP values obtained for each model.

Comment 6: The authors state on lines 213-215, “On visual inspection of the individual case, there was notable similarity between areas of high SHAP scores for the COG 3-way task and region-specific semiquantitative neuropathological scores obtained from autopsy”. What is considered “notable similarity”? For example, I see a lot of differences between the distribution of the SHAP values and the neuropathological scores among the parietal and frontal lobes of the brain.

Response 6: We share the reviewer’s view that “notable [visual] similarity” is a subjective standard by which to judge correlation between our deep learning predictions and neuropathology data. To this end, we have removed this remark from our Results section and instead focus on the remainder of our statistical analysis (i.e., Pearson correlations and ANOVA with Tukey-Kramer testing).

Comment 7: Since the neurologists and neuroradiologists were asked to perform different tasks (neurologists did the COGnc and COGde tasks, while the neuroradiologists did the AD vs nADD task), and they were provided different data (the neuroradiologists only received MRI, age and gender, while the neurologists also received information from clinical assessments), the text-based analysis seems biased so that the clustering would inherently result in the separation of the two types of clinicians. Because of these confounding factors, I’m not convinced that this analysis revealed patterns that were truly distinct to each of these clinicians’ expertise as the authors suggest in lines 252-259.

Response 7: We agree that the differing tasks given to clinicians may introduce bias in textual analysis. Upon much discussion since our initial submission, we have elected to remove all text-based analyses from this manuscript.

Comment 8: Figure 4a is missing the scale for SHAP value.

Response 8: We thank the reviewer for noting this important detail. We have included a SHAP scale in our former Figure 4a (now revised Figure 5a).

Comment 9: Is PET imaging of amyloid or quantifying amyloid or tau by CSF a part of the standard of care for clinicians who specialize in dementia? There are studies that have shown that clinicians’ diagnoses or subsequent care management can change after receiving information about a patient’s amyloid status (see Rabinovici et al 2019 JAMA

<https://jamanetwork.com/journals/jama/fullarticle/2729371>, for example). How do the authors think the lack of information regarding amyloid may have impacted the diagnostic accuracy of the clinicians in their study?

Response 9: We thank the reviewer for sharing this fascinating article from Rabinovici and colleagues demonstrating changes in patient management after amyloid PET. This work was conducted as part of the Imaging Dementia-Evidence for Amyloid Scanning (IDEAS) study, which is a research initiative to assess the impact of PET scans on AD care and outcomes. To our knowledge, however, neither PET imaging nor quantification of CSF markers are part of standard of care for AD diagnosis outside of research settings. Recently, a consensus of the Amyloid Imaging Task Force, the Society of Nuclear Medicine, and the Alzheimer's Association concluded that amyloid imaging is not appropriate for patients meeting core clinical criteria for probable AD with a typical age of onset [1]. Furthermore, the most recent National Institute on Aging/Alzheimer's Association diagnostic guidelines for Alzheimer's diagnosis explicitly do *not* advocate the use of biomarkers such as CSF tau [2]. The lack of endorsement for biomarkers stems from several factors, including the limited availability of these studies in community settings, evidence of insufficient standardization of measurements between locales, and the strong performance of clinical criteria for diagnosis. Therefore, we believe that the lack of amyloid markers (either from PET or CSF) played little or no role in impacting the diagnostic accuracy of clinicians, as most of the trained neurologists and neuroradiologists participating in our study practice without this information in most suspected dementia cases.

[1] Johnson, Keith A., et al. "Appropriate use criteria for amyloid PET: a report of the Amyloid Imaging Task Force, the Society of Nuclear Medicine and Molecular Imaging, and the Alzheimer's Association." *Alzheimer's & Dementia* 9.1 (2013): E1-E16.

[2] McKhann, Guy M., et al. "The diagnosis of dementia due to Alzheimer's disease: recommendations from the National Institute on Aging-Alzheimer's Association workgroups on diagnostic guidelines for Alzheimer's disease." *Alzheimer's & dementia* 7.3 (2011): 263-269.

Reviewer #4:

Comment 1: This is a well-designed study presenting a novel deep learning framework able to differentiate normal cognition, MCI, Alzheimer's disease and other dementias leveraging on a large multi-study cohort. The study reports superior performance for the developed framework compared to experienced radiographers and radiologists. One of the main advantages of the study is that a comprehensive evaluation of the cases by neurologists and neuroradiologists is reported and was used for comparisons. Four categories were classified: normal cognition, MCI, AD, non-AD dementia. Two tasks were evaluated first was cognition based on MRI – 3 class and then given dementia, AD or non-AD. Results are interpretable due to the use of SHAP. Overall conclusions are supported by the presented data.

Response 1: We sincerely thank the reviewer for noting that our study was *well-designed* and that *we presented a novel deep learning framework*. We also appreciate the reviewer for recognizing the value and advantages of our work.

Comment 2: It appears the authors have relied on normalisation to MNI space as a step for harmonization however, this might not be enough. Methods such as ComBat (Fortin et al. 2018) have been developed and applied for multi-site data harmonisation. This would be particularly important since it needs to be ruled out that the observed performance might be driven partly by site differences.

Response 2: We thank the reviewer for sharing the concern on MRI harmonization across multiple cohorts and suggesting the ComBat method to further reduce the effect of site/scanner difference. Fortin et al. demonstrated that the ComBat method is essential and vital to remove scanner effects in brain MRI radiomic study. However, in our study, we didn't use any radiomic-level features, for example, cortical thickness, as inputs to the model. To our best knowledge, the ComBat method is not suitable to harmonize the raw image modality.

The major variance of MRI scans comes from various voxel intensity distribution and geometric alignment. The intensities of voxels in MRI don't have clear physical meaning and thus make it intractable to compare 2 MRIs. Unifying the intensity of white matter, gray matter and cerebrospinal fluid to the same reference scan and removing the background noise thus became a critical step when processing MRIs collected with different scanners and from multiple cohorts. The skull removal step removed all background signals and set all background voxels to value zero. Then the FAST [1] approach from FSL was used to adjust the contrast across different tissue types and correct spatial intensity variation (i.e., bias-field). After the FAST step, a final z-score normalization was applied to eliminate the global intensity bias. The geometric variance was unified using a set of registration steps which were discussed in the Methods section in detail. Furthermore, in our response to subsequent comments from the reviewer, we demonstrate clustering analyses that we feel exclude the influence of site differences (see Response 3 below).

[1]: Zhang, Y. and Brady, M. and Smith, S. *Segmentation of brain MR images through a hidden Markov random field model and the expectation-maximization algorithm.* *IEEE Trans Med Imag*, 20(1):45-57, 2001.

[2]: Li, Yingping, et al. "Impact of Preprocessing and Harmonization Methods on the Removal of Scanner Effects in Brain MRI Radiomic Features." *Cancers* 13.12 (2021): 3000.

Comment 3: Though in figure SF11 the authors demonstrate no clear clustering, they should also demonstrate that the deep learning scheme is not sensitive to study/site effects. Hence, I suggest a further exploratory analysis where study ID will be included as a feature in the applied machine learning pipelines, especially when imaging data are included.

Response 3: We thank the reviewer for well-thought questions into whether cohort-specific information is unduly informing model predictions. We felt that it was critical in our work to include a comprehensive set of analyses aimed at excluding site-, cohort-, and scanner-specific biases in our overall training pipeline. Similarly, we believe it is important to exclude the possibility that the deep learning model is unduly learning confounding signatures instead of useful, disease-specific information. Therefore, we have expanded our tSNE analysis to include both raw MRIs and hidden layer activations from the penultimate layer of our CNN. We felt that this was important enough to be included as its own figure within the main manuscript; we now include three separate tSNE plots colored by i) cohort label ii) NACC ADRC ID and iii) scanner manufacturer. Further details may be found in Response 8 to Reviewer 1 within this document, and we also kindly ask the reviewer to see our updated Figure 2 with all tSNE plots on raw scans and hidden layer values alike. Overall, we have found that the sort of exploratory analysis suggested by the reviewer has yielded strong evidence that our deep learning model did *not* learn study or site effects, but rather kept its predictions to the clinical outcomes in question.

We also thank the reviewer for suggesting including the study ID as a non-imaging feature. We want to mention that the model was trained solely on the NACC cohort and tested on other cohorts. If study ID was included as a feature, the feature value within the training set will only contain a constant entity value, i.e., NACC. It is also difficult for the model to generalize well on the never seen cohorts IDs during the testing stages.

Comment 4: Line 42: There is a discordance in the number of regions reported here and in the caption of figure 3 (95 vs 57).

Response 4: We apologize for the confusion regarding this mismatch. To clarify, there are a total of 95 regions in the brain atlas that we used in this study. This corresponds to the number of regions used for graphical analysis of regional SHAP values with Pearson correlations. In Figure 4d-4e, we presented the networks of brain regions, where nodes represent regions and edges represent correlation between nodes, in both axial and sagittal projections. Because projecting a 3D structure into a plane causes nodes to overlap, we selectively presented some of the regions (nodes) in each view. In the sagittal view, we focused on visualizing the correlation between the temporal lobe, frontal lobe, parietal lobe, occipital lobe, cerebellum, and brainstem. More specifically, we merged the same structures from the left and right hemisphere as a single node in the sagittal projection, thus ending up with a total of 33 final nodes as defined in the Table S5. In the axial view, we excluded some of the structures that have been already shown in the sagittal view, for example, insula, the third ventricle etc. The focus of the axial view is to reveal the correlation between cerebrum structures from the left and right hemispheres. Our selection of the axial nodes yielded a total 57 regions as defined in the Table S9. We have clarified the region definitions in both the figure captions and the method section and thus resolved the discordance. Specifically, our updated Figure caption for this plot now contains the following:

Index	Adult brain atlas [region]	Sagittal view [node index]	axial view [node index]
1	T1_hypocampus_R	21	27
2	T1_hypocampus_L		28
3	T1_amygdala_R	18	26
4	T1_amygdala_L		25
5	T1_ventric temporal lobe medial part R		22
6	T1_ventric temporal lobe medial part L		24
7	T1_ventric temporal lobe lateral part R	19	23
8	T1_ventric temporal lobe lateral part L		25
9	T1_pontoparaventricular and subventric gyrus R		42
10	T1_pontoparaventricular and subventric gyrus L	23	41
11	T1_superior temporal gyrus medial part R		40
12	T1_superior temporal gyrus medial part L		46
13	T1_medial wall anterior temporal gyrus R		29
14	T1_medial wall anterior temporal gyrus L	22	40
15	T1_fusiform gyrus R		33
16	T1_fusiform gyrus L	20	36
17	caudate_R		
18	caudate_L	27	
19	basal ganglia including subthalamic nuclei	25	
20	basal ganglia including thalamus		
21	basal ganglia including thalamus R	29	
22	CC, lateral occipital sulcus L		28
23	CC, lateral occipital sulcus R	12	29
24	C10 anterior cingulate gyrus L		
25	C10 anterior cingulate gyrus R	1	
26	C10 posterior cingulate gyrus L		2
27	C10 posterior cingulate gyrus R	2	1
28	IL, middle frontal gyrus L		
29	IL, middle frontal gyrus R	4	
30	IL, posterior temporal lobe L		44
31	IL, posterior temporal lobe R		43
32	IL, superior gyrus L		
33	IL, superior gyrus R	14	
34	insula anterior L		30
35	insula anterior R	26	40
36	insula anterior lateral L		
37	insula anterior lateral R	36	
38	insula lateral L		35
39	insula lateral R	31	34
40	insula lateral L		37
41	insula lateral R	33	36

Index	Adult brain atlas [region]	Sagittal view [node index]	axial view [node index]
42	pedunculus L		52
43	pedunculus R		51
44	corpus callosum	24	31
45	Lateral ventricle including temporal horn R		9
46	Lateral ventricle including temporal horn L		
47	Lateral ventricle temporal horn R		10
48	Lateral ventricle temporal horn L	10	
49	Third ventricle	24	
50	IL, paracentral gyrus L		16
51	IL, paracentral gyrus R	6	15
52	IL, uncal gyrus L		18
53	IL, uncal gyrus R	7	17
54	IL, anterior orbital gyrus L		4
55	IL, anterior orbital gyrus R		3
56	IL, inferior frontal gyrus L	1	8
57	IL, inferior frontal gyrus R		7
58	IL, superior frontal gyrus L		22
59	IL, superior frontal gyrus R	8	23
60	IL, paracentral gyrus L		15
61	IL, paracentral gyrus R	15	
62	IL, superior parietal gyrus L		
63	IL, superior parietal gyrus R	19	
64	IL, lingual gyrus L		28
65	IL, lingual gyrus R	13	27
66	IL, middle L		24
67	IL, middle R	11	23
68	IL, medial orbital gyrus L		16
69	IL, medial orbital gyrus R		9
70	IL, lateral orbital gyrus L		9
71	IL, lateral orbital gyrus R	1	7
72	IL, paracentral orbital gyrus L		12
73	IL, paracentral orbital gyrus R		11
74	substantia nigra L		
75	substantia nigra R	32	
76	IL, subcallosal frontal cortex L		26
77	IL, subcallosal frontal cortex R		19
78	IL, subcallosal area L		
79	IL, subcallosal area R		
80	IL, pre-subcallosal frontal cortex L		14
81	IL, pre-subcallosal frontal cortex R	5	13
82	IL, superior temporal gyrus anterior part L		46

Comment 5: Line 94: It is unclear what imaging data have gone in the models

Response 5: We apologize for confusion regarding the input imaging data to our models. We adjusted the Methods section so that readers can clearly identify the input for each model. Overall, volumetric MRIs preprocessed according to our harmonization protocol (see Methods and Fig. S11) were sent into the 3D convolutional neural network. No other imaging data was used.

Comment 6: Lines 142-144. More details are needed about the manual alignment step in order for it to be reproducible.

Response 6: We appreciate the opportunity to provide further information about our MRI preprocessing pipeline. We have now adjusted our Methods section with an updated description, which reads as follows:

“To harmonize neuroimaging data between cohorts, we developed a pipeline of preprocessing operations (Fig. S11) that was applied in identical fashion to all MRIs used in our study. This pipeline broadly consisted of two phases of registration to a standard MNI-152 template. We describe Phase 1 as follows:

- *Scan axes were reconfigured to match the standard orientation of MNI-152 space.*
- *Using an automated thresholding technique, a 3D volume-of-interest within the original MRI was identified containing only areas with brain tissue.*
- *The volume-of-interest was skull-stripped to isolate brain pixels.*

- *A preliminary linear registration of the skull-stripped brain to a standard MNI-152 template was performed. This step approximated a linear transformation matrix from the original MRI space to the MNI-152 space.*

Phase 2 was designed to fine-tune the quality of linear registration and parcellate the brain into discrete regions. These goals were accomplished by the following steps:

- *The transformation matrix computed from linear registration in Phase 1 was applied to the original MRI scan.*
- *Skull stripping was once again performed after applying the linear registration computed from the initial volume of interest to isolate brain tissue from the full registered MRI scan.*
- *Linear registration was applied again to alleviate any misalignments to MNI-152 space.*
- *Bias field correction was applied to account for magnetic field inhomogeneities.*
- *The brain was parcellated by applying a nonlinear warp of the Hammersmith Adult brain atlas to the postprocessed MRI.*

All steps of our MRI-processing pipeline were conducted using FMRIB Software Library v6.0 (FSL) ([Analysis Group, Oxford University](https://www.fmrib.ox.ac.uk/fsl/)). The overall preprocessing workflow was inspired by the harmonization protocols of the UK Biobank (https://git.fmrib.ox.ac.uk/falmagro/UK_biobank_pipeline_v_1). We manually inspected the outcome of the MRI pipeline on each scan to filter out cases with poor quality or significant processing artifacts.”

Comment 7: Since interpretability of the results following deep learning is a key issue, more methodological details are needed about SHAP.

Response 7: We thank the reviewer for the opportunity to expand our explanation of the SHAP methodology. We have now added in a paragraph to our Methods section that reads as follows:

“SHAP is a unified framework for interpreting machine learning models which estimates the contribution of each feature by averaging over all possible marginal contributions to a prediction task.¹⁹ Though initially developed for game theory applications,²⁰ this approach may be used in deep learning-based computer vision by considering each image voxel or a network node as a unique feature. By assigning SHAP values to specific voxels or by mapping internal network nodes back to the native imaging space, heatmaps may be constructed over input MRIs.

Though a variety of methods exist for estimating SHAP values, we implemented a modified version of the DeepLIFT algorithm,²¹ which computes SHAP by estimating differences in model activations during backpropagation relative to a standard reference. We established this reference by integrating over a “background” of training MRIs to estimate a dataset-wide expected value. For each testing example, we then calculated SHAP values for the overall CNN model as well as for specific internal layers. Two sets of SHAP values were estimated for the COG and ADD tasks, respectively. SHAP values calculated over the full model were directly mapped back to native MRI pixels whereas those derived for internal layers were translated to the native imaging space via nearest neighbor interpolation.”

Comment 8: Results. The number of excluded scans due to the applied QC should be reported.

Response 8: We thank the reviewer for pointing out the unreported number of excluded scans and we added more details of the data inclusion criteria that we followed in this study in the beginning of the method section which reads as below:

*“Subjects from each cohort were eligible for study inclusion if they had at least one T1-weighted volumetric MRI scan within 6 months of an officially documented diagnosis. We additionally excluded all MRI scans with fewer than 60 slices. For subjects with multiple MRIs and diagnosis records within a 6-month period, we selected the closest pairing of neuroimaging and diagnostic label. Therefore, only one MRI per subject was used. For the NACC and the OASIS cohorts, we further queried all available variables relating to demographics, past medical history, neuropsychological testing, and functional assessments. We did not use the availability of non-imaging features to exclude individuals in these cohorts and used k-nearest neighbor imputation for any missing data fields. Our overall data inclusion workflow may be found in **Fig. S10**, where we reported the total number of subjects from each cohort before and after application of the inclusion criterion.”*

Comment 9: Though the fusion model improves performance on the non-parkinsonian dementias, there is no mention or explanation about a slightly worse performance in VD and FTD (Figures 1 and 2 - Lines 139-141)

Response 9: We added the relevant discussion on the comparison between the MRI model and the fusion model’s performance in the Discussion section. Please see below.

“Interestingly, it should be noted that the performance of a non-imaging model alone approached that of the fusion model. However, the inclusion of neuroimaging data was critical to enable verification of our modeling results by clinical criteria (e.g., cross-correlation with post-mortem neuropathology reports). Such confirmatory data sources cannot be readily assimilated to non-imaging models, thus limiting the ability to independently ground their performance in non-computational standards. Therefore, rather than viewing the modest contribution of neuroimaging to diagnostic accuracy as a drawback, we argue that our results suggest a path towards balancing demands for transparency with the need to build models using routinely collected clinical data. Models such as ours may be validated in high-resource areas where the availability of advanced neuroimaging aids interpretability; however, the set of non-imaging models that we developed can be easily used to perform diagnosis when only limited amounts of “traditional” clinical data are available.”

Comment 10: Discussion. As neuroradiologist 1 points, there wasn’t a consensus approach between radiologists. This should be reported as a limitation.

Response 10: We acknowledge that a consensus approach was not employed among radiologists. According to the work of Rosenkrantz [1] secondary interpretation of radiologic scans may indeed improve disease detection. However, we also feel that comparison to individual

practitioners may be viewed as a strength, given that consensus review may be less reflective of the unique approaches used by different neuroradiologists. Indeed, a key promise of deep learning within radiology is the chance to improve diagnosis in settings where significant variability may persist among experienced clinicians. Nevertheless, we have included the following prose within our Discussion:

“Lastly, although we have compared our model to the performance of individual neurologists and neuroradiologists, future studies may consider comparison to consensus reviews by teams of collaborating clinicians.”

[1] Rosenkrantz, Andrew B., et al. "Discrepancy rates and clinical impact of imaging secondary interpretations: a systematic review and meta-analysis." *Journal of the American College of Radiology* 15.9 (2018): 1222-1231.

Comment 11: It is unclear whether atypical AD cases existed in any of the cohorts, if this information is available it should be included and discussed in relation to the findings. This should also be discussed in the limitations.

Response 11: We thank the reviewer for their detailed comment. The data across the various studies that we utilized did not have atypical AD cases. We have updated the manuscript and included the comment as a limitation. Specifically, we cite Graff-Radford and colleagues [1] who note the epidemiology of atypical AD.

“Our cohorts also did not contain any confirmed cases of atypical AD, which is estimated to affect approximately 6% of elderly-onset cases and one-third of patients with early-onset disease [1].”

[1] Graff-Radford, Jonathan, et al. "New insights into atypical Alzheimer's disease in the era of biomarkers." *The Lancet Neurology* 20.3 (2021): 222-234.

Comment 12: The achieved performance and importance of the framework are discussed, however the discussion would benefit from an additional paragraph discussing the observed findings in figures 3 and 4.

Response 12: We thank the reviewer for inviting the opportunity for further elaboration upon Figures 3 and 4. Given that we have now submitted an additional figure with our revision, the original Figures 3 and 4 now correspond to updated Figures 4 and 5. We describe the findings in these visuals with the following additional material in the Discussion, making specific reference to recent concerns about explainability in AI:

“Furthermore, post-hoc analyses demonstrated that the performance of our machine learning models was grounded in well-established patterns of dementia-related neurodegeneration. Notably, network analyses evinced differing regional distributions of SHAP values between AD and nADD populations, which were most pronounced in areas such as the hippocampus, amygdala, and temporal lobes. The SHAP values in these regions also exhibited a strong correlation with atrophy ratings from neuroradiologists. Although recent work has shown that explainable machine learning methods may identify spurious correlations in

imaging data [1] we feel that our ability to link regional SHAP distributions to both anatomic atrophy and also semi-quantitative scores of A β amyloid, neurofibrillary tangles, and neuritic plaques links our modeling results to a gold-standard of postmortem diagnosis. More generally, our approach demonstrates a means by which to assimilate deep learning methodologies with validated clinical evidence in health care."

[1] DeGrave, Alex J., Joseph D. Janizek, and Su-In Lee. "AI for radiographic COVID-19 detection selects shortcuts over signal." *Nature Machine Intelligence* (2021): 1-10.

Reviewers' comments:

Reviewer #1 not re-engaged (re-review performed by Reviewer #3):

I appreciated the authors' efforts in responding to Reviewer 1, and I think they adequately responded to much of Reviewer 1's feedback. The following are some remaining points that the authors can improve upon.

1. Response to Reviewer 1, Comment 2: The main results of the paper are centered around the fusion model though, which relies heavily on MRI. While GPs have some experience with ordering MRIs, MRI is typically not the standard of care when it comes to diagnosing dementia within the primary care setting, and even if imaging is requested, CT is often more widely available. I understand the authors want to present their highest performing results in the main parts of the paper, but it's quite difficult to imagine deploying their fusion model in any clinical setting other than a specialist's practice. Maybe the authors can put more emphasis on using the non-imaging model in primary care settings in the discussion.

2. Response to Reviewer 1, Comment 8: The labels in the rebuttal that are given for the revised Figure 2 don't seem to match the figure, but I think the version in the manuscript and its associated figure caption are correct. It would be good for the authors to double check.

3. Response to Reviewer 1, Comment 9 & 12: I think a lot of readers would assume, like Reviewer 1 did, that APOE genotype would have been a feature that the authors would have included, so maybe explicitly listing the chosen features for the non-imaging model in the main text will clear that up. For example, in the Methods section, under Harmonization of non-imaging data, lines 507-508, the authors could include the variables there, like "demographics (e.g. age, sex), past medical history, neuropsychological test results (e.g. digit span, NPIQ, MMSE), and functional assessments (e.g. FAQ)". It's a bit interesting actually that APOE was not used as an input feature, when it is a common feature across similar work in the literature, so perhaps this will present an opportunity for the authors to provide an explanation for their choice.

4. Response to Reviewer 1, Comment 10: Can the authors please add a legend to explain the symbol markers in Figure 5c?

5. Response to Reviewer 1, Comment 11: The entire OASIS cohort is missing education. Did the authors impute education levels for each individual in OASIS? Are there any caveats to this, given that OASIS was the main test dataset, and especially given that education ranks quite high in the SHAP plots as an important feature for the predictions?

Reviewer #2 (Remarks to the Author):

I was Reviewer 2 in prior submission. The manuscript has addressed all my comments and I have no further comments.

Arman Eshaghi

Reviewer #3 (Remarks to the Author):

The manuscript by Qiu et al has appreciably improved, and the authors thoughtfully considered the reviewers' suggestions. I have some points that I'd like the authors to consider (and I would like to apologize to the authors for neglecting to mention some of them in the previous round of review).

Major comments:

- 1) It would have been helpful to have a figure that puts the ROC curves of the three different models together (MRI-only, non-imaging, fusion) in one plot for each task to make it easier for readers to compare the performance. The A panels of Figure 3 and supplementary figures S3 and S4 could have been combined into one, and same with the B panels of those same figures.
- 2) The performance metrics of the non-imaging and fusion models are very similar, where the fusion model (confidence intervals of AUCs ranged from 0.941 to 0.975 across COGnc and COGde for example) is only marginally better than the non-imaging model (confidence intervals of AUCs ranged from 0.931 to 0.973 across COGnc and COGde). Is there much added value in including the MRI, given how much it costs relative to neuropsychological testing? Since the authors mentioned that one motivation for having machine learning models is to aid neurological diagnoses in settings with low resources (e.g. remote or developing regions with few dementia specialists) in their introduction, it seems counter-intuitive to rely heavily on a relatively expensive modality like MRI. I did appreciate that the authors mentioned that any gains from the addition of the MRI were modest in the discussion. However, I wonder if the authors can expand their justification of including MRI as a critical modality for an Alzheimer's disease diagnostic tool when the majority of patients will be seen in low-resource settings, like in primary care, where MRI is not often considered.
- 3) Is it clinically useful, especially in a low-resource setting, to deploy a machine learning model (with or without MRI inputs) to classify cognitively normal vs impaired (COGnc task) when a brief screening test like the MMSE alone can do that? How do the COGnc models compare against simply thresholding scores on a cognitive test to classify cognitively normal vs impaired?
- 4) In the methods section, non-imaging model subsection, lines 552-554: "Like the MRI-only model, each non-imaging model was sequentially trained to complete the COG and the ADD tasks by calculating the DEMO and the ALZ scores, respectively." This phrase suggests that the non-imaging model produces ALZ and DEMO scores. However, in the Results section, lines 134-141 ("(i) MRI-only model: A convolutional neural network (CNN) that internally computed a continuous DEmentia MOdel ("DEMO") score to complete the COG task, as well as an ALZheimer's ("ALZ") score to complete the ADD task.") explains that the ALZ and DEMO scores are generated from the CNN of the MRI-only model. Please clarify this potential discrepancy.

Minor comments:

- 5) Figure 6, panel D: It would be nice to have a legend to show that the blue and red designate different hemispheres.
- 6) Supplementary Figure S5: Panels A and B are not labeled in the figure caption.
- 7) Results section, Deep learning model performance subsection, line 199: There seems to be a typo in the lower bound of the AUC confidence intervals for the COGde task of the non-imaging model for the NACC dataset: "COGde task, with AUC/PR pairs of 0.963 [CI: 0.9955, 0.971]/0.905 [0.888, 0.922] (NACC)".
- 8) Response to Reviewer 3, Comment 6: The authors stated that they removed the remark about visual similarity between SHAP values and neuropathological scores, but it is still present in the manuscript in lines 251-253.

Multimodal deep learning for Alzheimer's disease dementia assessment

Reviewer 1 (Re-Review Performed by Reviewer 3):

Comment 1: The main results of the paper are centered around the fusion model though, which relies heavily on MRI. While GPs have some experience with ordering MRIs, MRI is typically not the standard of care when it comes to diagnosing dementia within the primary care setting, and even if imaging is requested, CT is often more widely available. I understand the authors want to present their highest performing results in the main parts of the paper, but it's quite difficult to imagine deploying their fusion model in any clinical setting other than a specialist's practice. Maybe the authors can put more emphasis on using the non-imaging model in primary care settings in the discussion.

Response 1: We thank the reviewer for carefully considering the clinical implications of our modeling approach. While the proliferation of free-standing imaging centers and the strengthening of regional referral networks have facilitated the ability of GPs to order imaging studies for their patients, we certainly acknowledge that primary care providers may be less confident in interpreting the results of neuroimaging and are less likely than specialists to order these tests [1]. To this end, we appreciate the opportunity to point out the non-imaging model's utility in primary care settings and have taken the reviewer's suggestion to specifically mention it in our Discussion section. Therefore, we now include the following sentences within our main manuscript:

Interestingly, it should be noted that the performance of a non-imaging model alone approached that of the fusion model. However, the inclusion of neuroimaging data was critical to enable verification of our modeling results by clinical criteria (e.g., cross-correlation with post-mortem neuropathology reports). Such confirmatory data sources cannot be readily assimilated to non-imaging models, thus limiting the ability to independently ground their performance in non-computational standards. Therefore, rather than viewing the modest contribution of neuroimaging to diagnostic accuracy as a drawback, we argue that our results suggest a path towards balancing demands for transparency with the need to build models using routinely collected clinical data. Models such as ours may be validated in high-resource areas where the availability of advanced neuroimaging aids interpretability. Given that many physicians have difficulty entrusting medical decision-making to black box model in artificial intelligence,²⁸ grounding our machine learning results in the established neuroscience of dementia may help to facilitate clinical uptake. Nevertheless, we do note that our non-imaging model may be best suited for deployment among general practitioners (GPs) and in low-resource settings.

Comment 2: The labels in the rebuttal that are given for the revised Figure 2 don't seem to match the figure, but I think the version in the manuscript and its associated figure caption are correct. It would be good for the authors to double check.

Response 2: We thank the reviewer for pointing out the inconsistency between the figure and captions from the rebuttal version. We have double checked the correct correspondence between all figures and captions in the revised manuscript.

Comment 3: I think a lot of readers would assume, like Reviewer 1 did, that APOE genotype would have been a feature that the authors would have included, so maybe explicitly listing the chosen features for the non-imaging model in the main text will clear that up. For example, in the Methods section, under Harmonization of non-imaging data, lines 507-508, the authors could include the variables there, like “demographics (e.g. age, sex), past medical history, neuropsychological test results (e.g. digit span, NPIQ, MMSE), and functional assessments (e.g. FAQ)”. It's a bit interesting actually that APOE was not used as an input feature, when it is a common feature across similar work in the literature, so perhaps this will present an opportunity for the authors to provide an explanation for their choice.

Response 3: We appreciate the reviewer's suggestion to explicitly state which non-imaging features have been incorporated into our modeling approaches. With regards to our decision to not include APOE genotype as an input feature, we made this decision out of concern that APOE genotyping is not part of the standard clinical work-up for patients presenting with cognitive dysfunction. Although direct-to-consumer genetic screening services may provide this information to patients willing to pay for this information, its scope in clinical practice is currently limited [1]. Given that our goal in this work is to create a model that can be flexibly scaled to many different clinical practices and settings, we felt that this type of genetic information could be outside the current standard of practice among neurologists and GPs. We have clarified this point (along with the additional non-imaging features used in our modeling) with a sentence in the Methods section that reads as follows:

*To mimic a clinical neurology setting, we developed a non-imaging model using data that is routinely collected for dementia diagnosis. A full listing of these variables used as input may be found in **Box 2**. While some features such as genetic status (APOE e4 allele),⁵¹ or cerebrospinal fluid measures have great predictive value, we have purposefully not included them for model development because they are not part of the standard clinical work-up of dementia.*

Additionally, we feel that it would be in the best interest of this work to include an easily-accessible table in which readers can review all non-imaging variables used for modeling purposes. We have compiled these into a glossary box (Box 2), which we hope to include within the body of our main manuscript.

Box 2. Non-Imaging Features Used in Model Development

Demographics

Age
Gender
Education

Medical History

Family history of cognitive impairment
History of heart attack/cardiac arrest
History of atrial fibrillation
History of angioplasty/endarterectomy/cardiac stenting
History of cardiac bypass procedure
History of pacemaker
History of hypertension
History of hypercholesterolemia
History of heart failure
History of other cardiovascular disease
History of stroke
History of transient ischemic attack
History of seizures
History of traumatic brain injury
History of diabetes
History of vitamin B12 deficiency
History of thyroid disease
History of urinary incontinence
History of bowel incontinence
History of depression within preceding two years
History of depression greater than two years ago
History of other psychiatric disorder
History of alcohol use disorder
Has smoked >100 cigarettes in life
Total years smoking cigarettes
Packets of cigarettes smoked per day
History of other drug use

Neuropsychiatric Inventory

Delusions
Hallucinations
Agitation/Aggression
Dysphoria/Depression
Anxiety
Euphoria/Elation
Apathy/Indifference
Disinhibition
Irritability/Lability
Aberrant Motor Activity
Nighttime Behavior
Appetite/Eating

Neuropsychological Testing

Trail Making Test Part A/B
Boston Naming Test
Digit span backward trials correct
Digit span backward length
Digit span forward trials correct
Digit span forward length
Animals
Geriatric Depression Scale (GDS)
Logical memory immediate recall
Logical memory delayed recall
Mini Mental State Exam (MMSE)

Functional Activities

Paying Bills
Assembling tax records
Shopping alone

Playing a game
Meal preparation
Keeping track of current events
Paying attention to TV, books, or magazines
Remembering dates

[1] Choudhury, Parichita, Vijay K. Ramanan, and Bradley F. Boeve. "APOE ϵ 4 Allele Testing and Risk of Alzheimer Disease." *JAMA* 325.5 (2021): 484-485.

Comment 4: Can the authors please add a legend to explain the symbol markers in Figure 5c?

Response 4: We thank the reviewer for this suggestion, which will certainly improve the clarity of this figure. We added a legend to Figure 5c. The full Figure 5 may be found as below

Figure 5

Comment 5: The entire OASIS cohort is missing education. Did the authors impute education levels for each individual in OASIS? Are there any caveats to this, given that OASIS was the main test dataset, and especially given that education ranks quite high in the SHAP plots as an important feature for the predictions?

Response 5: We thank the reviewer for this astute observation regarding our imputation strategy and appreciate the opportunity to address this point. Indeed, given the extensive degree to which educational data in OASIS was missing, we elected to impute these values using a k-nearest neighbors approach. Certainly, this may be viewed as a limitation of our external validation strategy, and we are happy to acknowledge it as such within our Discussion if the reviewer wishes. However, we have also taken efforts to run additional non-imaging and fusion models in the OASIS cohort *without* education as an input variable in order to assess the degree to which our imputation impacted external performance. These results may be summarized in the ROC and PR curves below, which contrast the performance of our (a) non-imaging model **with** education included (b) non-imaging model **without** education (c) fusion model **with** education included (d) fusion model **without** education included. As in our original submission, Rows 1-3 in each subsection represent performance on the COG_{NC}, COG_{DE}, and ADD tasks, respectively.

We have chosen to include these curves as our new Supplementary Figure S4, which covers both fusion and non-imaging models. Within the body of our Results section, we also now include the following text:

Additionally, we demonstrate the performance of non-imaging and fusion models in the OASIS cohort (Fig. S4), both with and without education as an imputed variable. Given that education information was unavailable from OASIS, these results demonstrate negligible impact on external performance due to our data imputation strategy.

With our imputation strategy, it was certainly conceivable that our adjustment could have led to an underestimation of external performance by estimating missing information from one dataset (OASIS) on the basis of covariates in another (NACC). As may be observed, however, our imputation of education makes only a minimal impact on the model's performance in the OASIS test set. Therefore, we feel that this caveat of imputation is unlikely to have impacted the results that we have reported within the submitted manuscript.

Although our initial SHAP analyses from the NACC test set identified education as one of the important features, we feel that the continued strong performance of our model without this information speaks to its ability to derive accurate diagnostic information from a diffuse set of imaging and non-imaging data. In other words, despite an individual feature's importance relative to others, our framework remains capable of strong performance by harnessing the collective data available to it rather than relying upon a sub-selection of important information. Given that missing patient data is an omnipresent complication of patient care, we therefore feel that this ablation study demonstrates the power of our approach to flexibly adjust to a variety of clinical scenarios; patients with differing information availability may be easily accommodated,

saving clinicians the difficult and possibly error-prone task of divining key information from a complicated assortment of past medical records.

Reviewer 2:

Comment 1: The manuscript has addressed all my comments and I have no further comments.

Response 1: We thank the reviewer for their insightful comments on our original submission, and we are pleased to have met their expectations.

Reviewer 3:

Comment 1: The manuscript by Qiu et al has appreciably improved, and the authors thoughtfully considered the reviewers' suggestions. I have some points that I'd like the authors to consider (and I would like to apologize to the authors for neglecting to mention some of them in the previous round of review).

Response 1: We greatly appreciate the reviewer's assessment, and for suggesting additional points to further improve our manuscript. Not many note an apology for suggesting additional comments in the advanced rounds of the review process. This speaks to the generosity of the reviewer and we sincerely appreciate their time.

Comment 2: It would have been helpful to have a figure that puts the ROC curves of the three different models together (MRI-only, non-imaging, fusion) in one plot for each task to make it easier for readers to compare the performance. The A panels of Figure 3 and supplementary figures S3 and S4 could have been combined into one, and same with the B panels of those same figures.

Response 2: We thank the reviewer for this suggestion, which will certainly improve the visual and informational aesthetic of Figure 3. The revised A and B panels are now displayed as below:

Comment 3: The performance metrics of the non-imaging and fusion models are very similar, where the fusion model (confidence intervals of AUCs ranged from 0.941 to 0.975 across COGnc and COGde for example) is only marginally better than the non-imaging model (confidence intervals of AUCs ranged from 0.931 to 0.973 across COGnc and COGde). Is there much added value in including the MRI, given how much it costs relative to neuropsychological testing? Since the authors mentioned that one motivation for having machine learning models is to aid neurological diagnoses in settings with low resources (e.g., remote or developing regions with few dementia specialists) in their introduction, it seems counter-intuitive to rely heavily on

a relatively expensive modality like MRI. I did appreciate that the authors mentioned that any gains from the addition of the MRI were modest in the discussion. However, I wonder if the authors can expand their justification of including MRI as a critical modality for an Alzheimer's disease diagnostic tool when the majority of patients will be seen in low-resource settings, like in primary care, where MRI is not often considered.

Response 3: We very much appreciate the reviewer's perspective regarding the performance impact of MRI and its implications for clinical deployment. We feel that it is crucial to develop diagnostic tools that are capable of interfacing with all steps of standard clinical practice, from the primary care setting to specialized neurologists. Therefore, we feel that inclusion of MRI enables our framework to be a truly universal tool for clinicians across specialties-adapting to both the often-limited resources available to GPs, as well as the radiologic information that is central to specialist practice. In the case of the latter, we particularly note that the American Academy of Neurology [1] recommends MRI in the standard workup of patients with dementia; therefore, we believed that we would be remiss to create models without neuroimaging information.

Additionally, a key aspect of our study is the ability to provide interpretability in our deep learning models. In our view, a central piece of this aim (as well as a significant source of innovation and contribution to the literature from this work) is the ability to map computationally-derived predictions to established anatomic (e.g., atrophic) and histologic markers of neurodegenerative change. Without the inclusion of MRI data, these tasks become impossible. As currently stands, many physicians have difficulty entrusting medical decision-making to black box models in artificial intelligence [2]; consequently, we believe that the ability to ground our framework's predictions in the established neuroscience of dementia makes our work well-positioned to earn the confidence of clinician stakeholders across a variety of specialists.

Further, we thank the reviewer for the opportunity to expound upon these points beyond what was originally written in our Discussion section. We have elected to include additional writing about the necessity of including MRI in this work. Our writing is as follows.

Interestingly, it should be noted that the performance of a non-imaging model alone approached that of the fusion model. However, the inclusion of neuroimaging data was critical to enable verification of our modeling results by clinical criteria (e.g., cross-correlation with post-mortem neuropathology reports). Such confirmatory data sources cannot be readily assimilated to non-imaging models, thus limiting the ability to independently ground their performance in non-computational standards. Therefore, rather than viewing the modest contribution of neuroimaging to diagnostic accuracy as a drawback, we

argue that our results suggest a path towards balancing demands for transparency with the need to build models using routinely collected clinical data. Models such as ours may be validated in high-resource areas where the availability of advanced neuroimaging aids interpretability. Given that many physicians have difficulty entrusting medical decision-making to black box model in artificial intelligence,²⁸ grounding our machine learning results in the established neuroscience of dementia may help to facilitate clinical uptake. Nevertheless, we do note that our non-imaging model may be best suited for deployment among general practitioners (GPs) and in low-resource settings.

Comment 4: Is it clinically useful, especially in a low-resource setting, to deploy a machine learning model (with or without MRI inputs) to classify cognitively normal vs impaired (COGnc task) when a brief screening test like the MMSE alone can do that? How do the COGnc models compare against simply thresholding scores on a cognitive test to classify cognitively normal vs impaired?

Response 4: We thank the reviewer for posing the valuable question of this model’s necessity in low-resource settings given the availability of in-office neuropsychiatric testing. While screening tests such as the MMSE are capable of delineating NC, MCI, and AD patients, we do note that their usage by PCPs remains highly variable according to nationwide surveys of US physicians [1,2] and that even “low tech” cognitive batteries often require specialized neuropsychiatric clinicians to perform properly. Moreover, these preliminary tests are unable to infer the etiology of a major neurocognitive disorder, thus requiring transfers of care from generalist to specialist clinics that often delay timely access to care [3]. Conversely, our system provides strong performance in identifying both AD and non-AD causes of dementia after performing the initial COGnc task, thereby providing an all-in-one solution that could help to alleviate inefficiencies in referral pipelines. For these reasons, we still believe that having a system that can quickly perform a reasonably-accurate differential diagnosis regardless of clinical expertise would be a significant benefit to primary care physicians in low-resource areas.

Nevertheless, we value the reviewer’s suggestion to compare our various machine learning models to simple thresholding of common neuropsychological tests, and we agree that it is important to address this point. Therefore, for the COGnc, COGde, and ADD tasks, we calculated the area under ROC (AUC) and area under precision-recall curve (AP) that would result from testing all possible numerical thresholds of MMSE. We compared these results to the values obtained from our non-imaging, MRI-only, and fusion models respectively, and summarized these metrics in the table below. Red text indicates instances in which simple MMSE thresholding exceeded the diagnostic performance of the deep learning model.

Model	COGnc task AUC	COGnc task AP	COGde task AUC	COGde task AP	ADD task AUC	ADD task AP
-------	-------------------	------------------	-------------------	------------------	-----------------	----------------

MMSE Threshold	0.881	0.848	0.931	0.814	0.616	0.896
Non-Imaging Model	0.936	0.936	0.963	0.905	0.717	0.926
MRI-Only	0.844	0.830	0.869	0.712	0.766	0.934
Fusion Model	0.945	0.946	0.971	0.917	0.773	0.938

The reviewer may note that the thresholding approach outperforms an MRI-only model on the COG_{NC} and COG_{DE} task. Overall, we anticipated this result given that focused neurocognitive testing such as the MMSE is specifically designed to separate NC, MCI, and DE cases. However, we would also like to point out that simple thresholding was otherwise unable to match the performance of any other models on these three tasks, and that the MRI-only still outperformed the MMSE in classifying AD vs. nADD etiologies.

Lastly, we felt that it would be in the best interest of the manuscript to compare our deep learning models to simple thresholding of *all* available neuropsychological testing available in the NACC cohort. To this end, we have repeated simple thresholding with 33 *additional* tests. The results from these experiments are summarized below, and we now wish to include these numbers as our new Supplementary Table S8. We thank the reviewer for bringing up this important point, which will once again help to highlight the potential of our deep learning approaches in the context of current clinical standards.

Lastly, we have included a brief summary of our simple thresholding experiments within the Methods section. Specifically, we have now added the following:

Additionally, to compare our machine learning models to routine neuropsychological assessments, we performed the COG_{NC}, COG_{DE}, and ADD tasks using all possible whole number cutoffs of neuropsychiatric test scores available in the NACC dataset. Following this approach, we performed simple thresholding for binary classifications.

Supplementary Table S8

Variable	COG _{NC} task AUC	COG _{NC} task AP	COG _{DE} task AUC	COG _{DE} task AP	ADD task AUC	ADD task AP
trailA	0.783	0.79	0.817	0.587	0.52	0.877
trailB	0.818	0.839	0.853	0.564	0.532	0.869

boston	0.791	0.762	0.825	0.59	0.569	0.887
digitB	0.725	0.719	0.753	0.458	0.533	0.891
digitBL	0.704	0.69	0.735	0.413	0.522	0.884
digitF	0.66	0.649	0.684	0.383	0.528	0.881
digitFL	0.632	0.624	0.654	0.329	0.54	0.885
animal	0.839	0.824	0.878	0.702	0.501	0.869
gds	0.647	0.633	0.6	0.275	0.608	0.895
lm_imm	0.872	0.86	0.907	0.722	0.638	0.913
lm_del	0.895	0.886	0.916	0.706	0.713	0.93
mmse	0.881	0.848	0.931	0.814	0.616	0.896
npiq_DEL	0.545	0.543	0.58	0.339	0.522	0.871
npiq_HALL	0.526	0.533	0.544	0.294	0.55	0.878
npiq_AGIT	0.597	0.574	0.628	0.357	0.501	0.86
npiq_DEPD	0.588	0.57	0.6	0.301	0.523	0.872
npiq_ANX	0.608	0.582	0.642	0.348	0.539	0.877
npiq_ELAT	0.513	0.527	0.516	0.246	0.508	0.868
npiq_APA	0.623	0.59	0.67	0.417	0.58	0.887

npiq_DISN	0.556	0.55	0.569	0.299	0.566	0.882
npiq_IRR	0.603	0.578	0.607	0.321	0.52	0.87
npiq_MOT	0.559	0.551	0.589	0.338	0.528	0.873
npiq_NITE	0.567	0.554	0.577	0.307	0.552	0.878
npiq_APP	0.575	0.561	0.595	0.32	0.541	0.875
faq_BILLS	0.794	0.742	0.928	0.79	0.511	0.859
faq_TAXES	0.807	0.762	0.936	0.801	0.522	0.872
faq_SHOPPING	0.733	0.676	0.88	0.752	0.538	0.875
faq_GAMES	0.706	0.673	0.841	0.689	0.571	0.879
faq_STOVE	0.632	0.602	0.73	0.55	0.53	0.878
faq_MEALPREP	0.709	0.677	0.853	0.71	0.521	0.885
faq_EVENTS	0.75	0.687	0.867	0.723	0.54	0.874
faq_PAYATTN	0.736	0.674	0.846	0.684	0.518	0.872
faq_REMDATES	0.82	0.756	0.925	0.776	0.527	0.871
faq_TRAVEL	0.781	0.716	0.908	0.766	0.501	0.864

Comment 5: In the methods section, non-imaging model subsection, lines 552-554: “Like the MRI-only model, each non-imaging model was sequentially trained to complete the COG and the ADD tasks by calculating the DEMO and the ALZ scores, respectively.” This phrase suggests that the non-imaging model produces ALZ and DEMO scores. However, in the Results section, lines 134-141 (“(i) MRI-only model: A convolutional neural network (CNN) that

internally computed a continuous DEmentia MOdel (“DEMO”) score to complete the COG task, as well as an ALZheimer’s (“ALZ”) score to complete the ADD task.”) explains that the ALZ and DEMO scores are generated from the CNN of the MRI-only model. Please clarify this potential discrepancy.

Response 5: We very much appreciate the reviewer’s attention to this aspect of our methodology which, upon further consideration, will benefit from additional clarification. Both the MRI-only and non-imaging models compute respective versions of the DEMO/ALZ scores. In isolation, these respective versions can be used for all classification tasks. In the fusion model, however, the MRI-only DEMO/ALZ scores (derived from the CNN) are *recycled* as a feature to be used alongside non-imaging variables. In order to clarify these points, we have added the following text to our manuscript:

*We also created three separate models: (i) MRI-only model: A convolutional neural network (CNN) that internally computed a continuous DEmentia MOdel (“DEMO”) score to complete the COG task, as well as an ALZheimer’s (“ALZ”) score to complete the ADD task. (ii) Non-imaging model: A traditional machine learning classifier that took as input only scalar-valued clinical variables from demographics, past medical history, neuropsychological testing, and functional assessments. As in the MRI-only model, the non-imaging model also computed the DEMO and the ALZ scores from which the COG and the ADD tasks could be completed. We tested multiple machine learning architectures for these purposes and ultimately selected a CatBoost model as our final non-imaging model architecture. (iii) Fusion model: This framework linked a CNN to a CatBoost model. With this approach, the DEMO and the ALZ scores computed by the CNN were recycled and used alongside available clinical variables. The CatBoost model then recalculated these scores in the context of the additional non-imaging information. We provide definitions of our various prediction tasks, cognitive metrics, and model types within **Box 1**. Further details of model design may be found within the *Methods*.*

Comment 6: Figure 6, panel D: It would be nice to have a legend to show that the blue and red designate different hemispheres.

Response 6: We thank the reviewer for pointing out the missing legend for different hemispheres. The adjust Figure 6 now looks as below:

Figure 6

Comment 6: Supplementary Figure S5: Panels A and B are not labeled in the figure caption.

Response 6: We thank the reviewer for this attention to detail. We made these changes.

Comment 7: Results section, Deep learning model performance subsection, line 199: There seems to be a typo in the lower bound of the AUC confidence intervals for the COGde task of

the non-imaging model for the NACC dataset: “COGde task, with AUC/PR pairs of 0.963 [CI: 0.9955, 0.971]/0.905 [0.888, 0.922] (NACC)”.

Response 7: We thank the reviewer for pointing out this typo. After carefully reexamining the script that was used for generating confidence intervals, we found a mistake in the degree of freedom that should be used and made a correction on the confidence intervals formula. All confidence intervals reported in the manuscript or tables have now been corrected. Additionally, we also added a brief description on how we estimated the confidence intervals within the methods section which reads as below:

All statistical analyses were conducted at a significance level of 0.05. Confidence intervals for model performance were calculated by assuming a normal distribution of AUC and AP values across cross-validation experiments.

Comment 8: Response to Reviewer 3, Comment 6: The authors stated that they removed the remark about visual similarity between SHAP values and neuropathological scores, but it is still present in the manuscript in lines 251-253.

Response 8: We thank the reviewer for noting this and apologize that the change was not made. We have double-checked our reference to visual similarity in our updated submission, and it has now been removed.

Reviewers' comments:

Reviewer #3 (Remarks to the Author):

Reviewer 1 (Re-review by Reviewer 3):

The authors have addressed my comments, and the paper is much improved, but I have one outstanding concern.

1. Re: Education in OASIS. I appreciate the additional analyses that the authors presented where they evaluated their models on OASIS with and without the imputed education levels. I have to note that, as a researcher who has personally performed analyses on OASIS, education is a variable that is publicly available from this dataset, so the imputation was not necessary. Furthermore, the preprint describing the release of the OASIS-3 cohort explicitly mentions education as a demographic variable that is available for download (<https://www.medrxiv.org/content/medrxiv/early/2019/12/15/2019.12.13.19014902.full.pdf>). In previous rounds of review, the authors mentioned that they obtained OASIS through GAAIN, which may explain our different experiences, because I obtained access and downloaded the OASIS-3 dataset from <https://www.oasis-brains.org/> and the CENTRAL XNAT platform (<https://central.xnat.org/>). I understand that it takes a lot of effort to aggregate all these different datasets together and perhaps there were unfortunate mistakes along the way, especially if multiple team members were involved in the curation. Given that the authors reported they were missing education in this dataset and they imputed that feature, I'm worried that readers who have also worked with OASIS may question the validity of the current paper's analyses on OASIS or that readers may develop even worse uncharitable opinions (e.g. academic misconduct).

Reviewer 3:

The authors have addressed all of my comments and I have no further comments.

Response to Reviewers:

In this document, we address a remaining question relating to our usage of education information from the OASIS-3 dataset. Our steps to address this matter have resulted in only minor adjustments to our results and have not altered the overall conclusions drawn in the remainder of our manuscript. We are otherwise pleased to see that the rest of our Reviewers' concerns have been addressed.

Below, we detail the exact steps that we have taken to meet the Reviewer's recommendations with regards to OASIS-3, and present updated figures and tables with the updated performance of our machine learning models.

Reviewer 1 (Re-review by Reviewer 3):

Comment: I appreciate the additional analyses that the authors presented where they evaluated their models on OASIS with and without the imputed education levels. I must note that, as a researcher who has personally performed analyses on OASIS, education is a variable that is publicly available from this dataset, so the imputation was not necessary. Furthermore, the preprint describing the release of the OASIS-3 cohort explicitly mentions education as a demographic variable that is available for download (<https://www.medrxiv.org/content/medrxiv/early/2019/12/15/2019.12.13.19014902.full.pdf>). In previous rounds of review, the authors mentioned that they obtained OASIS through GAAIN, which may explain our different experiences, because I obtained access and downloaded the OASIS-3 dataset from <https://www.oasis-brains.org/> and the CENTRAL XNAT platform (<https://central.xnat.org/>).

Response: We are greatly appreciative that the Reviewer has lent their personal perspective in working with the OASIS-3 dataset to our work, thereby empowering us to fully address the question of imputing education. Following the data access instructions in the links below, we have indeed found that education information is available from OASIS-3 data when accessed via the CENTRAL XNAT platform. We sincerely apologize that our unfamiliarity with this portal led us to initially report this variable as missing; indeed, we must acknowledge that this was an error within our initial submission because we focused on other publicly available sources to collect OASIS data (i.e., GAAIN).

In response to the Reviewer's helpful guidance, we acquired education data from OASIS-3 and repeated all runs of the fusion- and non-imaging models that previously made use of this variable in imputed form. The results of these experiments are further detailed below.

Comment: Given that the authors reported they were missing education in this dataset, and they imputed that feature, I'm worried that readers who have also worked with OASIS may question the validity of the current paper's analyses on OASIS or that readers may develop even worse uncharitable opinions (e.g., academic misconduct).

Response: We are thankful that the Reviewer has the best interests of our work in mind when anticipating the way in which it will be received by readers of *Nature Communications*. It is of the utmost importance

to us that there be no questions regarding our academic integrity in conducting these studies, and we feel grateful for the chance to revise our manuscript with this issue addressed with all required seriousness.

With these considerations in mind, we have now used the education information from the CENTRAL XNAT platform to repeat all external validation experiments with the fusion and non-imaging models in the OASIS-3 dataset. Inclusion of this data has fortunately not impacted our results in a major way and our overall conclusions remain intact. However, we have revised a few figures and tables to transparently report the updated performance. Below, we include the Tables and Figures that required updating. All necessary metrics (e.g., AUC and AP values) have been adjusted as appropriate when mentioned within the body of the manuscript as well.

Table 1: We have updated Table 1 to reflect the mean and distribution of education in years from the OASIS-3 dataset. An excerpted version of our updated table, reflecting this change, is supplied below with the adjusted column highlighted in red. The full table including other datasets is supplied in the updated version of our manuscript with this submission.

Dataset (group) [subjects]		Age Mean ±std	Gender Male (percent)	Education In years Mean ±std	Race (White; Black; Asian; Indian; Pacific; Multi-race)	ApoE4 Positive (percent)	MMSE Mean ±std	CDR Mean ±std	MOCA Mean ±std
OASIS	NC [n=424]	71.34 ±9.43	164 (38.70%)	15.79 ±2.62 [^]	(53, 18, 1, 0, 0, 0) [^]	121 (29.88%)	28.99 ±1.25 [^]	0.00 ±0.02	N.A.
	MCI [n=27]	75.04 ±7.25	14 (51.85%)	15.19 ±2.76	(4, 1, 0, 0, 0, 0) [^]	9 (36.00%)	28.15 ±1.67	0.52 ±0.09	N.A.
	AD [n=193]	76.01 ±8.01	108 (55.96%)	14.68 ±3.09	(35, 9, 0, 0, 0, 0, 0) [^]	102 (56.98%)	23.84 ±4.17	0.77 ±0.33	N.A.
	Non-AD [n=22]	72.64 ±8.77	16 (72.73%)	15.00 ±2.91	(6, 0, 0, 0, 0, 0) [^]	8 (47.06%)	24.14 ±4.69 [^]	0.75 ±0.47	N.A.
	p-value	<0.001	<0.001	<0.001	0.810	<0.001	<0.001	<0.001	N.A.

Table 2: We have adjusted Table 2b to reflect update Fusion model performance in the OASIS-3 dataset.

(a)

	COG	COGNC	COGMCI	COGDE	ADD	4-way
Accuracy	0.782±0.011 [0.769-0.796]	0.856±0.008 [0.846-0.866]	0.790±0.010 [0.777-0.803]	0.919±0.006 [0.912-0.926]	0.806±0.033 [0.765-0.847]	0.748±0.012 [0.734-0.763]
F-1	0.752±0.013 [0.736-0.768]	0.862±0.009 [0.852-0.873]	0.566±0.022 [0.539-0.593]	0.827±0.012 [0.812-0.842]	0.884±0.019 [0.860-0.908]	0.611±0.027 [0.577-0.645]
Sensitivity	0.752±0.013 [0.736-0.768]	0.863±0.016 [0.843-0.883]	0.563±0.032 [0.524-0.603]	0.831±0.020 [0.806-0.856]	0.878±0.019 [0.854-0.901]	0.612±0.029 [0.575-0.648]
Specificity	0.886±0.006 [0.879-0.893]	0.848±0.014 [0.831-0.865]	0.863±0.013 [0.846-0.880]	0.946±0.007 [0.937-0.955]	0.417±0.140 [0.244-0.590]	0.906±0.004 [0.900-0.911]
MCC	0.638±0.018 [0.616-0.660]	0.711±0.016 [0.691-0.732]	0.428±0.027 [0.395-0.462]	0.775±0.016 [0.755-0.794]	0.283±0.139 [0.110-0.456]	0.517±0.031 [0.478-0.555]

(b)

	COG	COGNC	COGMCI	COGDE	ADD	4-way
Accuracy	0.769±0.019 [0.745-0.793]	0.862±0.013 [0.845-0.878]	0.780±0.022 [0.753-0.808]	0.895±0.005 [0.889-0.901]	0.786±0.014 [0.769-0.804]	0.720±0.019 [0.696-0.745]
F-1	0.613±0.008 [0.603-0.622]	0.884±0.013 [0.867-0.900]	0.143±0.006 [0.136-0.150]	0.811±0.011 [0.797-0.825]	0.875±0.010 [0.863-0.886]	0.480±0.011 [0.466-0.494]
Sensitivity	0.658±0.008 [0.648-0.668]	0.825±0.026 [0.794-0.857]	0.452±0.049 [0.391-0.513]	0.697±0.019 [0.673-0.721]	0.832±0.019 [0.809-0.855]	0.518±0.006 [0.511-0.525]
Specificity	0.903±0.005 [0.897-0.909]	0.926±0.009 [0.915-0.936]	0.794±0.025 [0.764-0.825]	0.990±0.004 [0.985-0.995]	0.382±0.068 [0.297-0.466]	0.914±0.004 [0.909-0.918]
MCC	0.535±0.009 [0.524-0.546]	0.727±0.020 [0.703-0.752]	0.118±0.010 [0.105-0.131]	0.761±0.011 [0.748-0.774]	0.165±0.043 [0.112-0.218]	0.411±0.011 [0.398-0.424]

Table S2: We have adjusted Table S2b to reflect the performance of a non-imaging CatBoost model in OASIS-3 with Education data.

(a)

	COG	COGNC	COGMCI	COGDE	ADD	4-way
Accuracy	0.782±0.011 [0.769-0.796]	0.856±0.008 [0.846-0.866]	0.790±0.010 [0.777-0.803]	0.919±0.006 [0.912-0.926]	0.806±0.033 [0.765-0.847]	0.748±0.012 [0.734-0.763]
F-1	0.752±0.013 [0.736-0.768]	0.862±0.009 [0.852-0.873]	0.566±0.022 [0.539-0.593]	0.827±0.012 [0.812-0.842]	0.884±0.019 [0.860-0.908]	0.611±0.027 [0.577-0.645]
Sensitivity	0.752±0.013 [0.736-0.768]	0.863±0.016 [0.843-0.883]	0.563±0.032 [0.524-0.603]	0.831±0.020 [0.806-0.856]	0.878±0.019 [0.854-0.901]	0.612±0.029 [0.575-0.648]
Specificity	0.886±0.006 [0.879-0.893]	0.848±0.014 [0.831-0.865]	0.863±0.013 [0.846-0.880]	0.946±0.007 [0.937-0.955]	0.417±0.140 [0.244-0.590]	0.906±0.004 [0.900-0.911]
MCC	0.638±0.018 [0.616-0.660]	0.711±0.016 [0.691-0.732]	0.428±0.027 [0.395-0.462]	0.775±0.016 [0.755-0.794]	0.283±0.139 [0.110-0.456]	0.517±0.031 [0.478-0.555]

(b)

	COG	COGNC	COGMCI	COGDE	ADD	4-way
Accuracy	0.769±0.019 [0.745-0.793]	0.862±0.013 [0.845-0.878]	0.780±0.022 [0.753-0.808]	0.895±0.005 [0.889-0.901]	0.786±0.014 [0.769-0.804]	0.720±0.019 [0.696-0.745]
F-1	0.613±0.008 [0.603-0.622]	0.884±0.013 [0.867-0.900]	0.143±0.006 [0.136-0.150]	0.811±0.011 [0.797-0.825]	0.875±0.010 [0.863-0.886]	0.480±0.011 [0.466-0.494]
Sensitivity	0.658±0.008 [0.648-0.668]	0.825±0.026 [0.794-0.857]	0.452±0.049 [0.391-0.513]	0.697±0.019 [0.673-0.721]	0.832±0.019 [0.809-0.855]	0.518±0.006 [0.511-0.525]
Specificity	0.903±0.005 [0.897-0.909]	0.926±0.009 [0.915-0.936]	0.794±0.025 [0.764-0.825]	0.990±0.004 [0.985-0.995]	0.382±0.068 [0.297-0.466]	0.914±0.004 [0.909-0.918]
MCC	0.535±0.009 [0.524-0.546]	0.727±0.020 [0.703-0.752]	0.118±0.010 [0.105-0.131]	0.761±0.011 [0.748-0.774]	0.165±0.043 [0.112-0.218]	0.411±0.011 [0.398-0.424]

Figure 3: We have adjusted Figure 3b to demonstrate model performance across our various classification tasks using the OASIS-3 education data.

Figure S2: We have adjusted the Figure S2b, which demonstrates the performance of the fusion model in the OASIS-3 dataset.

Figure S4: We have adjusted Figure S4b (previously this was Fig. S5), which assesses the performance of the non-imaging and fusion models with various feature combinations in the OASIS-3 dataset.

Reviewers' comments:

Reviewer #3 (Remarks to the Author):

The authors have adequately addressed my previous comments and I have no further comments.